# TimeLAVA: Learning-Agnostic Valuation for Time Series Data

## Abstract

Valuing temporal segments and individual time points within time series is crucial for tasks like data curation and robust learning, yet poses unique challenges. Existing methods often fail in this domain because they ignore the critical factors determining a segment's value, such as local patterns, temporal dependencies, and the broader distributional context. To address this, we introduce TimeLAVA, a learning-agnostic framework that values time series segments by quantifying their *marginal* contribution to minimizing the distributional discrepancy between evaluated and reference data. The core of this approach is a novel Selective Wavelet-based Wasserstein ($\mathcal{W}_{\text{SW}}$) discrepancy. This dissimilarity measure integrates multi-scale wavelet transforms to capture localized, intra-segment patterns. Additionally, it leverages unbalanced optimal transport to robustly handle non-stationarity and distributional shifts between the sets of segments. The intrinsic value of each segment is then efficiently derived via a sensitivity analysis of the $\mathcal{W}_{\text{SW}}$ discrepancy, and point-wise values are subsequently aggregated from these segment values. We provide theoretical guarantees linking our segment-based valuation to model-agnostic generalization and demonstrate its robustness. Empirical validation across diverse real-world datasets shows TimeLAVA significantly outperforming baselines at identifying influential and harmful temporal segments for applications like anomaly detection, data pruning, and label noise detection.

## 1 Introduction

Time series data forms the backbone of modern decision-making systems. In intensive care units, continuous monitoring of physiological signals informs life-critical interventions (Celi et al., 2013; Johnson et al., 2016); in financial markets, millisecond-level price fluctuations guide trading strategies (Franses & Van Dijk, 2000; Sezer et al., 2020); in industrial IoT systems, sensor streams enable predictive maintenance to avert costly failures (Carvalho et al., 2019). In all these domains, identifying the temporal segments most influential to downstream tasks, including both informative patterns and anomalies, is essential for building accurate predictive models and ensuring reliable operational decisions (Hamilton, 2020; Pang et al., 2021). This raises a central question: *how can we quantify the intrinsic value of individual time points or temporal segments in time series data?*

Valuing time series data presents a distinct set of challenges stemming from its sequential nature. Observations in sequential data are linked through *temporal dependencies*, meaning their value depends not only on their content but also on the surrounding context and the order in which they appear. This intrinsic structure is further complicated by the fact that real-world time series are often *non-stationary*: regime shifts, seasonal effects, and evolving dynamics can alter statistical properties over time. Furthermore, valuable information is often distributed across *multiple temporal scales*, with long-term cyclic trends and short-lived transient events contributing in complementary ways. In addition, practical constraints demand that any valuation method be able to process *large-scale*, high-frequency datasets efficiently while still preserving fine-grained insight.

Existing data valuation methods, predominantly designed for i.i.d. settings, prove ill-suited for these challenges. Model-dependent approaches like Influence Functions (Koh & Liang, 2017) measure data value relative to specific models. While TimeInf (Zhang et al., 2025) extends this to time series through temporal blocking, it remains fundamentally model-specific and assumes stationarity. Learning-agnostic methods using optimal transport, such as LAVA (Just et al., 2023) and SAVA (Kessler et al.,

Figure 1: **Overview of the TIMELAVA Framework.** Given a *reference* time series, TIMELAVA assigns a value to each segment of an *evaluated* series. The pipeline consists of: (1) segmenting both series, (2) extracting multi-scale wavelet features, (3) computing wavelet-based distances to form a cost matrix, (4) solving the $W_{SW}$ optimization problem to obtain optimal dual potentials, and deriving segment-level values from these potentials. The resulting valuations support applications such as anomaly detection, data pruning, and label noise detection.

2025) for i.i.d. data, define value through distributional similarity but are fundamentally unsuited for time series due to temporal dependencies and non-stationary dynamics. This leaves a critical gap: *the absence of a valuation framework that is simultaneously learning-agnostic, robust to non-stationarity, and sensitive to multi-scale temporal patterns.*

**Contributions.** To address this gap, we introduce TIMELAVA, a *novel learning-agnostic framework* that values time series segments by quantifying their *marginal* contribution to minimizing distributional discrepancy between training and reference data. Our Selective Wavelet-based Wasserstein ($(\mathcal{W}_{SW})$) discrepancy captures localized temporal patterns via multi-scale wavelets while maintaining robustness to non-stationarity through unbalanced optimal transport (Chizat et al., 2018b). Building upon this purpose-built discrepancy, we develop an efficient valuation approach via sensitivity analysis of the $\mathcal{W}_{SW}$ dual, providing interpretable, segment-level contributions without costly retraining. We establish theoretical guarantees for our approach and demonstrate its empirical superiority, showing robust and consistent gains over strong baselines across diverse real-world datasets from healthcare, finance, and Internet of Things.

## 2 RELATED WORK

**Model-Dependent Valuation.** Model-dependent methods quantify data value relative to specific predictive models. Leave-one-out retraining provides direct performance measurements but is computationally prohibitive for large datasets. Shapley value methods (Ghorbani & Zou, 2019; Jia et al., 2019b; Kwon & Zou, 2021) attribute value through marginal contributions across data subsets, though exact computation requires exponential evaluations. Influence Functions (Hampel, 1974; Koh & Liang, 2017) estimate how reweighting a data point perturbs model parameters without retraining, with TIMEINF (Zhang et al., 2025) extending this to time series through temporal blocking. However, all model-dependent approaches remain fundamentally tied to specific model architectures and training procedures, limiting their generalizability across tasks. Other contemporary works, such as RIOT (Kraus et al., 2025), take a model-centric perspective by constraining explanations through human-in-the-loop feedback to improve model reliability.

**Learning-Agnostic Valuation.** Learning-agnostic approaches aim to quantify intrinsic data value independent of specific models. The most prominent paradigm uses distributional similarity via optimal transport: LAVA (Just et al., 2023) and its memory-efficient variant SAVA (Kessler et al., 2025) values i.i.d. data points by their contribution to reducing Wasserstein distance between training and validation distributions, providing efficient model-agnostic estimates that generalize across tasks. However, these frameworks assume independent samples, fundamentally unsuited for time series where value depends on temporal context, and evolving non-stationary dynamics.

**Optimal Transport for Time Series.** While optimal transport (OT) provides a principled framework for comparing distributions (Villani et al., 2008; Peyré et al., 2019), its application to sequential data requires careful cost function design. Classical approaches like Dynamic Time Warping (Berndt & Clifford, 1994) align sequences but lack the distributional perspective of OT. Recent work has incorporated unbalanced OT to address non-stationarity; for example, spectral-domain costs have been used for time-series imputation (Wang et al., 2025) to improve robustness to regime shifts.

However, frequency-domain approaches remain insensitive to transient events and time-varying patterns critical for data valuation.

## 3 PRELIMINARIES

### 3.1 OPTIMAL TRANSPORT

OT provides a powerful framework for comparing probability distributions, which quantifies the minimal effort required to reconfigure a source distribution into a target distribution (Villani et al., 2008; Peyré et al., 2019). Consider two discrete probability measures, a source $\mu_t = \sum_{i=1}^{n} a_i \delta_{z_i}$ and a target $\mu_v = \sum_{j=1}^{m} b_j \delta_{z_j'}$, defined by locations $(z_i, z_j')$ and their corresponding probability masses (weights $\mathbf{a}, \mathbf{b}$). The cost of moving a unit of mass from a source location $z_i$ to a target location $z_j'$ is encoded in a cost matrix $\mathbf{C}$. The goal is to find an optimal transport plan $\mathbf{T}$, where each entry $T_{ij} \geq 0$ specifies the amount of mass to move from $z_i$ to $z_j'$. The Wasserstein distance is the minimal total cost to transport all mass from the source to the target:

$$\mathcal{W}(\mu_t, \mu_v) = \min_{\mathbf{T} \in \Pi(\mu_t, \mu_v)} \langle \mathbf{T}, \mathbf{C} \rangle, \tag{1}$$

where the transport plan $\mathbf{T}$ must be in the set of valid plans $\Pi(\mu_t, \mu_v)$, which enforces mass conservation. This ensures that the total mass transported out of each source location $i$ equals its original mass $a_i$ ($\mathbf{T}\mathbf{1}_m = \mathbf{a}$), and the total mass transported into each target location $j$ equals its required mass $b_j$ ($\mathbf{T}^\top \mathbf{1}_n = \mathbf{b}$).

**Unbalanced Optimal Transport (UOT).** UOT relaxes strict mass conservation through KL-divergence regularization (Chizat et al., 2018b; Séjourné et al., 2019; Wang et al., 2025):

$$\mathcal{W}_{\text{UOT}}^{\kappa}(\mu_t, \mu_v) = \min_{\mathbf{T} \in \mathbb{R}_+^{n \times m}} \langle \mathbf{T}, \mathbf{C} \rangle + \kappa D_{\text{KL}}(\mathbf{T}\mathbf{1}_m \| \mathbf{a}) + \kappa D_{\text{KL}}(\mathbf{T}^\top \mathbf{1}_n \| \mathbf{b}), \tag{2}$$

where $\kappa > 0$ controls regularization. This enables *selective matching*: high-cost segments to remain unmatched rather than forced into poor alignments, which is crucial for handling non-stationarity in time series data. The UOT problem admits a dual formulation with optimal potentials $(\mathbf{f}^*, \mathbf{g}^*)$, where $\nabla_{\mathbf{a}} \mathcal{W}_{\text{UOT}}^{\kappa}(\mu_t, \mu_v) = \mathbf{f}^*$ and $\nabla_{\mathbf{b}} \mathcal{W}_{\text{UOT}}^{\kappa}(\mu_t, \mu_v) = \mathbf{g}^*$. These potentials encode marginal contributions of each sample, which will be utilized for data valuation.

### 3.2 PROBLEM FORMULATION AND WAVELET REPRESENTATION

**Notations and Problem Setup.** Let $\mathbf{X}_{\text{eval}} \in \mathbb{R}^{T_{\text{eval}} \times d}$ and $\mathbf{X}_{\text{ref}} \in \mathbb{R}^{T_{\text{ref}} \times d}$ denote the evaluated and reference time series, respectively, where $T$ represents the temporal length and $d$ is the feature dimensionality. To preserve local temporal dependencies, we partition each time series into overlapping segments via a sliding window of length $L$ with stride $S$, yielding $n$ evaluated segments $\mathcal{D}_{\text{eval}} = \{(\mathbf{x}_i, y_i)\}_{i=1}^{n}$ and $m$ reference segments $\mathcal{D}_{\text{ref}} = \{(\mathbf{x}_j', y_j')\}_{j=1}^{m}$, where each segment $\mathbf{x}_i, \mathbf{x}_j' \in \mathbb{R}^{L \times d}$ comprises $L$ consecutive time points. Labels $y_i, y_j' \in \mathcal{Y}$ encode task-specific information when available (e.g., activity type for supervised tasks). These segment sets induce empirical distributions $\mu_{\text{eval}}$ and $\mu_{\text{ref}}$ over the segment space $\mathcal{X}$, and joint distributions $\mu_e^{f_e}$ and $\mu_r^{f_r}$ over the product space $\mathcal{Z} = \mathcal{X} \times \mathcal{Y}$ when labels are utilized. Our objective is to assign a value $v(\mathbf{x}_i) \in \mathbb{R}$ to each *evaluated* segment that quantifies its *marginal* contribution to minimizing the distributional distance.

**Definition 1** (Discrete Wavelet Transform (DWT) (Heil & Walnut, 1989)). *Let* $\mathbf{x} = [x_0, x_1, ..., x_{L-1}]^\top \in \mathbb{R}^L$ *be a time series sequence. The DWT of* $\mathbf{x}$ *is defined by its coefficients:*

$$\Psi(\mathbf{x})_{j,k} = \sum_{l=0}^{L-1} x_l \psi_{j,k}[l], \quad j, k \in \mathbb{Z}, \tag{3}$$

*where* $\psi_{j,k}[l] = 2^{-j/2} \psi(2^{-j} l - k)$ *are the discrete wavelets generated by scaling (by $j$) and translating (by $k$) a mother wavelet function $\psi$.*

For multivariate time series, the DWT is typically applied independently to each feature dimension along the time axis. Based on this wavelet representation, we define the distance metric between time series segments as follows.

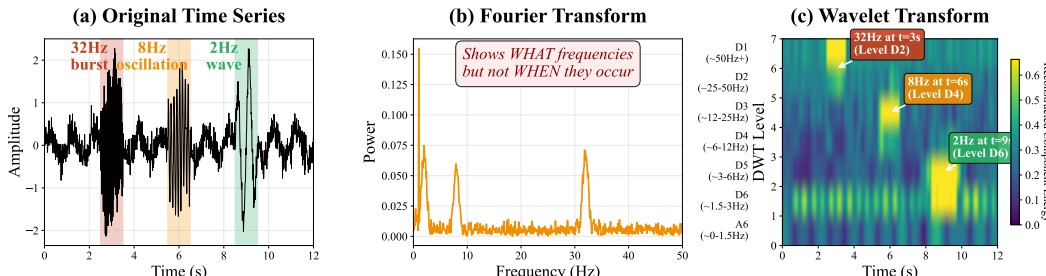

Figure 2: **Wavelet vs. Fourier Transform.** (a) a signal containing a low-frequency baseline and two localized high-frequency events ("Burst" at $t = 2$s and "Chirp" at $t = 6$s). (b) the Fourier transform captures the overall frequency content but loses temporal localization. (c) the wavelet transform preserves both time and frequency, enabling precise localization of transient event.

**Definition 2** (Wavelet Distance). *For two time series segments $\mathbf{x}_i, \mathbf{x}_j \in \mathbb{R}^L$, the wavelet distance is defined as:*

$$d_{\text{wav}}(\mathbf{x}_i, \mathbf{x}_j) = \|\Psi(\mathbf{x}_i) - \Psi(\mathbf{x}_j)\|_1, \tag{4}$$

*where $\Psi(\cdot)$ denotes the DWT operation and $\| \cdot \|_1$ is the $L_1$-norm.*

**Lemma 1** (Wavelet Distance Properties (Proof in Appendix C.1)). *The wavelet distance $d_{\text{wav}}(\mathbf{x}_i, \mathbf{x}_j)$ defined in Eq. 4 is a metric. It satisfies non-negativity, identity of indiscernibles, symmetry, and the triangle inequality.*

## 4 THE TIMELAVA FRAMEWORK

The TIMELAVA framework is built on a principled foundation established by LAVA (Just et al., 2023): *a data point's value is determined by its contribution to reducing the distributional distance between the evaluated distribution $\mu_{eval}$ and the reference distribution $\mu_{ref}$.* Applying this principle to time series presents unique challenges, as value is often encoded in local patterns and trends rather than isolated points. Our framework addresses this by representing data as a distribution of segments, where each segment constitutes a time-ordered sequence that preserves local temporal context. This segmentation-based approach shifts the central challenge to defining a metric that is sensitive to such structure. To this end, we introduce the Selective Wavelet-based Wasserstein ($\mathcal{W}_{\text{SW}}$) discrepancy at first. The valuation scheme, grounded in this segmentation strategy and our purpose-built metric, is formalized in Theorem 3. Figure 1 illustrates the complete workflow, which we detail in the following subsections.

### 4.1 CAPTURING TEMPORAL PATTERNS WITH WAVELETS

A central objective in time series data valuation is to identify the segments that contribute most positively or negatively to downstream task performance. These influential segments can manifest in multiple forms: persistent global trends that shape long-term behavior, localized patterns that capture short-term dynamics, or transient events like sensor spikes and financial shocks. Traditional methods like the Fourier transform, decompose signals into global frequency components, thereby discarding temporal locality (Bloomfield, 2004). In the context of data valuation, this loss of temporal resolution is problematic: knowing *what* frequencies exist without knowing *when* they occur conceals the events that often determine a sample's value.

The DWT addresses this limitation by providing a joint time–frequency representation by decomposing the signal into coefficients across multiple frequencies and times (Heil & Walnut, 1989; Rhif et al., 2019). This multi-resolution property simultaneously captures long-term trends and short-term fluctuations. As illustrated in Fig. 2, while Fourier analysis reveals overall frequency content, wavelets preserve both the timing and frequency characteristics of transient events. This localization property is particularly valuable for non-stationary time series where the distributional properties evolve over time, making wavelets a better choice for our distance metric design.

### 4.2 SELECTIVE WAVELET-BASED WASSERSTEIN DISTANCE

Time series data often exhibit *non-stationarity*, where statistical properties evolve, leading to the coexistence of different temporal patterns or "modes". For instance, industrial sensor streams shift

when production lines reconfigure, and traffic data patterns differ dramatically between rush hours and holidays. This variation, along with potential contamination by isolated outliers, makes it difficult to compute meaningful distributional discrepancies. The canonical Wasserstein distance (Eq. 1) is particularly ill-suited for this challenge. Its reliance on strict mass conservation forces every sample's mass to be matched, even across fundamentally different modes. This leads to erroneous pairings and an imprecise, inflated estimate of the discrepancy. This instability is not just empirical; it is formally captured by analyzing the impact of a single outlier mode. Prior work provides a lower bound showing that the standard Wasserstein distance increases unboundedly as an outlier deviates from the target distribution (Fatras et al., 2021).

To overcome this vulnerability, we adopt ideas from UOT, which relaxes the hard marginal constraints through a selective matching regularizer (Chizat et al., 2018b). As illustrated in Fig. 3, this allows the system to penalize and effectively ignore dissimilar points rather than forcing them into high-cost matches. This mechanism is powerful because it addresses both of the primary challenges found in time series data: the modal shifts that are a direct consequence of non-stationarity, and isolated outliers that can arise from sources like sensor noise or data corruption. Although their origins differ, both issues manifest computationally as high-cost mismatches that destabilize standard OT.

Building on this principle, we define the *Selective Wavelet-based Wasserstein* ($\mathcal{W}_{\text{SW}}$) discrepancy. It integrates the robustness of the UOT formulation with a wavelet-based ground metric introduced in §4.1 to capture the multi-scale temporal structures unique to time series data.

**Definition 3** (Selective Wavelet-based Wasserstein discrepancy). *Let $\mu_e^{f_e}$ and $\mu_r^{f_r}$ be the joint distributions over evaluated segments and labels $\{(\mathbf{x}_i^e, y_i^e)\}_{i=1}^n$ and reference segments and labels $\{(\mathbf{x}_j^r, y_j^r)\}_{j=1}^m$, respectively. The Selective Wavelet-based Wasserstein discrepancy is the UOT problem*

$$\mathcal{W}_{\text{SW}}(\mu_e^{f_e}, \mu_r^{f_r}) := \min_{\mathbf{T} \in \mathbb{R}_+^{n \times m}} \langle \mathbf{T}, \mathbf{D}^{(W)} \rangle + \kappa D_{\text{KL}}(\mathbf{T}\mathbf{1}_m \,\|\, \boldsymbol{\Delta}_n) + \kappa D_{\text{KL}}(\mathbf{T}^\top \mathbf{1}_n \,\|\, \boldsymbol{\Delta}_m), \quad (5)$$

*where $\mathbf{T}$ is the transport plan, $\kappa > 0$ controls the relaxation of marginal constraints via KL regularization to uniform targets $\boldsymbol{\Delta}_n = \mathbf{1}_n/n$ and $\boldsymbol{\Delta}_m = \mathbf{1}_m/m$, and $\mathbf{D}^{(W)}$ is the cost matrix*

$$\mathbf{D}^{(W)}\left((\mathbf{x}_i^e, y_i^e), (\mathbf{x}_j^r, y_j^r)\right) = d_{\text{wav}}(\mathbf{x}_i^e, \mathbf{x}_j^r) + c\, \mathcal{W}_{d_{\text{wav}}}\left(\mu_t(\cdot \mid y_i^e),\, \mu_v(\cdot \mid y_j^r)\right), \quad (6)$$

*where $d_{\text{wav}}$ is the wavelet distance (Eq. 4), $\mu_t(\cdot \mid y_i^e)$ is the conditional distribution of evaluated segments given label $y_i^e$, $\mathcal{W}_{d_{\text{wav}}}$ is the Wasserstein distance (Eq. 1) using $d_{\text{wav}}$ as the ground metric, and $c \geq 0$ is a weight balancing feature and label dissimilarity.*

Having established that non-stationarity induces mismatched modes that destabilize standard OT, we now formalize the robustness of our proposed $\mathcal{W}_{\text{SW}}$ discrepancy. As shown in Lemma 5, the instability of standard OT is most evident in its unbounded response to a single outlier, the simplest form of modal mismatch. We therefore analyze $\mathcal{W}_{\text{SW}}$ under the same setting to enable direct comparison. The following theorem demonstrates that, in precisely the case where standard OT fails, our method maintains stability with a bounded response, providing a crucial guarantee of resilience to the broader distributional shifts induced by non-stationarity.

**Theorem 2** (Robustness to Outlier Contamination (Proof in Appendix C.2)). *Consider a evaluated distribution $\mu_e^f$ contaminated by an outlier mode $(\mathbf{z}, y_z)$, resulting in $\tilde{\mu}_t^{f_t} = \zeta\delta_{(\mathbf{z},y_z)} + (1 - \zeta)\mu_e^{f_e}$ with relative mass $\zeta \in (0, 1)$. The $\mathcal{W}_{\text{SW}}$ discrepancy (Def. 3) between the contaminated distribution and the reference distribution $\mu_v^{f_v}$ is bounded:*

$$\mathcal{W}_{\text{SW}}(\tilde{\mu}_e^{f_e}, \mu_r^{f_r}) \leq (1 - \zeta)\mathcal{W}_{\text{SW}}(\mu_e^{f_e}, \mu_r^{f_r}) + 2\zeta\kappa\left(1 - e^{-\bar{d}(\mathbf{z},y_z)/(2\kappa)}\right), \quad (7)$$

*where $\bar{d}(\mathbf{z}, y_z)$ is the average cost $\mathbf{D}^{(W)}$ of transporting the outlier mode $(\mathbf{z}, y_z)$ to samples in $\mu_v^{f_v}$.*

This theorem mathematically formalizes the robustness conferred by the UOT component. Although the impact of an outlier scales with its mass $\zeta$, its influence is bounded. Even if an outlier is infinitely far away ($\bar{d} \to \infty$), its marginal contribution to the discrepancy is bounded by $2\zeta\kappa$. This provides a crucial distinction from standard OT, where the discrepancy would be inflated without bound. While this property is proven for a single outlier, it serves as the fundamental building block for understanding the method's broader stability. Since non-stationarity manifests as a collection of points forming a new, mismatched mode, the UOT mechanism addresses each point's mass individually. As

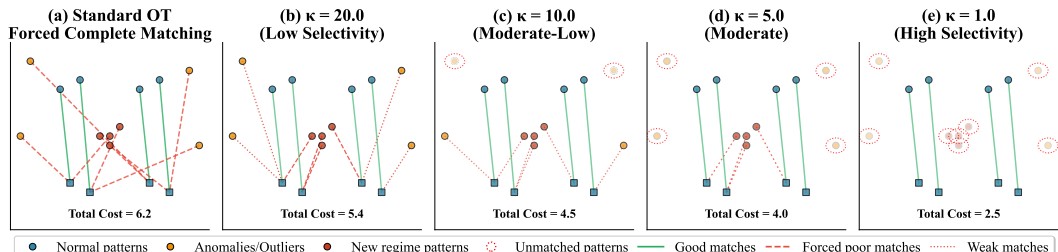

Figure 3: **UOT vs. OT** (a) Standard OT forces a complete matching, creating costly, poor matches for outliers and different data regimes. (b-e) The parameter $\kappa$ tunes this selectivity: as $\kappa$ decreases, the penalty for being unmatched decreases, allowing UOT to become more selective in its matching. UOT provides robustness by selectively matching similar points while ignoring dissimilar ones.

a result, the stability shown for a single point naturally extends to a group, and the total influence of an entire new regime is also bounded and proportional to its total mass.

This bounded-influence property is what makes the resulting data valuation from TIMELAVA both stable and highly informative. Crucially, the data value of a segment is a *relative* measure, reflecting how much it stands out from the other samples in the dataset. An isolated anomaly, being a unique and severe mismatch, creates a stark contrast against the predominantly normal data, resulting in an extremely low (i.e., harmful) data value score that clearly flags it for scrutiny. Conversely, a segment within a new operational regime is also a mismatch, but it is one among many similar peers that collectively alter the dataset's average characteristics. Therefore, while such a segment still receives a low score, it is not as extreme as that of the lone outlier. This is the key practical advantage: the *extremity of the data value score* becomes a powerful signal. It allows TIMELAVA to do more than just label data as 'good' or 'bad'; it provides a nuanced basis to distinguish between isolated, genuine anomalies and large-scale, systemic regime shifts, as demonstrated in Appendix C.2.

### 4.3 DATA VALUATION VIA SENSITIVITY ANALYSIS

Having introduced the $\mathcal{W}_{\text{SW}}$ discrepancy between time series distributions, it is essential to establish its relevance to performance. This connection is a prerequisite for building a theoretically grounded data valuation framework. We achieve this by proving a learning-agnostic generalization bound in which $\mathcal{W}_{\text{SW}}$ directly controls the gap between training and validation loss for a broad class of predictors.

We begin with a notion of *probabilistic cross-Lipschitzness* in the wavelet domain, which quantifies the consistency between labeling functions across domains.

**Definition 4** (Probabilistic Cross-Lipschitzness in the Wavelet Domain). *Two labeling functions* $f_t : \mathcal{X} \to \mathcal{Y}$ *and* $f_v : \mathcal{X} \to \mathcal{Y}$ *are* $(\epsilon_{tv}, \delta_{\text{wav}})$-*probabilistic cross-Lipschitz with respect to a joint distribution* $\pi$ *over* $\mathcal{X} \times \mathcal{X}$ *in the wavelet domain if:*

$$\mathbb{P}_{(\mathbf{x}_t, \mathbf{x}_v) \sim \pi} \left[ \left| f_t(\mathbf{x}_t) - f_v(\mathbf{x}_v) \right| > \epsilon_{tv} \cdot d_{\text{wav}}(\mathbf{x}_t, \mathbf{x}_v) \right] \leq \delta_{\text{wav}}, \tag{8}$$

*where* $d_{\text{wav}}$ *is the wavelet distance defined in Eq. 4,* $\epsilon_{tv} > 0$ *is the cross-Lipschitzness constant, and* $\delta_{\text{wav}} \in [0, 1]$ *is the violation probability bound.*

**Remark 1.** *In the context of UOT, the coupling* $\pi^*$ *over* $(\mathcal{X} \times \mathcal{Y}) \times (\mathcal{X} \times \mathcal{Y})$ *corresponds to transport matrix* $\mathbf{T}^*$. *The induced marginal distribution on* $\mathcal{X} \times \mathcal{X}$ *is used to verify the cross-Lipschitz condition.*

Intuitively, this condition requires that the two labeling functions assign similar labels to time series segments that are close in the wavelet domain with high probability. We then establish theoretical properties justifying the use of $\mathcal{W}_{\text{SW}}$ as a learning-agnostic measure of dataset utility for time series. These results connect the distributional discrepancy to generalization performance.

**Theorem 3** (Performance Bound (Proof in Appendix C.3)). *Let* $\mu_t^{f_t}$ *and* $\mu_v^{f_v}$ *be the training and reference distributions. Let* $f : \mathcal{X} \to [0, 1]$ *be any predictive model with loss* $\mathcal{L} : \mathcal{Y} \times [0, 1] \to \mathbb{R}_+$. *Assume: (i)* $\mathcal{L}$ *is* $k$-*Lipschitz in both arguments; (ii)* $f$ *is* $\epsilon$-*Lipschitz w.r.t.* $d_{\text{wav}}$; *(iii)* $|f(\mathbf{x})|, |y| \leq M$ *for all* $\mathbf{x}, y$; *(iv)* $f_t$ *and* $f_v$ *satisfy Def. 4 w.r.t. the marginal of* $\pi^*$ *from* $\mathcal{W}_{\text{SW}}$. *Then, the expected reference loss is bounded by:*

$$\mathbb{E}_{\mu_v^{f_v}} \left[ \mathcal{L}\big(y, f(\mathbf{x})\big) \right] \leq \mathbb{E}_{\mu_t^{f_t}} \left[ \mathcal{L}\big(y, f(\mathbf{x})\big) \right] + k\epsilon \, \mathcal{W}_{\text{SW}}(\mu_t^{f_t}, \mu_v^{f_v}) + 2kM \, \delta_{\text{wav}}.$$

Theorem 3 establishes $\mathcal{W}_{SW}$ as a learning-agnostic measure of dataset similarity relevant to generalization. It shows that a smaller $\mathcal{W}_{SW}$ discrepancy between the training and reference distributions implies a potentially smaller gap between training and reference performance for *any* well-behaved model $f$. This justifies using $\mathcal{W}_{SW}$ to assess the intrinsic quality or representativeness of the evaluated dataset relative to the reference target, specifically accounting for time series properties through the wavelet-based cost function.

## 5 EFFICIENT TIME SERIES DATA VALUATION

Building on Theorem 3, which links reference and training performance, and on $\mathcal{W}_{SW}$ (Def. 3) as a measure of distributional alignment for time series, we propose an efficient framework for valuing individual temporal segments via the dual formulation of UOT. The key idea is that the optimal dual variables of the UOT problem capture each segment's marginal contribution to distributional alignment, enabling direct value extraction without repeated dataset perturbations.

### 5.1 SEGMENT-WISE VALUATION

**Theorem 4** (Time Series Segment Valuation (Proof in Appendix C.4))**.** *Let* $(\mathbf{f}^*, \mathbf{g}^*)$ *be the optimal dual variables obtained from solving the* $\mathcal{W}_{SW}$ *problem in Eq. 5. Define* $\psi_\kappa(u) = \kappa(1 - e^{-u/\kappa})$*. The data value of the evaluated segment* $(\mathbf{x}_i, y_i)$ *can be derived as*

$$v(\mathbf{x}_i) = -\left( \psi_\kappa(\mathbf{f}_i^*) - \frac{1}{n-1} \sum_{j \neq i} \psi_\kappa(\mathbf{f}_j^*) \right).$$

This value measures the sensitivity of the $\mathcal{W}_{SW}$ discrepancy to perturbations in segment $i$'s mass. A positive value ($v(\mathbf{x}_i) > 0$) indicates that segment $i$ makes an above-average contribution to the discrepancy and helps align the evaluated and reference distributions. Conversely, a negative value ($v(\mathbf{x}_i) < 0$) suggests that the segment is potentially harmful for alignment. From a computational perspective, while Theorem 4 provides the formal definition of data value, solving the UOT problem directly can be computationally expensive. To enable a practical and efficient algorithm, we leverage entropy regularization (Chizat et al., 2018a; Séjourné et al., 2019; Pham et al., 2020), a technique originally developed for balanced OT (Cuturi, 2013) and extended to the unbalanced setting. This involves adding an entropy term $-\varepsilon H(\mathbf{T})$ to the UOT objective, where $\varepsilon > 0$ is the regularization strength. The validity of this approximation is formally established in Theorem 6 (see Appendix C.5 for the full statement and proof), which shows that as the regularization strength $\varepsilon \to 0$, the approximate value $v_\varepsilon(\mathbf{x}_i)$ converges to the true value $v(\mathbf{x}_i)$. Crucially, this result also guarantees that the relative ranking between segments is preserved for a sufficiently small $\varepsilon$. This convergence property justifies using efficient entropy-regularized UOT solvers to compute approximate data values, providing a reliable estimate of the relative importance of data points.

### 5.2 POINT-WISE VALUATION

For many applications, it is more practical to assign values to individual time points rather than to entire segments. Since our sliding-window approach generates overlapping segments, a single time point appears in multiple temporal contexts. Its value should therefore capture its aggregated influence across these contexts. To this end, we define the value of a time point $t$, denoted $v(t)$, as the weighted average of the values of all segments that include it:

$$v(t) = \frac{\sum_{\mathbf{z}^{[m]} \in \mathcal{S}_t} w(t, \mathbf{z}^{[m]}) \cdot v_\varepsilon(\mathbf{z}^{[m]})}{\sum_{\mathbf{z}^{[m]} \in \mathcal{S}_t} w(t, \mathbf{z}^{[m]})}, \tag{9}$$

where $\mathcal{S}_t$ is the set of segments containing time point $t$, and $v_\varepsilon(\mathbf{z}^{[m]})$ is the segment's value computed via Theorem 4. The weight $w(t, \mathbf{z}^{[m]})$ allows flexible emphasis, for example giving higher importance to central positions, though we adopt a uniform choice $w = 1$ for efficiency and robustness. This aggregation method provides a context-aware estimation of each point's contribution. This completes our TIMELAVA framework for learning-agnostic time series data valuation. The algorithms for segment valuation and point-wise valuation, and further discussion on the role of the reference time series, are provided in Appendix D.1.

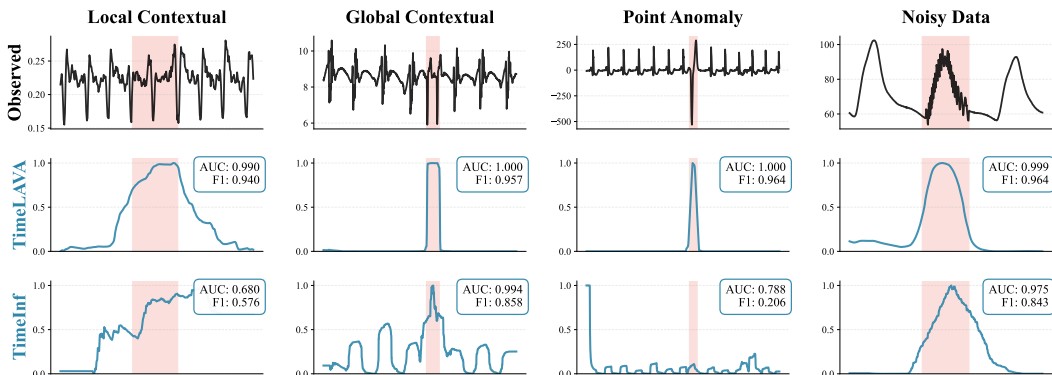

Figure 4: **Qualitative analysis on the UCR Time Series Anomaly Archive.** Each column corresponds to one anomaly type: *Local Contextual*, *Global Contextual*, *Point Anomaly*, and *Noisy Data*. The first row shows the observed time series with ground-truth anomalous regions shaded in red. Subsequent rows plot the normalized anomaly scores ($[0, 1]$) produced by each method. Boxes on the right report AUC/F1 scores (higher is better); higher curves in red regions with flat responses elsewhere indicate better localization and fewer false alarms.

## 6 USE CASES OF TIMELAVA

We conduct a comprehensive empirical study to evaluate TIMELAVA across diverse time series data valuation scenarios. Our experiments span multiple real-world datasets and tasks, assessing point-wise valuation in anomaly detection, and segment-wise valuation in both data selection/pruning and temporal label noise detection, thereby covering both predictive and diagnostic use cases.

### 6.1 ANOMALY DETECTION

In many time series applications, parts of the training data may contain anomalies from sensor faults, rare events, or unexpected operational states that deviate from normal temporal dynamics (Xu et al., 2022). This task aims to evaluate whether TIMELAVA's value estimates can distinguish anomalous from normal segments without prior location information, enabling robust anomaly identification.

**Experimental Setup.** Following (Jiang et al., 2022; Schmidl et al., 2022; Zhang et al., 2025), all methods compute anomaly scores directly on the potentially contaminated data with unknown anomaly proportion. A clean validation time series is used only as a reference for computing data values. We set $c = 0$ in Eq. 6 since we focus on unsupervised anomaly detection without labels.

**Datasets.** We evaluate TIMELAVA on several widely used benchmark datasets covering both univariate: UCR (Wu & Keogh, 2021), NAB (Ahmad et al., 2017) and multivariate time series: SMD (Su et al., 2019), SMAP and MSL (Hundman et al., 2018), PSM (Abdulaal et al., 2021), SWaT (Mathur & Tippenhauer, 2016a), WADI (Ahmed et al., 2017), and KDD-CUP99 (Stolfo et al., 2000). These datasets span diverse application domains and exhibit varying sampling frequencies, sequence lengths, and anomaly characteristics, providing a comprehensive basis for evaluation.

**Baselines.** We compare TIMELAVA against two categories of baselines: (1) time series data valuation methods: LWCV (Ghosh et al., 2020), and the recent state-of-the-art TimeInf (Zhang et al., 2025), and (2) anomaly detection algorithms: Isolation Forest (Liu et al., 2008), and an LSTM-based detector (Hundman et al., 2018), recent deep learning approaches (TranAD (Tuli et al., 2022), DCdetector (Yang et al., 2023), Anomaly Transformer (Xu et al., 2022), ModernTCN (donghao & wang xue, 2024) and CATCH (Wu et al., 2025)).

**Results.** We qualitatively evaluate on the UCR Time Series Anomaly Archive (Wu & Keogh, 2021), examining how different methods assign values to normal and anomalous across four representative anomaly types (Fig. 4). TIMELAVA produces discriminative scores that robustly identify diverse anomalies with high values, while keeping scores in normal regions low to prevent false alarms. Quantitative evaluations on multivariate benchmark datasets (Table 1) align with these qualitative findings. TIMELAVA attains the highest or near-highest AUC and F1 scores across all datasets, while preserving competitive computational efficiency. Its consistent F1 gains highlight improved precision in anomaly detection without sacrificing recall, offering tangible operational benefits. These results hold against both established anomaly detection algorithms and alternative data valuation methods,

Table 1: **Quantitative evaluation of anomaly detection performance and runtime.** Higher AUC and F1 scores indicate better detection accuracy. Across four real-world datasets, TIMELAVA achieves superior or competitive accuracy over established baselines, while maintaining low computational cost.

| | SMD | | | SMAP | | | MSL | | | PSM | | |
|---|---|---|---|---|---|---|---|---|---|---|---|---|
| | AUC | F1 | Time (s) | AUC | F1 | Time (s) | AUC | F1 | Time (s) | AUC | F1 | Time (s) |
| Isolation Forest | 0.82 | 0.37 | 0.30 | 0.67 | 0.33 | 0.14 | 0.68 | 0.31 | 0.11 | 0.71 | 0.51 | 1.12 |
| LSTM | 0.79 | 0.30 | 307.08 | 0.55 | 0.31 | 245.52 | 0.70 | 0.34 | 8.77 | 0.62 | 0.38 | 817.13 |
| ARIMA / VAR | 0.71 | 0.16 | 6.38 | 0.58 | 0.26 | 20.43 | 0.62 | 0.24 | 6.85 | 0.55 | 0.33 | 41.64 |
| LWCV | 0.79 | 0.29 | 3369.52 | 0.63 | 0.26 | 354.30 | 0.65 | 0.32 | 109.84 | 0.56 | 0.34 | 1852.07 |
| TranAD | 0.65 | 0.05 | 3.54 | 0.55 | 0.21 | 0.84 | 0.40 | 0.06 | 0.24 | 0.65 | 0.42 | 17.66 |
| DCdetector | 0.70 | 0.25 | 106.70 | 0.67 | 0.35 | 194.52 | 0.69 | 0.34 | 50.13 | 0.49 | 0.27 | 4075.59 |
| Anomaly Transformer | 0.51 | 0.05 | 246.80 | 0.49 | 0.04 | 51.24 | 0.42 | 0.10 | 20.13 | 0.48 | 0.15 | 2823.42 |
| ModernTCN | 0.72 | 0.15 | 225.66 | 0.46 | 0.12 | 650.70 | 0.63 | 0.12 | 81.56 | 0.59 | 0.09 | 4257.66 |
| CATCH | 0.81 | 0.24 | 3332.20 | 0.50 | 0.06 | 1314.04 | 0.66 | 0.14 | 677.21 | 0.65 | 0.12 | 1367.47 |
| TimeInf | 0.87 | 0.25 | 20.85 | 0.73 | 0.34 | 10.59 | 0.72 | 0.35 | 6.17 | 0.63 | 0.02 | 350.10 |
| **TIMELAVA (Ours)** | **0.91** | **0.52** | 14.01 | **0.74** | **0.54** | 0.72 | **0.81** | **0.49** | 0.25 | **0.77** | **0.58** | 8.16 |

underscoring the effectiveness of a learning-agnostic approach for time series anomaly identification. Additional experimental details and results across all datasets, along with parameter sensitivity analysis for segment length $L$ and stride $S$ and UOT regularizer $\kappa$, are provided in Appendix D.2.

## 6.2 DATA PRUNING AND SELECTION

A primary objective of data valuation is to identify high-quality segments that improve model performance while detecting corrupted or low-value segments that degrade it. This section evaluates TIMELAVA's ability to produce meaningful value rankings for time series segments.

**Experimental Setup.** We evaluate data valuation methods through two complementary experiments: (1) Data Selection: retaining only the top-k% highest-valued segments for training, and (2) Data Pruning: progressively removing the lowest-valued segments. To simulate realistic data quality issues, we inject synthetic noise (Gaussian, spike, drift, and scale perturbations) into 20% of randomly selected training segments. A linear AR model is trained on the selected segments and evaluate forecasting performance (RMSE, $R^2$) on a held-out test set.

**Datasets.** We evaluate on four benchmark time series datasets commonly used in forecasting literature: ETTh1 (Zhou et al., 2021), Traffic, Exchange, and Electricity (Lai et al., 2018). Each dataset is segmented into fixed-length, non-overlapping segments and divided into training, validation, and test sets, with the validation set serving as the reference time series.

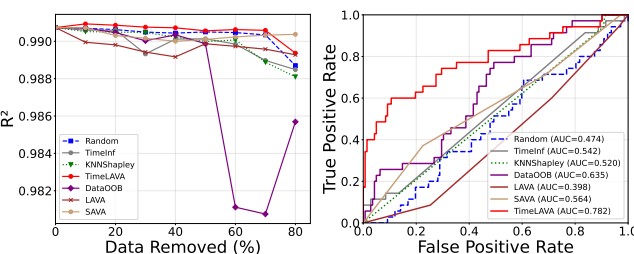

Figure 5: Data pruning performance ($R^2$) and noise detection (ROC) on Exchange dataset with 20% corrupted segments.

**Baselines.** We evaluate TIMELAVA against several representative data valuation methods: TimeInf (Zhang et al., 2025) adapts influence function to time series by estimating the effect of removing a segment on test loss. KNN-Shapley (Jia et al., 2019a) approximates Shapley values via $k$NN-based marginal utility estimation. Data-OOB (Kwon & Zou, 2023) assigns values using out-of-bag predictions from a bagged decision-tree ensemble. LAVA (Just et al., 2023) measures training–validation alignment via the Wasserstein distance, and SAVA (Kessler et al., 2025) provides a scalable batch-based variant. A Random baseline is also included for reference.

**Results.** Fig. 5 demonstrates TIMELAVA's effective performance in both data quality assessment and noise detection on the Exchange dataset. In the data pruning experiment, TIMELAVA maintains the highest $R^2$ as low-value segments are removed, with performance degrading only after removing 45% of the data, indicating accurate identification of detrimental segments. Other methods show earlier performance degradation, with Data-OOB experiencing a sharp drop after 50% removal. The ROC curve show TIMELAVA achieves the highest AUC for detecting corrupted segments, substantially outperforming all baselines including TimeInf and KNN-Shapley. Additional experimental details and results across all datasets are provided in Appendix D.3.

### 6.3 Noisy label detection

In real-world detection systems, label quality often varies over time due to operational conditions, annotator fatigue, or system degradation (Carpenter, 2003; Hsieh & Kocielnik, 2016; Nagaraj et al., 2025). Traditional approaches assume static noise distributions, fundamentally misspecifying the noise model and leading to suboptimal performance. This task evaluates whether TimeLAVA's temporal modeling can effectively identify and handle time-varying label corruption patterns.

**Experimental Setup.** Following established practices for label noise evaluation (Jiang et al., 2023; Just et al., 2023), we adopt a semi-synthetic framework that treats original dataset labels as ground truth and systematically injects temporal noise patterns. Labels are corrupted by flipping to the opposite class at time segments selected according to four distinct temporal noise patterns: Random, Periodic, Decay, and Growth. Datasets are split temporally into 70% training and 30% validation (reference) sets, with noise injected only into training labels to simulate realistic deployment scenarios. We set $c = 1$ in Eq. 6 to explicitly incorporate label information for detecting mislabeled samples. We compare against the same baseline methods used in the pruning experiments §6.2, ensuring consistent evaluation across different aspects of our work.

**Datasets.** We evaluate on three healthcare classification datasets: Moving (Reyes-Ortiz et al., 2013), Senior (Logacjov & Ustad, 2023), and Blinking (Roesler, 2013). These tasks involve sequential labeling where temporal corruption patterns can realistically occur in practical applications.

**Results.** Fig. 6 shows results on the Moving dataset. For periodic noise, TimeLAVA consistently outperforms other methods across all noise rates, showing robust performance in realistic low-corruption scenarios. This superiority stems from TimeLAVA's ability to model temporal dependencies, enabling it to identify systematic patterns in label corruption that occur in real-world scenarios such as periodic sensor failures or shift-based annotation quality variations. For random noise, TimeLAVA performs comparably to KNNShapley, which is expected when corruption lacks temporal structure. This validates that our method exploits temporal patterns when present without overfitting when absent. Results for all datasets and noise patterns are provided in Appendix D.4.

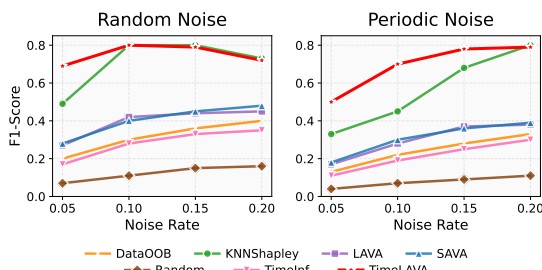

Figure 6: **F1 scores**: random vs. periodic label noise on Moving dataset.

## 7 Conclusion and Limitations

This paper introduced TimeLAVA, a learning-agnostic framework for valuing individual time points and temporal segments in time series data. By combining multi-scale wavelet transforms with selective matching via unbalanced optimal transport, the proposed $\mathcal{W}_{SW}$ discrepancy provides robust representations that capture localized patterns while remaining resilient to noise and non-stationarity. Segment values are derived through a sensitivity analysis of $\mathcal{W}_{SW}$, with theoretical guarantees providing its links to model-agnostic generalization bounds. Our experiments demonstrate that TimeLAVA effectively identifies both highly valuable and detrimental segments, enabling improved model training efficiency. However, its performance relies on the representativeness of the reference set; biased or incomplete references may cause rare yet important patterns to be undervalued (Jahagirdar et al., 2024). Future work should therefore explore valuation methods robust to imperfect and diverse reference sets, ensuring broader applicability in real-world scenarios.

## Ethics Statement

This research on time series data valuation raises no significant ethical concerns. Our work does not involve human subjects, sensitive data collection, or applications with direct societal harm potential. The datasets used in our experiments are publicly available benchmarks from established repositories. All datasets are used in accordance with their respective licenses, and our methodology does not

introduce additional discrimination or fairness issues beyond the properties of the original datasets. While our method could be applied to identify and remove data from models, we emphasize its primary purpose is to improve model performance and data quality assessment. We acknowledge that data valuation techniques could potentially be misused for data manipulation or to undervalue certain data sources, and we encourage responsible application of these methods with appropriate oversight. The computational resources used for this research were standard academic computing facilities with no special environmental concerns beyond typical deep learning experiments.

## REPRODUCIBILITY STATEMENT

To ensure the reproducibility of our work, we provide comprehensive implementation details and experimental protocols throughout the paper and appendices. All algorithms are fully specified: Algorithm 1 details the TimeLAVA segment-wise valuation procedure, Algorithm 2 describes point-wise contribution calculation, and the $\mathcal{W}_{\mathrm{SW}}$ discrepancy computation is formally defined in Def. 3. All hyperparameters are specified in Appendix D, including segment length $L$, stride $S$, wavelet parameters (wavelet type and decomposition levels), and UOT regularization coefficients $\kappa$ and $\varepsilon$. All datasets used are publicly available with details provided in corresponding experimental sections: UCR, NAB, SMAP/MSL, SMD, PSM, SWaT, WADI, and KDD-Cup99 for anomaly detection (Section 6.1 and Appendix D.2); ETTh1, Electricity, Traffic, and Exchange for data selection and pruning (Section 6.2 and Appendix D.3); Moving, Senior, and Blinking for temporal label noise detection (Section 6.3 and Appendix D.4). All data preprocessing steps, including segmentation procedures and noise injection protocols for semi-synthetic experiments, are fully detailed in the respective experimental sections where applicable. The mathematical foundations, including all proofs for Theorems 2-6 and Lemma 1 are provided in Appendix C. The computational requirements and approximate runtimes are documented in Appendix D.1.3. To support transparency and broader use, we will release code and evaluation scripts upon publication, enabling full reproducibility of the reported results.

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

# Appendix

## Table of Contents

## A ACKNOWLEDGMENT OF LLM USAGE

During the preparation of this manuscript, large language models (LLMs) were used exclusively for language editing purposes, including correcting typographical errors, improving grammar, and refining phrasing. LLMs were not used for generating research ideas, performing analyses, producing results, or interpreting findings. The authors take full responsibility for all scientific content presented in this work.

## B NOTATION SUMMARY

Table 2: Table of Notations

| Symbol | Description |
|---|---|
| **Time Series and Data Structures** | |
| $X_{\text{eval}} \in \mathbb{R}^{T_{\text{train}} \times d}$ | Evaluated time series |
| $X_{\text{ref}} \in \mathbb{R}^{T_{\text{ref}} \times d}$ | Reference (validation) time series |
| $x_i^t, x_j^e, x_k^v \in \mathbb{R}^L$ | Training evaluated and validation time series segments of length $L$ |
| $t$ | Time index |
| $T$ | Total length of time series |
| $L$ | Length of each segment (window size) |
| $S$ | Stride for sliding window |
| $d$ | Number of features in multivariate time series |
| $\mathcal{D}_{eval}$ | Evaluated segment-label pairs |
| $\mathcal{D}_{\text{ref}}$ | Reference segment-label pairs |
| $n, m$ | Number of segments in $\mathcal{D}_t$ and $\mathcal{D}_v$ |
| $\mathcal{X}$ | Input space of time series segments |
| $\mathcal{Y}$ | Output or label space |
| $\mathcal{Z}$ | Joint input-output space, $\mathcal{X} \times \mathcal{Y}$ |
| $\mathcal{S}_t$ | Set of segments containing time point $t$ |
| **Functions and Models** | |
| $f : \mathcal{X} \to [0, 1]$ | Predictive model |
| $f_t, f_v$ | Labeling functions for training and validation data |
| $\mathcal{L} : \mathcal{Y} \times [0, 1] \to \mathbb{R}_+$ | Loss function |
| **Distributions and Measures** | |
| $\mu_t, \mu_v$ | Empirical distributions over training and validation segments |
| $\mu_t^{f_t}, \mu_v^{f_v}$ | Joint distributions over segments and labels |
| $\mu_t(\cdot|y), \mu_v(\cdot|y)$ | Conditional distributions given label $y$ |
| $\delta_z$ | Dirac delta measure centered at $z$ |
| $\tilde{\mu}_t^{f_t}$ | Contaminated distribution |
| **Wavelet Transform** | |
| $\psi$ | Mother wavelet function |
| $\psi_{j,k}[t]$ | Discrete wavelet at scale $j$ and position $k$ |
| $\Psi(\cdot)$ | Discrete Wavelet Transform (DWT) operator |
| $\Psi(\mathbf{x})_{j,k}$ | DWT coefficient at scale $j$ and position $k$ |
| $d_{wav}(\mathbf{x}_i, \mathbf{x}_j)$ | Wavelet distance: $\|\Psi(\mathbf{x}_i) - \Psi(\mathbf{x}_j)\|_1$ |
| **Optimal Transport (OT)** | |
| $\mathbf{T} \in \mathbb{R}_+^{n \times m}$ | Transport plan (coupling matrix) |
| $\mathbf{C}$ | Cost matrix for standard OT |
| $\mathbf{D}^{(W)}$ | Joint feature-label cost matrix for $\mathcal{W}_{\text{SW}}$ distance |
| $\Pi(\mu_t, \mu_v)$ | Set of valid transport plans |
| $\mathcal{W}(\mu_t, \mu_v)$ | Standard Wasserstein distance |

| Symbol | Description |
|---|---|
| $\mathcal{W}_{d_{wav}}$ | Wasserstein distance with wavelet ground metric |
| $\mathcal{W}_{SW}$ | Selective Wavelet-based Wasserstein ($\mathcal{W}_{\text{SW}}$) discrepancy |

**Unbalanced OT and Regularization**

| | |
|---|---|
| $\kappa > 0$ | Regularization parameter for UOT marginal relaxation |
| $D_{KL}(\cdot\|\cdot)$ | Kullback-Leibler divergence |
| $\mathbf{\Delta}_n = \mathbf{1}_n/n$ | Uniform distribution over $n$ samples |
| $\mathbf{\Delta}_m = \mathbf{1}_m/m$ | Uniform distribution over $m$ samples |
| $\mathbf{f}^* \in \mathbb{R}^n, \mathbf{g}^* \in \mathbb{R}^m$ | Optimal dual potentials |
| $\mathbf{f}_\varepsilon^*, \mathbf{g}_\varepsilon^*$ | Optimal dual potentials with entropy regularization |
| $\psi_\kappa(u)$ | Transform function from KL penalty |
| $\varepsilon > 0$ | Entropy regularization strength |
| $H(\mathbf{T})$ | Entropy of transport plan |

**Theoretical Constants and Bounds**

| | |
|---|---|
| $k$ | Lipschitz constant of loss function $L$ |
| $\epsilon$ | Lipschitz constant of model $f$ w.r.t. $d_{wav}$ |
| $M$ | Upper bound on $|f(\mathbf{x})|$ and $|y|$ |
| $\epsilon_{tv}$ | Probabilistic cross-Lipschitz constant |
| $\delta_{wav} \in [0,1]$ | Cross-Lipschitz violation probability |
| $\pi^*$ | Optimal coupling for $\mathcal{W}_{\text{SW}}$ |
| $\zeta \in (0,1)$ | Relative mass of outlier mode |
| $\bar{d}(z, y_z)$ | Average cost of transporting outlier |

**Data Valuation**

| | |
|---|---|
| $v(\mathbf{x}_i)$ | Data value of segment $\mathbf{x}_i$ |
| $v_\varepsilon(\mathbf{x}_i)$ | Entropy-regularized data value |
| $v(t)$ | Point-wise data value at time $t$ |
| $w(t, \mathbf{z}^{[m]})$ | Weight for point-wise aggregation |

**Miscellaneous**

| | |
|---|---|
| $\mathbf{1}_k \in \mathbb{R}^k$ | Column vector of ones |
| $\langle \cdot, \cdot \rangle$ | Frobenius inner product |
| $\| \cdot \|_1$ | $L_1$ norm (sum of absolute values) |
| $\| \cdot \|_\infty$ | $L_\infty$ norm (maximum absolute value) |
| $c \geq 0$ | Hierarchical weight in $\mathbf{D}^{(W)}$ |
| $\mathcal{N}$ | Set of corrupted time points (noise injection) |
| $\eta$ | Noise rate |

## C    PROOFS

This section provides the detailed proofs for the lemmas and theorems presented in the main text.

### C.1    PROOF OF LEMMA 1: WAVELET DISTANCE PROPERTIES

We need to prove that the wavelet distance $d_{\text{wav}}(\mathbf{x}_i, \mathbf{x}_j) = \|\mathcal{W}(\mathbf{x}_i) - \mathcal{W}(\mathbf{x}_j)\|_1$ is a metric, satisfying the four metric properties: non-negativity, identity of indiscernibles, symmetry, and the triangle inequality.

*Proof.* Let $\mathbf{x}_i, \mathbf{x}_j, \mathbf{x}_k \in \mathbb{R}^T$ be time series segments (for simplicity, consider $d = 1$; the extension to $d > 1$ by applying DWT channel-wise and summing L1 norms maintains metric properties if the sum is used, or if applied to concatenated wavelet coefficients). Let $\mathcal{W}(\mathbf{x})$ denote the vector of wavelet coefficients for $\mathbf{x}$.

    1. **Non-negativity:** For any time series $\mathbf{x}_i, \mathbf{x}_j$,

$$d_{\text{wav}}(\mathbf{x}_i, \mathbf{x}_j) = \|\mathcal{W}(\mathbf{x}_i) - \mathcal{W}(\mathbf{x}_j)\|_1 \geq 0$$

This follows directly from the definition of the L1-norm, which is always non-negative.

2. **Identity of indiscernibles:** We need to show $d_{\text{wav}}(\mathbf{x}_i, \mathbf{x}_j) = 0 \iff \mathbf{x}_i = \mathbf{x}_j$.

- ($\Rightarrow$): If $d_{\text{wav}}(\mathbf{x}_i, \mathbf{x}_j) = 0$, then $\|\mathcal{W}(\mathbf{x}_i) - \mathcal{W}(\mathbf{x}_j)\|_1 = 0$. By properties of norms, this implies $\mathcal{W}(\mathbf{x}_i) - \mathcal{W}(\mathbf{x}_j) = \mathbf{0}$, so $\mathcal{W}(\mathbf{x}_i) = \mathcal{W}(\mathbf{x}_j)$. Since the DWT (as typically implemented with a complete basis) is invertible, we can apply the inverse DWT $\mathcal{W}^{-1}$ to both sides:
$$\mathcal{W}^{-1}(\mathcal{W}(\mathbf{x}_i)) = \mathcal{W}^{-1}(\mathcal{W}(\mathbf{x}_j)) \implies \mathbf{x}_i = \mathbf{x}_j$$
- ($\Leftarrow$): If $\mathbf{x}_i = \mathbf{x}_j$, then $\mathcal{W}(\mathbf{x}_i) = \mathcal{W}(\mathbf{x}_j)$ since the DWT is a deterministic linear transform. Therefore, $\|\mathcal{W}(\mathbf{x}_i) - \mathcal{W}(\mathbf{x}_j)\|_1 = \|\mathbf{0}\|_1 = 0$.

3. **Symmetry:** We need to show $d_{\text{wav}}(\mathbf{x}_i, \mathbf{x}_j) = d_{\text{wav}}(\mathbf{x}_j, \mathbf{x}_i)$.
$$\begin{aligned}
d_{\text{wav}}(\mathbf{x}_i, \mathbf{x}_j) &= \|\mathcal{W}(\mathbf{x}_i) - \mathcal{W}(\mathbf{x}_j)\|_1 \\
&= \| - (\mathcal{W}(\mathbf{x}_j) - \mathcal{W}(\mathbf{x}_i))\|_1 \\
&= \|\mathcal{W}(\mathbf{x}_j) - \mathcal{W}(\mathbf{x}_i)\|_1 \quad (\text{since } \| - \mathbf{v}\|_1 = \|\mathbf{v}\|_1) \\
&= d_{\text{wav}}(\mathbf{x}_j, \mathbf{x}_i)
\end{aligned}$$

Thus, symmetry holds.

4. **Triangle inequality:** We need to show $d_{\text{wav}}(\mathbf{x}_i, \mathbf{x}_k) \leq d_{\text{wav}}(\mathbf{x}_i, \mathbf{x}_j) + d_{\text{wav}}(\mathbf{x}_j, \mathbf{x}_k)$.
$$\begin{aligned}
d_{\text{wav}}(\mathbf{x}_i, \mathbf{x}_k) &= \|\mathcal{W}(\mathbf{x}_i) - \mathcal{W}(\mathbf{x}_k)\|_1 \\
&= \|\mathcal{W}(\mathbf{x}_i) - \mathcal{W}(\mathbf{x}_j) + \mathcal{W}(\mathbf{x}_j) - \mathcal{W}(\mathbf{x}_k)\|_1 \\
&\leq \|\mathcal{W}(\mathbf{x}_i) - \mathcal{W}(\mathbf{x}_j)\|_1 + \|\mathcal{W}(\mathbf{x}_j) - \mathcal{W}(\mathbf{x}_k)\|_1 \quad (\text{by triangle inequality of L1-norm}) \\
&= d_{\text{wav}}(\mathbf{x}_i, \mathbf{x}_j) + d_{\text{wav}}(\mathbf{x}_j, \mathbf{x}_k)
\end{aligned}$$

Therefore, the triangle inequality holds.

Having proved all four properties, we conclude that $d_{\text{wav}}(\mathbf{x}_i, \mathbf{x}_j)$ is a metric. $\qquad\square$

### C.2 PROOF OF THEOREM 2: ROBUSTNESS TO NON-STATIONARITY

**Lemma 5.** *Suppose that $\tilde{\alpha} = \zeta\delta_z + (1 - \zeta)\alpha$ is a distribution perturbed by a Dirac mode at $z$ with relative mass $\zeta \in (0, 1)$. For a sample $y^*$ in the support of $\beta$, Fatras et al. (2021) demonstrates:*

$$\mathcal{W}(\tilde{\alpha}, \beta) \geq (1 - \zeta)\mathcal{W}(\alpha, \beta) + \zeta\left(D(z, y^*) - g(y^*) + \int g \, \mathrm{d}\beta\right),$$

*where $D(z, y^*)$ is the deviation of $\delta_z$, and $g$ is the optimal dual potential of $\mathcal{W}(\alpha, \beta)$.*

*Proof.* This theorem builds upon the foundation of Lemma 5 by Fatras et al. (2021). To establish this robustness bound, we construct a parametric family of feasible transport plans and optimize over the parameter to obtain the tightest upper bound. Let $\mathbf{T}^*$ be the optimal transport plan for the clean problem $\tilde{\mathcal{W}}_{\text{SW}}(\mu_t^{f_t}, \mu_v^{f_v})$. We construct a parametric family of transport plans $\tilde{\mathbf{T}}_\phi$ with parameter $\phi \in [0, 1]$ as follows:

$$\tilde{\mathbf{T}}_\phi = (1 - \zeta)\mathbf{T}^* + \zeta\phi(\delta_{\mathbf{z}} \otimes \boldsymbol{\Delta}_m) = \begin{pmatrix} (1 - \zeta)\mathbf{T}^* \\ \frac{\zeta\phi}{m}\mathbf{1}_{1\times m} \end{pmatrix} \tag{10}$$

The parameter $\phi$ controls how the outlier mass is handled: when $\phi = 1$, all outlier mass $\zeta$ is transported to the validation set; when $\phi = 0$, the outlier mass is completely destroyed (incurring only mass penalty); and for $\phi \in (0, 1)$, we have partial transport with the remainder destroyed.

Since $\tilde{\mathbf{T}}_\phi$ is a feasible transport plan for the contaminated problem, we have the upper bound $\tilde{\mathcal{W}}_{\text{SW}}(\tilde{\mu}_t^{f_t}, \mu_v^{f_v}) \leq \text{Cost}(\tilde{\mathbf{T}}_\phi)$, where the cost function is

$$\text{Cost}(\tilde{\mathbf{T}}_\phi) = \langle \mathbf{D}^{(W)}, \tilde{\mathbf{T}}_\phi \rangle + \kappa D_{\text{KL}}(\tilde{\mathbf{T}}_\phi \mathbf{1}_m \| \tilde{\mu}_t^{f_t}) + \kappa D_{\text{KL}}(\tilde{\mathbf{T}}_\phi^\top \mathbf{1}_{n+1} \| \mu_v^{f_v}) \tag{11}$$

We now analyze each component of this cost function to express it in terms of the parameter $\phi$.

For the transport cost component, we have

$$\langle \mathbf{D}^{(W)}, \tilde{\mathbf{T}}_\phi \rangle = \langle \mathbf{D}^{(W)}, (1-\zeta)\mathbf{T}^* + \zeta\phi(\delta_{\mathbf{z}} \otimes \boldsymbol{\Delta}_m) \rangle \tag{12}$$

$$= (1-\zeta)\langle \mathbf{D}^{(W)}, \mathbf{T}^* \rangle + \zeta\phi \cdot \bar{d}(\mathbf{z}, y_z) \tag{13}$$

where $\bar{d}(\mathbf{z}, y_z) = \frac{1}{m}\sum_{j=1}^m \mathbf{D}^{(W)}((\mathbf{z}, y_z), (\mathbf{x}_j^v, y_j^v))$ is the average cost of transporting the outlier to the validation set.

For the first KL divergence term on the training side, the marginal distribution is $\tilde{\mathbf{T}}_\phi \mathbf{1}_m = (1 - \zeta)\mathbf{T}^* \mathbf{1}_m + \zeta\phi\delta_{\mathbf{z}}$. Using the joint convexity of the KL divergence, we obtain

$$D_{\mathrm{KL}}(\tilde{\mathbf{T}}_\phi \mathbf{1}_m \| \tilde{\mu}_t^{f_t}) \le (1-\zeta)D_{\mathrm{KL}}(\mathbf{T}^* \mathbf{1}_m \| \mu_t^{f_t}) + \zeta D_{\mathrm{KL}}(\phi\delta_{\mathbf{z}} \| \delta_{\mathbf{z}}) \tag{14}$$

where the key calculation shows that $D_{\mathrm{KL}}(\phi\delta_{\mathbf{z}} \| \delta_{\mathbf{z}}) = \phi \log \phi - \phi + 1$ for the generalized KL divergence.

Similarly, for the second KL divergence term on the validation side, the marginal is $\tilde{\mathbf{T}}_\phi^\top \mathbf{1}_{n+1} = (1 - \zeta)(\mathbf{T}^*)^\top \mathbf{1}_n + \zeta\phi\boldsymbol{\Delta}_m$, which gives us

$$D_{\mathrm{KL}}(\tilde{\mathbf{T}}_\phi^\top \mathbf{1}_{n+1} \| \mu_v^{f_v}) \le (1-\zeta)D_{\mathrm{KL}}((\mathbf{T}^*)^\top \mathbf{1}_n \| \mu_v^{f_v}) + \zeta D_{\mathrm{KL}}(\phi\boldsymbol{\Delta}_m \| \boldsymbol{\Delta}_m) \tag{15}$$

where $D_{\mathrm{KL}}(\phi\boldsymbol{\Delta}_m \| \boldsymbol{\Delta}_m) = \phi \log \phi - \phi + 1$ by the same calculation.

Combining all cost components, we obtain

$$\mathrm{Cost}(\tilde{\mathbf{T}}_\phi) \le (1-\zeta)\tilde{\mathcal{W}}_{\mathrm{SW}}(\mu_t^{f_t}, \mu_v^{f_v}) + \zeta E(\phi) \tag{16}$$

where the excess cost function is

$$E(\phi) = \phi \cdot \bar{d}(\mathbf{z}, y_z) + 2\kappa(\phi \log \phi - \phi + 1) \tag{17}$$

To find the tightest upper bound, we minimize $E(\phi)$ over $\phi \in [0, 1]$ by taking the derivative:

$$\frac{dE}{d\phi} = \bar{d}(\mathbf{z}, y_z) + 2\kappa \log \phi \tag{18}$$

Setting this equal to zero yields the optimal parameter $\phi^* = e^{-\bar{d}(\mathbf{z}, y_z)/(2\kappa)}$.

Substituting the optimal parameter $\phi^*$ into the excess cost function and simplifying, we get

$$E(\phi^*) = e^{-\bar{d}/(2\kappa)} \cdot \bar{d} + 2\kappa\left(e^{-\bar{d}/(2\kappa)} \cdot \left(-\frac{\bar{d}}{2\kappa}\right) - e^{-\bar{d}/(2\kappa)} + 1\right) \tag{19}$$

$$= 2\kappa(1 - e^{-\bar{d}(\mathbf{z}, y_z)/(2\kappa)}) \tag{20}$$

Therefore, we conclude that

$$\tilde{\mathcal{W}}_{\mathrm{SW}}(\tilde{\mu}_t^{f_t}, \mu_v^{f_v}) \le (1-\zeta)\tilde{\mathcal{W}}_{\mathrm{SW}}(\mu_t^{f_t}, \mu_v^{f_v}) + 2\zeta\kappa\left(1 - e^{-\bar{d}(\mathbf{z}, y_z)/(2\kappa)}\right) \tag{21}$$

This bound demonstrates the robustness of the $\mathcal{W}_{\mathrm{SW}}$ distance: the impact of the outlier is modulated by both its proportion $\zeta$ and its distance to the validation set, with the regularization parameter $\kappa$ controlling how aggressively distant outliers are penalized through the selective matching mechanism. $\square$

**Empirical Validation of Theorem 2.** To demonstrate that the distribution patterns of data value scores can distinguish isolated anomalies from systemic regime shifts, we designed a controlled synthetic experiment. This approach provides complete control over ground truth, enabling precise validation of the theoretical predictions from Theorem 2.

We generated a clean reference time series, $x(t) = \sin(2\pi t/100) + 0.5\sin(2\pi t/25) + \epsilon$, $\epsilon \sim \mathcal{N}(0, 0.01)$. We then constructed two test datasets. **Dataset A (Isolated Anomalies)** contains 50 sporadic spike anomalies (5% contamination) with magnitudes between 8 and 12 standard deviations at the time of injection, resulting in progressively larger absolute magnitudes as the contamination accumulates. This creates a realistic cascading effect often observed in deteriorating sensor systems where initial faults compound over time. **Dataset B (Regime Shift)** exhibits a systematic change at the midpoint, where the second half has an altered frequency and amplitude, simulating operational reconfiguration.

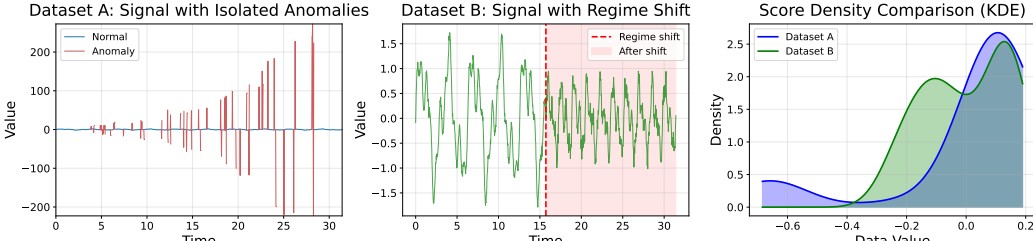

Figure 7: Experimental validation of Theorem 2: TIMELAVA distinguishes between isolated anomalies and regime shifts through characteristic score distributions. **Left**: Isolated anomalies manifest as sporadic spikes distributed across the time series (red markers indicate $50$ anomalies, $5\%$ of data). **Middle**: Regime shift exhibits systematic change after the transition point (red dashed line at $t = 500$), with the shaded region indicating the new regime. **Right**: Score density comparison reveals fundamentally different distributions: isolated anomalies produce a heavy-tailed distribution with extreme outliers (blue), while regime shifts yield a bimodal distribution (green) reflecting pre- and post-shift regimes. The JS divergence of $0.47$ confirms statistical distinguishability, demonstrating that the bounded response property of the $\mathcal{W}_{\mathrm{SW}}$ distance enables nuanced data quality assessment.

As shown in Figure 7, the induced score distributions are fundamentally different. Isolated anomalies yield a heavy-tailed unimodal distribution (kurtosis = 2.95, skewness = $-2.22$; minimum = $-0.67$), whereas the regime shift produces a bimodal distribution with two peaks corresponding to pre/post-shift regimes and no extreme values (bimodality score = 0.38; range = $[-0.23, 0.19]$). The Jensen–Shannon divergence of $0.47$ further confirms statistical distinguishability. This pattern is consistent with the bound in Eq. (21), which encodes a trade-off between anomaly severity $\bar{d}$ and prevalence $\zeta$: large $\bar{d}$ but small $\zeta$ pushes a few segments close to the bound, producing heavy-tailed distribution with extreme outliers, while moderate $\bar{d}$ with large $\zeta$ shifts many segments coherently, yielding bimodal distributions reflecting dual regimes.

A key insight from our experiments is that, as established by Theorem 2, the bounded-response property naturally induces distinguishable distribution patterns for different anomaly types, enabling actionable data curation without hand-crafted design. Consequently, a single scoring rule enables both detection and differentiation of anomaly types without hand-crafted detectors. Consistent real-world behavior is observed in Fig. 14, where TIMELAVA-OT shows sharp, isolated negative peaks for true anomalies in SMAP channels G-1 and A-2 while remaining stable elsewhere.

### C.3 PROOF OF THEOREM 3: VALIDATION PERFORMANCE BOUND

*Proof.* We adapt the proof structure from Just et al. (2023) for the wavelet distance $d_{\mathrm{wav}}$ and the $\mathcal{W}_{\mathrm{SW}}$ cost $\mathbf{D}^{(W)}$. Let $\Delta L$ denote the difference in expected loss between validation and training:

$$\Delta L = \mathbb{E}_{(\mathbf{x},y)\sim\mu_v^{f_v}}\Big[L\big(y, f(\mathbf{x})\big)\Big] - \mathbb{E}_{(\mathbf{x},y)\sim\mu_t^{f_t}}\Big[L\big(y, f(\mathbf{x})\big)\Big]. \tag{22}$$

Let $\mathcal{Z} = \mathcal{X} \times \mathcal{Y}$ be the joint space, and let $\pi^*$ be the optimal coupling that achieves the minimum in $\mathcal{W}_{HW}(\mu_t^{f_t}, \mu_v^{f_v})$. Since $\pi^*$ is a coupling between $\mu_t^{f_t}$ and $\mu_v^{f_v}$, we can express:

$$\Delta L = \int_{\mathcal{Z}^2} [L(y_v, f(\mathbf{x}_v)) - L(y_t, f(\mathbf{x}_t))] \, d\pi^*((\mathbf{x}_t, y_t), (\mathbf{x}_v, y_v))$$

$$= \int_{\mathcal{Z}^2} [L(y_v, f(\mathbf{x}_v)) - L(y_v, f(\mathbf{x}_t)) + L(y_v, f(\mathbf{x}_t)) - L(y_t, f(\mathbf{x}_t))] \, d\pi^*$$

$$\leq \underbrace{\int_{\mathcal{Z}^2} |L(y_v, f(\mathbf{x}_v)) - L(y_v, f(\mathbf{x}_t))| \, d\pi^*}_{U_1} + \underbrace{\int_{\mathcal{Z}^2} |L(y_v, f(\mathbf{x}_t)) - L(y_t, f(\mathbf{x}_t))| \, d\pi^*}_{U_2} \tag{23}$$

where the last inequality follows from the triangle inequality.

**Bounding $U_1$:**   Using the Lipschitz properties of $L$ and $f$:

$$U_1 \leq \int_{\mathcal{Z}^2} k|f(\mathbf{x}_v) - f(\mathbf{x}_t)| \, d\pi^*((\mathbf{x}_t, y_t), (\mathbf{x}_v, y_v)) \quad (L \text{ is } k\text{-Lipschitz})$$

$$\leq k\epsilon \int_{\mathcal{Z}^2} d_{\text{wav}}(\mathbf{x}_v, \mathbf{x}_t) \, d\pi^*((\mathbf{x}_t, y_t), (\mathbf{x}_v, y_v)) \quad (f \text{ is } \epsilon\text{-Lipschitz w.r.t. } d_{\text{wav}}). \tag{24}$$

**Bounding $U_2$:**   Using the $k$-Lipschitz property of $L$ in its first argument:

$$U_2 \leq \int_{\mathcal{Z}^2} k|y_v - y_t| \, d\pi^*((\mathbf{x}_t, y_t), (\mathbf{x}_v, y_v)). \tag{25}$$

Let $\pi^*_{\mathcal{X} \times \mathcal{X}}$ denote the marginal distribution of $\pi^*$ on $\mathcal{X} \times \mathcal{X}$:

$$\pi^*_{\mathcal{X} \times \mathcal{X}}(\mathbf{x}_t, \mathbf{x}_v) = \int_{\mathcal{Y}^2} d\pi^*((\mathbf{x}_t, y_t), (\mathbf{x}_v, y_v)). \tag{26}$$

Define the violation set:

$$V = \{(\mathbf{x}_t, \mathbf{x}_v) \in \mathcal{X} \times \mathcal{X} : |f_v(\mathbf{x}_v) - f_t(\mathbf{x}_t)| > \epsilon_{tv} \cdot d_{\text{wav}}(\mathbf{x}_t, \mathbf{x}_v)\}. \tag{27}$$

By assumption (iv) and Definition 4:

$$\pi^*_{\mathcal{X} \times \mathcal{X}}(V) = \mathbb{P}_{(\mathbf{x}_t, \mathbf{x}_v) \sim \pi^*_{\mathcal{X} \times \mathcal{X}}}[|f_v(\mathbf{x}_v) - f_t(\mathbf{x}_t)| > \epsilon_{tv} \cdot d_{\text{wav}}(\mathbf{x}_t, \mathbf{x}_v)] \leq \delta_{\text{wav}}. \tag{28}$$

Define the extended violation set in $\mathcal{Z}^2$:

$$\tilde{V} = \{((\mathbf{x}_t, y_t), (\mathbf{x}_v, y_v)) \in \mathcal{Z}^2 : (\mathbf{x}_t, \mathbf{x}_v) \in V\}. \tag{29}$$

Since $y_t = f_t(\mathbf{x}_t)$ and $y_v = f_v(\mathbf{x}_v)$ for the support of $\pi^*$:

$$\pi^*(\tilde{V}) = \pi^*_{\mathcal{X} \times \mathcal{X}}(V) \leq \delta_{\text{wav}}. \tag{30}$$

We can decompose $U_2$:

$$U_2 \leq \int_{\tilde{V}} k|y_v - y_t| \, d\pi^* + \int_{\mathcal{Z}^2 \setminus \tilde{V}} k|y_v - y_t| \, d\pi^*$$

$$\leq \int_{\tilde{V}} k \cdot 2M \, d\pi^* + \int_{\mathcal{Z}^2 \setminus \tilde{V}} k \cdot \epsilon_{tv} d_{\text{wav}}(\mathbf{x}_t, \mathbf{x}_v) \, d\pi^*$$

$$(\text{since } |y_v|, |y_t| \leq M \text{ and on } \mathcal{Z}^2 \setminus \tilde{V}: |y_v - y_t| \leq \epsilon_{tv} d_{\text{wav}}(\mathbf{x}_t, \mathbf{x}_v))$$

$$\leq 2kM \cdot \pi^*(\tilde{V}) + k\epsilon_{tv} \int_{\mathcal{Z}^2} d_{\text{wav}}(\mathbf{x}_t, \mathbf{x}_v) \, d\pi^*$$

$$\leq 2kM\delta_{\text{wav}} + k\epsilon_{tv} \int_{\mathcal{Z}^2} d_{\text{wav}}(\mathbf{x}_t, \mathbf{x}_v) \, d\pi^*. \tag{31}$$

**Combining the bounds:**   From Eqs. (24) and (31):

$$\Delta L \leq U_1 + U_2$$

$$\leq k\epsilon \int_{\mathcal{Z}^2} d_{\text{wav}}(\mathbf{x}_t, \mathbf{x}_v) \, d\pi^* + 2kM\delta_{\text{wav}} + k\epsilon_{tv} \int_{\mathcal{Z}^2} d_{\text{wav}}(\mathbf{x}_t, \mathbf{x}_v) \, d\pi^*$$

$$= k(\epsilon + \epsilon_{tv}) \int_{\mathcal{Z}^2} d_{\text{wav}}(\mathbf{x}_t, \mathbf{x}_v) \, d\pi^* + 2kM\delta_{\text{wav}}. \tag{32}$$

We now need to relate the integral $\int_{\mathcal{Z}^2} d_{\text{wav}}(\mathbf{x}_t, \mathbf{x}_v) d\pi^*$ to $\mathcal{W}_{\text{SW}}(\mu_t^{f_t}, \mu_v^{f_v})$. Let $\pi^*$ be the optimal coupling for $\mathcal{W}_{\text{SW}}$, i.e., the optimal transport plan that solves the above optimization problem. Consider the expected cost under this optimal plan:

$$\mathbb{E}_{\pi^*}[\mathbf{D}^{(W)}((\mathbf{x}_t, y_t), (\mathbf{x}_v, y_v))] = \int_{\mathcal{Z}^2} [d_{\text{wav}}(\mathbf{x}_t, \mathbf{x}_v) + c \cdot \mathcal{W}_{d_{\text{wav}}}(\mu_t(\cdot|y_t), \mu_v(\cdot|y_v))] d\pi^*$$

$$= \int_{\mathcal{Z}^2} d_{\text{wav}}(\mathbf{x}_t, \mathbf{x}_v) d\pi^* + c \cdot \int_{\mathcal{Z}^2} \mathcal{W}_{d_{\text{wav}}}(\mu_t(\cdot|y_t), \mu_v(\cdot|y_v)) d\pi^*$$

$$\tag{33}$$

Since the second term is non-negative (as the Wasserstein distance is non-negative), we have:

$$\int_{\mathcal{Z}^2} d_{\text{wav}}(\mathbf{x}_t, \mathbf{x}_v) d\pi^* \leq \mathbb{E}_{\pi^*}[\mathbf{D}^{(W)}((\mathbf{x}_t, y_t), (\mathbf{x}_v, y_v))] \tag{34}$$

Now, since $\pi^*$ is the optimal transport plan for $\mathcal{W}_{\text{SW}}$, we have:

$$\mathbb{E}_{\pi^*}[\mathbf{D}^{(W)}((\mathbf{x}_t, y_t), (\mathbf{x}_v, y_v))] = \langle \mathbf{D}^{(W)}, \pi^* \rangle \tag{35}$$

where $\langle \mathbf{D}^{(W)}, \pi^* \rangle$ denotes the Frobenius inner product between the cost matrix and transport plan.

Considering the contribution of the KL divergence terms at the optimal solution $\pi^*$, we have:

$$\mathcal{W}_{\text{SW}}(\mu_t^{f_t}, \mu_v^{f_v}) = \langle \mathbf{D}^{(W)}, \pi^* \rangle + \kappa D_{\text{KL}}(\pi^* \mathbf{1}_m \| \boldsymbol{\Delta}_n) + \kappa D_{\text{KL}}((\pi^*)^\top \mathbf{1}_n \| \boldsymbol{\Delta}_m) \tag{36}$$

This implies:

$$\langle \mathbf{D}^{(W)}, \pi^* \rangle \leq \mathcal{W}_{\text{SW}}(\mu_t^{f_t}, \mu_v^{f_v}) \tag{37}$$

since the KL divergence is non-negative, so $\kappa D_{\text{KL}}(\pi^* \mathbf{1}_m \| \boldsymbol{\Delta}_n) + \kappa D_{\text{KL}}((\pi^*)^\top \mathbf{1}_n \| \boldsymbol{\Delta}_m) \geq 0$.

Combining these relationships:

$$\int_{\mathcal{Z}^2} d_{\text{wav}}(\mathbf{x}_t, \mathbf{x}_v) d\pi^* \leq \mathbb{E}_{\pi^*}[\mathbf{D}^{(W)}((\mathbf{x}_t, y_t), (\mathbf{x}_v, y_v))] = \langle \mathbf{D}^{(W)}, \pi^* \rangle \leq \mathcal{W}_{\text{SW}}(\mu_t^{f_t}, \mu_v^{f_v}) \tag{38}$$

Therefore:

$$\int_{\mathcal{Z}^2} d_{\text{wav}}(\mathbf{x}_t, \mathbf{x}_v) d\pi^* \leq \mathcal{W}_{\text{SW}}(\mu_t^{f_t}, \mu_v^{f_v}) \tag{39}$$

Furthermore, in our derivation, if we assume that $\epsilon_{tv} \leq c\epsilon$ (i.e., the cross-Lipschitz constant is related to the hierarchical cost weight $c$ and the model Lipschitz constant $\epsilon$), then $\epsilon + \epsilon_{tv} \leq \epsilon(1 + c)$. This allows us to further simplify the upper bound:

$$\Delta L \leq k(\epsilon + \epsilon_{tv}) \int_{\mathcal{Z}^2} d_{\text{wav}}(\mathbf{x}_t, \mathbf{x}_v) d\pi^* + 2kM\delta_{\text{wav}} \tag{40}$$

$$\leq k\epsilon(1 + c) \int_{\mathcal{Z}^2} d_{\text{wav}}(\mathbf{x}_t, \mathbf{x}_v) d\pi^* + 2kM\delta_{\text{wav}} \tag{41}$$

$$\leq k\epsilon(1 + c) \mathcal{W}_{\text{SW}}(\mu_t^{f_t}, \mu_v^{f_v}) + 2kM\delta_{\text{wav}} \tag{42}$$

For simplicity of notation, we absorb the constant $(1 + c)$ into $\epsilon$, and write $k\epsilon(1 + c)$ as $k\epsilon$. This leads to:

$$\Delta L \leq k\epsilon \cdot \mathcal{W}_{\text{SW}}(\mu_t^{f_t}, \mu_v^{f_v}) + 2kM\delta_{\text{wav}} \tag{43}$$

This completes the derivation, showing that the gap between validation and training performance is bounded by the $\mathcal{W}_{\text{SW}}$ distance, which is our main result. □

### C.4 PROOF OF THEOREM 4: DATA VALUE SENSITIVITY

The proof proceeds in three main steps: establishing the dual formulation, proving uniqueness of the optimal solution, and deriving the properties of data values.

*Proof.* We begin with the primal unbalanced optimal transport problem. Using the KL divergence formulation $D_{\text{KL}}(p\|q) = \sum_i p_i \log(p_i/q_i) - p_i + q_i$, the primal problem becomes

$$\min_{\mathbf{T} \geq 0} \left\{ \langle \mathbf{D}^{(W)}, \mathbf{T} \rangle + \kappa \sum_{i=1}^n h\left(\frac{(\mathbf{T}\mathbf{1}_m)_i}{1/n}\right) + \kappa \sum_{j=1}^m h\left(\frac{(\mathbf{T}^\top \mathbf{1}_n)_j}{1/m}\right) \right\} \tag{44}$$

where $h(x) = x \log x - x + 1$ is the generalized KL divergence kernel. To derive the dual, we apply the Fenchel-Rockafellar duality theorem with the Lagrangian

$$\mathcal{L}(\mathbf{T}, \lambda) = \langle \mathbf{D}^{(W)}, \mathbf{T} \rangle + \kappa \sum_{i=1}^{n} h\left( \frac{(\mathbf{T}\mathbf{1}_m)_i}{1/n} \right) + \kappa \sum_{j=1}^{m} h\left( \frac{(\mathbf{T}^\top \mathbf{1}_n)_j}{1/m} \right) - \langle \lambda, \mathbf{T} \rangle. \quad (45)$$

Since the Fenchel conjugate of $h$ is $h^*(y) = e^y - 1$, we can apply the minimax theorem. Taking the gradient with respect to $\mathbf{T}$ and setting it to zero yields

$$\mathbf{D}_{ij}^{(W)} + \kappa \log\big(n(\mathbf{T}\mathbf{1}_m)_i\big) + \kappa \log\big(m(\mathbf{T}^\top \mathbf{1}_n)_j\big) = \lambda_{ij}. \quad (46)$$

Setting $\mathbf{f}_i = -\kappa \log\big(n(\mathbf{T}\mathbf{1}_m)_i\big)$ and $\mathbf{g}_j = -\kappa \log\big(m(\mathbf{T}^\top \mathbf{1}_n)_j\big)$, we obtain $\lambda_{ij} = \mathbf{D}_{ij}^{(W)} - \mathbf{f}_i - \mathbf{g}_j$. For $\mathbf{T} \geq 0$, we require $\lambda_{ij} \geq 0$, yielding the constraint $\mathbf{f}_i + \mathbf{g}_j \leq \mathbf{D}_{ij}^{(W)}$.

Substituting back into the dual objective and using $\psi_\kappa(u) = \kappa(1 - e^{-u/\kappa})$ as the Fenchel conjugate of the scaled entropy, we obtain the dual formulation:

$$\mathcal{W}_{\mathrm{SW}}(\mu_t, \mu_v) = \max_{\mathbf{f}, \mathbf{g}} \sum_{i=1}^{n} \frac{\kappa}{n} \psi_\kappa(\mathbf{f}_i) + \sum_{j=1}^{m} \frac{\kappa}{m} \psi_\kappa(\mathbf{g}_j) \quad (47)$$

subject to $\mathbf{f}_i + \mathbf{g}_j \leq \mathbf{D}_{ij}^{(W)}$ for all $i, j$.

To establish uniqueness of the optimal solution, we analyze the dual objective function

$$L(\mathbf{f}, \mathbf{g}) = \sum_{i=1}^{n} \frac{\kappa}{n} \psi_\kappa(\mathbf{f}_i) + \sum_{j=1}^{m} \frac{\kappa}{m} \psi_\kappa(\mathbf{g}_j). \quad (48)$$

The second derivative of $\psi_\kappa$ is $\frac{d^2\psi_\kappa(u)}{du^2} = -\frac{1}{\kappa} e^{-u/\kappa} < 0$, showing that $\psi_\kappa$ is strictly concave. Therefore, the objective function $L(\mathbf{f}, \mathbf{g})$ is strictly concave in $(\mathbf{f}, \mathbf{g})$, while the constraint set defined by linear inequalities is convex. By the fundamental theorem of convex optimization, the optimal dual solution $(\mathbf{f}^*, \mathbf{g}^*)$ is unique and satisfies the KKT conditions: $T_{ij}^* > 0 \Rightarrow \mathbf{f}_i^* + \mathbf{g}_j^* = \mathbf{D}_{ij}^{(W)}$, along with the primal-dual relationships $\sum_{j=1}^{m} T_{ij}^* = \frac{1}{n} e^{-\mathbf{f}_i^*/\kappa}$ and $\sum_{i=1}^{n} T_{ij}^* = \frac{1}{m} e^{-\mathbf{g}_j^*/\kappa}$.

Given the unique optimal dual solution, we define the relative data value as

$$v(\mathbf{x}_i) = -\left[ \psi_\kappa(\mathbf{f}_i^*) - \frac{1}{n-1} \sum_{j \neq i} \psi_\kappa(\mathbf{f}_j^*) \right] \quad (49)$$

The uniqueness and deterministic ranking properties follow directly from the uniqueness of $(\mathbf{f}^*, \mathbf{g}^*)$ and the strict monotonicity of $\psi_\kappa$. For interpretability, $v(\mathbf{x}_i) > 0$ indicates that $\psi_\kappa(\mathbf{f}_i^*)$ is below the average value of other samples, suggesting that this sample helps align the distributions (lower transport cost).

To establish the sensitivity interpretation, we consider a mass perturbation around sample $i$:

$$\mu^\delta = \left( \frac{1}{n} + \delta \right) \delta_{\mathbf{x}_i} + \sum_{j \neq i} \left( \frac{1}{n} - \frac{\delta}{n-1} \right) \delta_{\mathbf{x}_j} \quad (50)$$

which preserves total mass. The directional derivative of $\mathcal{W}_{\mathrm{SW}}$ with respect to this perturbation at $\delta = 0$ is

$$\frac{d}{d\delta} \mathcal{W}_{\mathrm{SW}}(\mu^\delta, \mu_v) \bigg|_{\delta=0} = \frac{\partial \mathcal{W}_{\mathrm{SW}}}{\partial \mu_i} - \frac{1}{n-1} \sum_{j \neq i} \frac{\partial \mathcal{W}_{\mathrm{SW}}}{\partial \mu_j}. \quad (51)$$

By the envelope theorem applied to the dual formulation, $\frac{\partial \mathcal{W}_{\mathrm{SW}}}{\partial \mu_i} = \frac{\kappa}{n} \psi_\kappa(\mathbf{f}_i^*)$, which yields

$$\frac{d}{d\delta} \mathcal{W}_{\mathrm{SW}}(\mu^\delta, \mu_v) \bigg|_{\delta=0} = \frac{\kappa}{n} \left[ \psi_\kappa(\mathbf{f}_i^*) - \frac{1}{n-1} \sum_{j \neq i} \psi_\kappa(\mathbf{f}_j^*) \right] = -\frac{\kappa}{n} v(\mathbf{x}_i). \quad (52)$$

This shows that $v(\mathbf{x}_i)$ measures the sensitivity of the $\mathcal{W}_{\mathrm{SW}}$ distance to local mass perturbations at sample $i$, providing an economic interpretation of data value in terms of marginal contribution to distributional alignment. $\quad\square$

## C.5 Proof of Theorem 6: Convergence of Entropy-Regularized Approximation

**Theorem 6** (Convergence of Entropy-Regularized Approximation). *Let $v(\mathbf{x}_i)$ and $v_\varepsilon(\mathbf{x}_i)$ be the data values derived from the optimal dual potentials $(\mathbf{f}^*, \mathbf{g}^*)$ and $(\mathbf{f}_\varepsilon^*, \mathbf{g}_\varepsilon^*)$ of the unregularized and entropy-regularized $\mathcal{W}_{SW}$ problems, respectively. Then:*

1. ***Pointwise Convergence:*** *As $\varepsilon \to 0$, we have $v_\varepsilon(\mathbf{x}_i) \to v(\mathbf{x}_i)$ for all $i \in [n]$.*

2. ***Ranking Preservation:*** *For any pair of time series segments such that $v(\mathbf{x}_i) > v(\mathbf{x}_j)$, there exists an $\varepsilon_0 > 0$ such that for all $0 < \varepsilon < \varepsilon_0$, the ranking is preserved: $v_\varepsilon(\mathbf{x}_i) > v_\varepsilon(\mathbf{x}_j)$.*

*Proof.* The proof builds on the convergence theory for unbalanced optimal transport (Séjourné et al., 2019; Pham et al., 2020). The data values are defined as functions of the optimal dual potentials:

$$v(\mathbf{x}_i) = \psi_\kappa(\mathbf{f}_i^*) - \frac{1}{n-1} \sum_{j \neq i} \psi_\kappa(\mathbf{f}_j^*) \tag{53}$$

$$v_\varepsilon(\mathbf{x}_i) = \psi_\kappa(\mathbf{f}_{\varepsilon,i}^*) - \frac{1}{n-1} \sum_{j \neq i} \psi_\kappa(\mathbf{f}_{\varepsilon,j}^*) \tag{54}$$

where $\psi_\kappa(u) = \kappa(1 - e^{-u/\kappa})$ is the transformation function arising from the KL divergence penalty.

Following Séjourné et al. (2019) (Proposition 21), the optimal dual potentials are uniformly bounded: there exists $M > 0$ independent of $\varepsilon$ such that $\|\mathbf{f}^*\|_\infty, \|\mathbf{f}_\varepsilon^*\|_\infty \leq M$ and $\|\mathbf{g}^*\|_\infty, \|\mathbf{g}_\varepsilon^*\|_\infty \leq M$. This boundedness is established through the coercivity of the dual functional and holds for unbalanced optimal transport with KL divergence penalties on compact domains.

**Convergence of Dual Potentials.** From Séjourné et al. (2019) (Proposition 10), for the unbalanced optimal transport setting with KL divergence penalties, under the assumptions of compact domain $\mathcal{X}$ and Lipschitz continuous cost $D^{(W)}$, the optimal dual potentials converge uniformly as the regularization parameter vanishes:

$$\lim_{\varepsilon \to 0} \|\mathbf{f}_\varepsilon^* - \mathbf{f}^*\|_\infty = 0 \quad \text{and} \quad \lim_{\varepsilon \to 0} \|\mathbf{g}_\varepsilon^* - \mathbf{g}^*\|_\infty = 0 \tag{55}$$

This convergence is established through weak continuous dependence of dual potentials on the regularization parameter. Furthermore, Séjourné et al. (2019) (Theorem 1) guarantees that for any fixed $\varepsilon > 0$, the optimal potentials $(\mathbf{f}_\varepsilon^*, \mathbf{g}_\varepsilon^*)$ can be computed via the Sinkhorn algorithm with linear convergence.

**Convergence of Data Values.** The function $\psi_\kappa(u) = \kappa(1 - e^{-u/\kappa})$ has derivative $\psi_\kappa'(u) = e^{-u/\kappa}$. Since the dual potentials are uniformly bounded in $[-M, M]$, $\psi_\kappa$ is Lipschitz continuous on this compact set with Lipschitz constant:

$$L = \sup_{u \in [-M,M]} |\psi_\kappa'(u)| = \sup_{u \in [-M,M]} e^{-u/\kappa} = e^{M/\kappa} < \infty \tag{56}$$

Using this Lipschitz property and the triangle inequality:

$$|v_\varepsilon(\mathbf{x}_i) - v(\mathbf{x}_i)| = \left| \left( \psi_\kappa(\mathbf{f}_{\varepsilon,i}^*) - \psi_\kappa(\mathbf{f}_i^*) \right) - \frac{1}{n-1} \sum_{j \neq i} \left( \psi_\kappa(\mathbf{f}_{\varepsilon,j}^*) - \psi_\kappa(\mathbf{f}_j^*) \right) \right| \tag{57}$$

$$\leq |\psi_\kappa(\mathbf{f}_{\varepsilon,i}^*) - \psi_\kappa(\mathbf{f}_i^*)| + \frac{1}{n-1} \sum_{j \neq i} |\psi_\kappa(\mathbf{f}_{\varepsilon,j}^*) - \psi_\kappa(\mathbf{f}_j^*)| \tag{58}$$

$$\leq L|\mathbf{f}_{\varepsilon,i}^* - \mathbf{f}_i^*| + \frac{L}{n-1} \sum_{j \neq i} |\mathbf{f}_{\varepsilon,j}^* - \mathbf{f}_j^*| \tag{59}$$

$$\leq L\|\mathbf{f}_\varepsilon^* - \mathbf{f}^*\|_\infty + L\|\mathbf{f}_\varepsilon^* - \mathbf{f}^*\|_\infty \tag{60}$$

$$= 2L\|\mathbf{f}_\varepsilon^* - \mathbf{f}^*\|_\infty \tag{61}$$

From above, we know that $\|\mathbf{f}_\varepsilon^* - \mathbf{f}^*\|_\infty \to 0$ as $\varepsilon \to 0$. Since $L$ is a finite constant (independent of $\varepsilon$), we obtain:

$$\lim_{\varepsilon \to 0} |v_\varepsilon(\mathbf{x}_i) - v(\mathbf{x}_i)| = \lim_{\varepsilon \to 0} 2L\|\mathbf{f}_\varepsilon^* - \mathbf{f}^*\|_\infty = 0 \quad \text{for all } i \in [n] \tag{62}$$

which establishes pointwise convergence.

**Ranking Preservation.** Suppose $v(\mathbf{x}_i) > v(\mathbf{x}_j)$ and let $\delta = v(\mathbf{x}_i) - v(\mathbf{x}_j) > 0$ be the gap between these data values. From the pointwise convergence established in Step 3, for the positive value $\delta/3$, there exists an $\varepsilon_0 > 0$ such that for all $0 < \varepsilon < \varepsilon_0$:

$$|v_\varepsilon(\mathbf{x}_k) - v(\mathbf{x}_k)| < \frac{\delta}{3} \quad \text{for all } k \in \{i, j\} \tag{63}$$

Using this bound:

$$v_\varepsilon(\mathbf{x}_i) - v_\varepsilon(\mathbf{x}_j) = (v(\mathbf{x}_i) - v(\mathbf{x}_j)) + (v_\varepsilon(\mathbf{x}_i) - v(\mathbf{x}_i)) - (v_\varepsilon(\mathbf{x}_j) - v(\mathbf{x}_j)) \tag{64}$$

$$= \delta + (v_\varepsilon(\mathbf{x}_i) - v(\mathbf{x}_i)) - (v_\varepsilon(\mathbf{x}_j) - v(\mathbf{x}_j)) \tag{65}$$

$$\geq \delta - |v_\varepsilon(\mathbf{x}_i) - v(\mathbf{x}_i)| - |v_\varepsilon(\mathbf{x}_j) - v(\mathbf{x}_j)| \tag{66}$$

$$> \delta - \frac{\delta}{3} - \frac{\delta}{3} \tag{67}$$

$$= \frac{\delta}{3} > 0 \tag{68}$$

Therefore, for all $0 < \varepsilon < \varepsilon_0$, we have $v_\varepsilon(\mathbf{x}_i) > v_\varepsilon(\mathbf{x}_j)$, which proves that the ranking is preserved.
$\square$

**Numerical Verification of Theorem 6.** We verify the convergence properties using synthetic time series data. We generate 40 training time series and 30 validation time series, each segmented into overlapping windows. This yields training segments labeled as high quality (clean sinusoids, $\sigma = 0.05$), medium quality (moderate noise with harmonics, $\sigma = 0.2$), or low quality (pure noise or severe outliers). We compute segment-wise data values using $\varepsilon \in [10^{-4}, 1]$ with 20 logarithmically-spaced values and $\kappa = 1.0$.

Fig.8 presents three verification aspects. The left plot shows segment value trajectories for 10 representative segments, demonstrating pointwise convergence as $\varepsilon \to 0$. The middle plot displays $L_1$ and $L_\infty$ errors between $v_\varepsilon$ and the reference solution ($\varepsilon_{\text{ref}} = 10^{-4}$) on a log-log scale. The empirical convergence rate of $O(\varepsilon^{0.88})$ closely matches the theoretical $O(\varepsilon)$ prediction, with the $L_\infty$ error naturally exceeding $L_1$ as it captures worst-case segment behavior. The right plot shows Spearman correlation coefficients consistently above 0.95, confirming that segment rankings remain stable across all $\varepsilon$ values. These results validate both claims of Theorem 6: (i) segment-wise convergence $v_\varepsilon(\mathbf{x}_i) \to v(\mathbf{x}_i)$ as $\varepsilon \to 0$, and (ii) ranking preservation across the full range of regularization parameters.

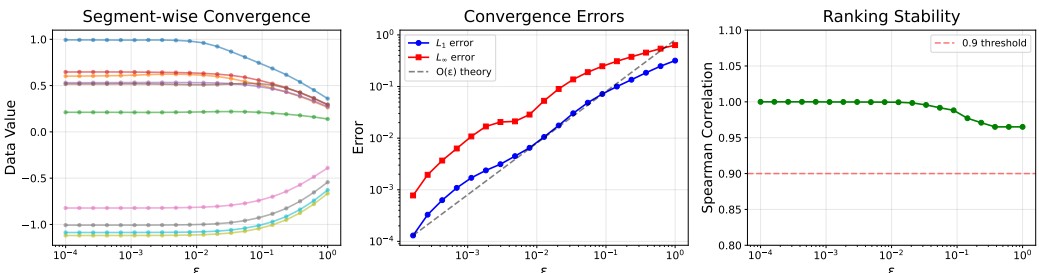

Figure 8: Numerical verification of entropy-regularized UOT convergence for time series segments. Left: Value trajectories showing segment-wise convergence. Middle: $L_1$ and $L_\infty$ convergence errors on log-log scale with theoretical $O(\varepsilon)$ reference (dashed). Right: Spearman correlation demonstrating ranking stability.

# D EXPERIMENTS

## D.1 THE TIMELAVA ALGORITHM

In this appendix, we present the segment-wise and point-wise valuation algorithms for time series, which together complete the TIMELAVA framework. We also provide an analysis of the computational complexity of the algorithm and clarify the role of the reference time series.

### D.1.1 SEGMENT-WISE VALUATION ALGORITHM

Building on the methodology developed, the segment-wise valuation procedure of TIMELAVA is summarized in Algorithm 1.

---

**Algorithm 1** TIMELAVA Algorithm

---

**Input:** Training set $\mathcal{D}_t$, validation set $\mathcal{D}_v$, wavelet parameters $(\psi, \text{level})$, cost parameters $(c, \kappa)$, entropy regularization $\epsilon$
**Output:** Data values $\{v_\epsilon(\mathbf{x}_i)\}_{i=1}^N$
    // Compute Wavelet Representations
 1: Compute DWT coefficients $\mathcal{W}(\mathbf{x}_i)$ for all $\mathbf{x}_i \in \mathcal{D}_t$ and $\mathcal{W}(\mathbf{x}'_j)$ for all $\mathbf{x}'_j \in \mathcal{D}_v$.
    // Compute Pairwise Cost Matrix $\mathbf{D}^{(W)}$
 2: **for** $i = 1$ to $N$ **do**
 3:     **for** $j = 1$ to $M$ **do**
 4:         Compute ground wavelet distance $d_{\text{wav}}(\mathbf{x}_i, \mathbf{x}'_j) = \|\mathcal{W}(\mathbf{x}_i) - \mathcal{W}(\mathbf{x}'_j)\|_1$.
 5:         **if** $c > 0$ **then**
 6:             Compute $\mathcal{W}_{d_{\text{wav}}}\big(\mu_t(\cdot|y_i), \mu_v(\cdot|y'_j)\big)$ using $d_{\text{wav}}$ as the ground metric.
 7:             Set $\mathbf{D}_{i,j}^{(W)} = d_{\text{wav}}(\mathbf{x}_i, \mathbf{x}'_j) + c \cdot \mathcal{W}_{d_{\text{wav}}}(\dots)$.
 8:         **else**
 9:             Set $\mathbf{D}_{i,j}^{(W)} = d_{\text{wav}}(\mathbf{x}_i, \mathbf{x}'_j)$.
10:         **end if**
11:     **end for**
12: **end for**
    // Solve Regularized UOT Problem
13: Solve the entropy-regularized UOT problem using Sinkhorn algorithm:

$$\min_{\mathbf{T} \geq 0} \langle \mathbf{D}^{(W)}, \mathbf{T} \rangle + \kappa D_{\text{KL}}(\mathbf{T}\mathbf{1}_m \| \mathbf{\Delta}_n) + \kappa D_{\text{KL}}(\mathbf{T}^T \mathbf{1}_n \| \mathbf{\Delta}_m) - \varepsilon H(\mathbf{T})$$

    to obtain the optimal dual variables $(\mathbf{f}_\varepsilon^*, \mathbf{g}_\varepsilon^*)$.
    // Compute Data Values
14: **for** $i = 1$ to $N$ **do**
15:     Compute $\phi_i = \psi_\kappa(\mathbf{f}_{\varepsilon,i}^*) = \kappa(1 - e^{-\mathbf{f}_{\varepsilon,i}^*/\kappa})$.
16: **end for**
17: Compute sum $S = \sum_{j=1}^N \phi_j$.
18: **for** $i = 1$ to $N$ **do**
19:     Compute $v_\varepsilon(\mathbf{x}_i) = -(\phi_i - \frac{S - \phi_i}{N - 1})$.
20: **end for**
21: **return** $\{v_\epsilon(\mathbf{x}_i)\}_{i=1}^N$.

---

### D.1.2 POINT-WISE VALUATION ALGORITHM

Algorithm 2 outlines the mapping process from segment contributions to point-wise contributions.

### D.1.3 COMPUTATIONAL COMPLEXITY

The computational complexity of TIMELAVA is dominated by three components. First, the wavelet transform for $N$ training and $M$ validation segments of length $L$ requires $O((N + M) \cdot L)$ using Mallat's pyramidal algorithm. Second, computing the pairwise cost matrix requires $O(N \cdot M \cdot L)$ operations. Third, solving the entropy-regularized UOT problem requires $O(N \cdot M \cdot \log(1/\epsilon))$

---

**Algorithm 2** Point-wise Contribution Calculation

---

**Input:** Segment contribution values $\{v_\epsilon(\mathbf{z}^{[m]})\}$, original time series length $N$, segment length $T$, sliding stride $s$

**Output:** Point-wise contribution values $\{v(t)\}_{t=1}^N$

1: Initialize contribution counters $\text{count}[t] = 0$ and contribution sums $\text{sum}[t] = 0$ for each time point $t = 1, 2, \ldots, N$
2: **for** each segment $\mathbf{z}^{[m]}$ and its contribution value $v_\epsilon(\mathbf{z}^{[m]})$ **do**
3:     Determine the time range $[\text{start}, \text{end}]$ covered by this segment
4:     **for** $t \in [\text{start}, \text{end}]$ **do**
5:         $\text{sum}[t] \mathrel{+}= v_\epsilon(\mathbf{z}^{[m]})$
6:         $\text{count}[t] \mathrel{+}= 1$
7:     **end for**
8: **end for**
9: **for** each time point $t = 1, 2, \ldots, N$ **do**
10:     **if** $\text{count}[t] > 0$ **then**
11:         $v(t) = \text{sum}[t]/\text{count}[t]$
12:     **else**
13:         $v(t) = 0$                                    ▷ Point not covered by any segment
14:     **end if**
15: **end for**
16: **return** $\{v(t)\}_{t=1}^N$

---

operations. Together, the overall complexity is $O((N + M) \cdot L + N \cdot M \cdot L)$, which simplifies to $O(N \cdot M \cdot L)$ when $N \cdot M >> N + M$, as is typical in practice. When using label information $(c > 0)$, an additional $O(N \cdot M \cdot K^2)$ term arises from computing conditional Wasserstein distances, where $K$ is the average number of segments per label class. All experiments were conducted on a workstation equipped with an Intel Xeon Gold 6448H CPU (2.0 GHz, 32 cores), and 512 GB RAM. While our implementation is CPU-based, it can achieve even faster runtimes with GPU acceleration. Unless otherwise specified, we set $\kappa = 2.0$ for the UOT regularization parameter, $\varepsilon = 0.01$ for the entropy regularization, and use a Daubechies-4 (db4) wavelet with decomposition level 2 across all experiments.

### D.1.4 ON THE ROLE OF REFERENCE TIME SERIES

In learning-agnostic valuation frameworks, the *reference distribution* is the distribution against which the contributions of training data are evaluated. In i.i.d. settings such as LAVA, the reference set is the validation set (Just et al., 2023). For time series applications, however, the choice of reference data depends on the task: validation sequences in forecasting, normal (non-anomalous) segments in anomaly detection, or clean segment–label pairs in label-noise detection. In our experiments, we adopt these task-specific settings when defining the reference time series. In all cases, the reference distribution provides a meaningful anchor, allowing valuation to quantify how much each training segment contributes to aligning the training distribution with the target task distribution.

### D.2 ANOMALY DETECTION

Details of the datasets used in our anomaly detection experiments are summarized in Table 3. For each dataset, we report the dimensionality, number of time series, average length, and average anomaly ratio. We categorize the datasets into two groups:

1. **Pre-partitioned datasets** provide predefined splits consisting of a clean validation set, a contaminated test set with anomalies, and the corresponding labels (e.g., SMD (Su et al., 2019), SMAP and MSL (Hundman et al., 2018), PSM (Abdulaal et al., 2021), SWaT (Mathur & Tippenhauer, 2016b)). We directly adopt these official splits for evaluating TIMELAVA.

2. **Single-set datasets** consist of a single continuous time series with anomaly labels only (e.g., WADI (Ahmed et al., 2017), NAB (Ahmad et al., 2017), KDD-CUP99 (Stolfo et al., 2000)). We manually partition each dataset into a clean validation set and a contaminated test set.

Table 3: Details of the anomaly detection datasets. Note that all discrete-valued dimensions are excluded from each dataset. The term "Average Length" refers to the average length across all time series in the dataset.

| Dataset | Dimensions | Num. of Time Series | Average Length | Average Anomaly Ratio |
|---|---|---|---|---|
| UCR | 1 | 250 | 56,205 | 0.8% |
| SMAP | 25 | 55 | 8,068 | 12.8% |
| MSL | 55 | 27 | 2,730 | 10.5% |
| NAB-Traffic | 1 | 7 | 2,238 | 10.0% |
| SMD | 38 | 28 | 25,300 | 4.2% |
| NAB-AdExchange | 1 | 6 | 1,602 | 10.0% |
| NAB-Tweets | 1 | 10 | 15,863 | 9.9% |
| NAB-Taxi | 1 | 1 | 10,320 | 5.2% |
| PSM | 25 | 1 | 87,841 | 27.8% |
| SWaT | 51 | 1 | 449,919 | 12.1% |
| WADI | 123 | 1 | 172,751 | 5.7% |
| KDD-CUP99 | 34 | 1 | 494,021 | 19.7% |

Table 4: Supplementary result details. Higher AUC/F1 is better (↑).

| Method | SWaT | | WADI | | NAB-Taxi | | NAB-Tweets | | NAB-AdExchange | | NAB-Traffic | | KDD-Cup99 | |
|---|---|---|---|---|---|---|---|---|---|---|---|---|---|---|
| | AUC ↑ | F1 ↑ | AUC ↑ | F1 ↑ | AUC ↑ | F1 ↑ | AUC ↑ | F1 ↑ | AUC ↑ | F1 ↑ | AUC ↑ | F1 ↑ | AUC ↑ | F1 ↑ |
| Isolation Forest | **0.87** | **0.69** | 0.74 | 0.34 | **0.64** | 0.13 | 0.55 | 0.18 | 0.50 | 0.11 | 0.57 | 0.20 | 0.74 | 0.41 |
| LSTM | 0.30 | 0.16 | 0.49 | 0.08 | 0.33 | 0.04 | 0.55 | 0.17 | 0.56 | 0.12 | 0.57 | 0.19 | **0.98** | 0.88 |
| ARIMA/VAR | 0.46 | 0.11 | 0.41 | 0.04 | 0.49 | 0.05 | 0.56 | 0.17 | 0.51 | 0.14 | 0.55 | 0.20 | 0.87 | 0.52 |
| Anomaly Transformer | 0.60 | 0.22 | 0.50 | 0.06 | 0.51 | 0.06 | 0.50 | 0.08 | 0.44 | 0.07 | 0.46 | 0.05 | 0.54 | 0.17 |
| TimeInf | 0.61 | 0.08 | 0.63 | 0.14 | 0.52 | 0.02 | 0.52 | 0.16 | 0.54 | **0.34** | 0.64 | **0.39** | 0.79 | 0.43 |
| TIMELAVA (Ours) | 0.86 | 0.68 | **0.85** | **0.43** | 0.56 | **0.18** | **0.64** | **0.24** | **0.73** | 0.27 | **0.74** | 0.34 | **0.98** | **0.93** |

### D.2.1 ADDITIONAL EXPERIMENTAL RESULTS

Table 4 and Fig. 10 demonstrate TIMELAVA's consistently strong performance across all datasets, achieving the highest AUC on four of six benchmarks and leading F1 scores on NAB-Traffic and NAB-Tweets. This performance spans both univariate and multivariate anomaly detection tasks.

In contrast, TimeInf, despite being designed for time series valuation, shows limited effectiveness, particularly for local contextual and point anomalies, as shown in Fig. 10. This weakness stems from TimeInf's reliance on influence functions, which assume approximate stationarity and struggle to capture highly localized temporal events. While influence functions excel at measuring global perturbation effects, they lack the time-frequency localization needed to identify brief, transient anomalies that manifest at specific temporal scales, a capability that is instead provided by TIMELAVA's wavelet-based approach. Moreover, TIMELAVA's performance benefits from larger datasets. Larger datasets provide more segment pairs for UOT matching, reducing valuation score variance and improving stability. This trend is evident across our benchmarks: as dataset size increases from SMAP (8k points, AUC=0.74) to PSM (87k, AUC=0.77) to SWaT (449k, AUC=0.86) to KDD-Cup99 (494k, AUC=0.98), performance consistently improves.

Additionally, our experimental setup intentionally restricts reference time series to approximately 2,000 consecutive time points for large datasets (e.g. PSM), reflecting realistic data-scarce scenarios that are common in practical deployments. This constraint particularly impacts deep learning methods: TranAD, DCdetector and AnomalyTransformer require extensive parameterization, which depends on large-scale training data to learn robust temporal representations. When reference data is scarce, TIMELAVA maintains strong performance while deep learning baselines deteriorate significantly, demonstrating its superior robustness and practical applicability, a common challenge in real-world anomaly detection systems.

We have also included the comparison of anomaly score distributions between normal and anomalous segments to *visually* assess the discriminative capability of different methods (TIMELAVA, TimeInf, IsolationForest, and AnomalyTransformer) on the PSM dataset. As shown in Fig. 9, TIMELAVA demonstrates the clearest separation with minimal overlap, indicating superior discriminative power compared to other methods.

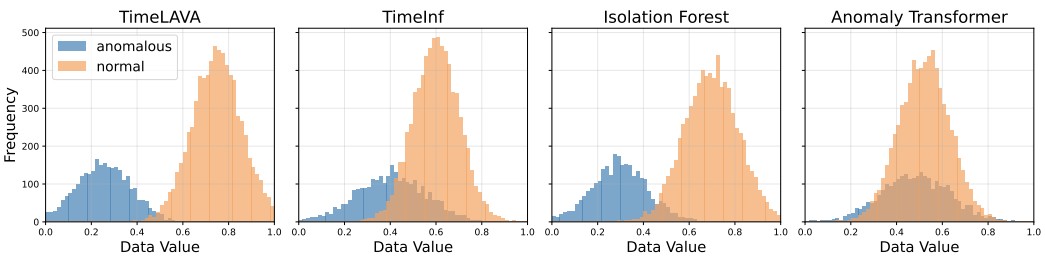

Figure 9: Data Selection and Pruning Results - Traffic

Figure 10: Comprehensive results on the **UCR Time Series Anomaly Archive.**

### D.2.2  PARAMETER SENSITIVITY ANALYSIS

In this section, we analyze the impact of key parameters on the performance of TIMELAVA using the SMAP and MSL datasets (Hundman et al., 2018). The main observations can be summarized as follows:

**Segment Length.** As shown in Fig. 11, increasing the segment length generally leads to improved AUC for both datasets. This observation is consistent with the intuition that longer segments provide a broader temporal context, enabling the wavelet transform to capture more comprehensive patterns and dependencies.

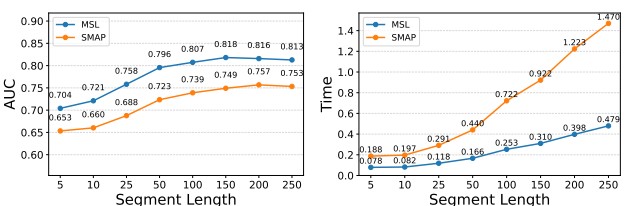

Figure 11: Parameter sensitivity analysis on segment length for TIMELAVA.

For the MSL dataset, AUC improves steadily from $0.704$ at a length of $5$ to approximately $0.818$ at a length of $150$, after which the gains plateau. A similar trend is observed for SMAP. The empirical growth in computational cost observed in Fig. 11 aligns with the theoretical time complexity of the discrete wavelet transform, which is $O(L)$ in the segment length $L$. Hence, there is a clear trade-off between detection performance and runtime: longer segments enhance detection accuracy by capturing richer temporal structure and preserving local patterns, but they also incur higher computational overhead. Moreover, the appropriate segment length is often domain dependent, as temporal dynamics unfold at different scales in applications such as electricity consumption, mobility, or traffic monitoring.

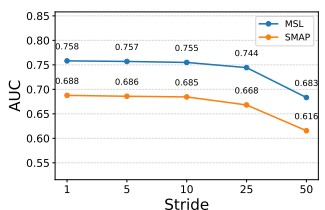

Figure 12: Parameter sensitivity analysis on stride for TIMELAVA.

**Stride.** Fig. 12 illustrates the impact of stride on performance. Smaller stride consistently yields higher AUC for both datasets, as increased overlap between consecutive segments preserves fine-grained temporal information, reducing the risk of missing anomalies of just a few points. As the stride increases from 1 to 50, AUC decreases notably, for instance, MSL drops from $0.76$ to $0.68$ and SMAP from $0.69$ to $0.62$. This degradation aligns with the intuition that larger stride will lose certain local information. Therefore, in practice we avoid setting the stride too large, as non-overlapping segments risk missing short anomalies and degrading detection accuracy.

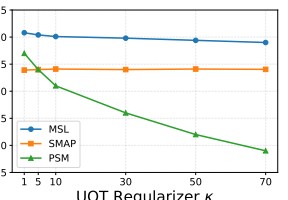

Figure 13: Parameter sensitivity analysis on UOT regularizer $\kappa$.

**UOT Regularizer.** Figure 13 illustrates the effect of the UOT regularization strength $\kappa$ on performance. TIMELAVA remains highly stable for $\kappa$ between 1 and 10, where the AUC varies only minimally; our default choice $\kappa=2$ lies well within this flat and reliable region. As $\kappa$ increases beyond 30, the formulation begins to resemble balanced OT, which forces mass-conserving matches between segments. This reduces the method's ability to downweight unmatched or outlier segments and leads to a gradual decline in performance. Overall, the trend demonstrates that TIMELAVA is robust to $\kappa$ over a wide operating range, with degradation only at extreme values.

### D.2.3 GENERALITY ANALYSIS

In this section, we explore alternative implementations to the key components of $\mathcal{W}_{\text{SW}}$ to justify its rationale and advantages. Table 5 present an ablation study evaluating the impact of two key components in TIMELAVA: the UOT formulation and the wavelet-based representation.

**Effects of the Discrepancy Measure.** Standard OT enforces strict mass-preserving matching constraints, requiring every unit of probability mass to be transported exactly. This becomes problematic under non-stationarity: when anomalies introduce or remove structure, OT spreads the cost across both normal and anomalous regions, blurring detection boundaries. UOT addresses this by relaxing mass conservation through divergence regularization (Fatras et al., 2021; Séjourné et al., 2019), allowing unmatched mass to be penalized directly. This produces sharper, more localized anomaly scores. Replacing UOT with standard OT (TIMELAVA-OT) leads to consistent drops in both AUC and F1 across most datasets (Table 5), with Fig. 14 illustrating how UOT's selective matching yields

Table 5: **Ablation study on key components of TIMELAVA.** TIMELAVA-OT replaces UOT with OT, and TIMELAVA-Fourier replaces Wavelet with Fourier. **Bold** = best, underline = second best.

| Dataset | TIMELAVA-OT | | | TIMELAVA-Fourier | | | **TIMELAVA (Ours)** | | |
|---|---|---|---|---|---|---|---|---|---|
| | AUC ↑ | F1 ↑ | Time (s) ↓ | AUC ↑ | F1 ↑ | Time (s) ↓ | AUC ↑ | F1 ↑ | Time (s) ↓ |
| SMD | 0.59 | 0.23 | 24.65 | 0.75 | 0.21 | 24.61 | **0.91** | **0.52** | 14.01 |
| SMAP | 0.74 | 0.48 | 2.13 | **0.79** | **0.63** | 1.04 | 0.74 | 0.54 | 0.72 |
| MSL | 0.79 | 0.49 | 0.68 | **0.84** | **0.57** | 0.41 | 0.81 | 0.49 | 0.25 |
| PSM | 0.59 | 0.38 | 51.18 | 0.74 | 0.57 | 10.68 | **0.77** | **0.58** | 8.16 |
| KDD-Cup99 | **0.98** | 0.92 | 815.46 | **0.98** | 0.92 | 296.01 | **0.98** | **0.93** | 101.52 |
| NAB-Taxi | 0.50 | 0.00 | 0.52 | 0.50 | 0.00 | 0.90 | **0.56** | **0.18** | 0.52 |
| NAB-Tweets | 0.62 | 0.19 | 0.92 | 0.58 | 0.05 | 1.10 | **0.64** | **0.24** | 0.47 |
| NAB-AdEx | 0.71 | 0.31 | 0.05 | **0.76** | **0.47** | 0.09 | 0.73 | 0.27 | 0.02 |
| SWaT | 0.32 | 0.10 | 578.68 | 0.40 | 0.13 | 226.11 | **0.86** | **0.68** | 245.83 |
| WADI | 0.64 | 0.25 | 741.74 | 0.63 | 0.23 | 148.89 | **0.85** | **0.43** | 91.13 |
| NAB-Traffic | 0.60 | 0.21 | 0.09 | 0.48 | 0.00 | 0.04 | **0.74** | **0.34** | 0.03 |

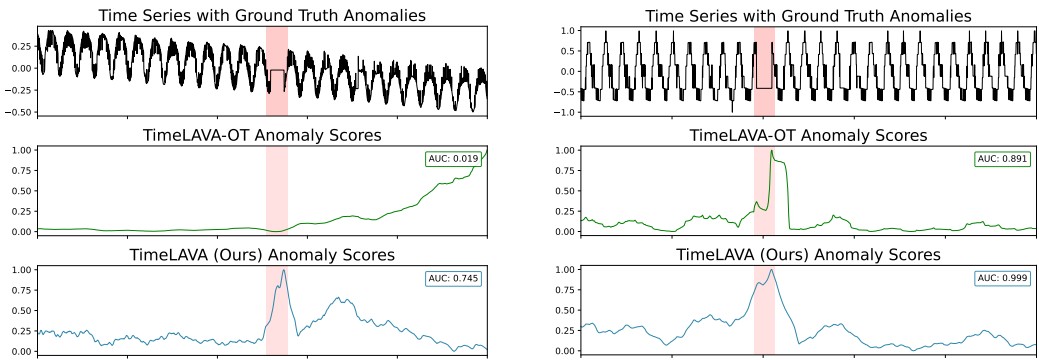

Figure 14: Anomaly detection on SMAP channels G-1 (left) and A-2 (right). TIMELAVA with UOT (bottom) shows superior anomaly detection compared to TIMELAVA-OT with standard OT (middle), achieving significantly higher AUC scores.

superior anomaly localization. These empirical findings also corroborate Theorem 2, which shows that the distribution patterns of data value scores can distinguish isolated anomalies from systemic regime shifts in real data.

**Effects of the Distance Metric.** Fourier transform provides global frequency decomposition but suffers from poor temporal localization. In contrast, wavelet transform offers joint time-frequency decomposition, preserving both spectral and temporal localization (Fig. 2). This multi-scale representation is crucial for detecting anomalies manifesting as short-lived bursts, shifts, or local contextual changes. Substituting wavelets with Fourier (TIMELAVA-Fourier) generally degrades performance, particularly for localized anomalies. Interestingly, TIMELAVA-Fourier achieves higher scores on SMAP and MSL datasets, but this stems from ground-truth label inaccuracies, where anomalies are annotated as extended intervals rather than precise points (Fig. 15). While TIMELAVA identifies exact anomaly onsets, TIMELAVA-Fourier flags broader regions, better matching the overly wide labels and inflating metrics, though TIMELAVA's finer localization more accurately reflects true anomaly occurrences.

The proposed TIMELAVA, integrating both UOT and wavelet transform, achieves the highest AUC and F1 in the majority of datasets, demonstrating that these design choices contribute synergistically to improved detection accuracy.

### D.3 DATA PRUNING AND SELECTION

To evaluate the practical effectiveness of TIMELAVA in identifying valuable time series segments, we conduct comprehensive data pruning and selection experiments across multiple public time series

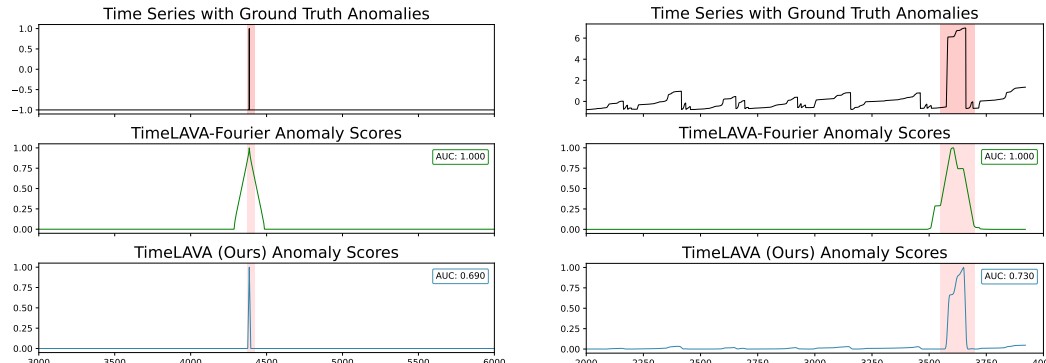

Figure 15: Comparison of TIMELAVA-Fourier and TIMELAVA on SMAP channel D-8 (left) and MSL channel F-5 (right). While TIMELAVA-Fourier achieves perfect AUC (1.000), this inflated score results from ground-truth labels that mark extended intervals rather than precise anomaly points. TIMELAVA provides more accurate point-wise localization despite lower AUC.

datasets. Our experiments compare TIMELAVA against several established baseline methods to show that it achieves the best or second-best performance in data quality assessment.

**Experimental Setup.** We conduct two complementary experiments to assess data quality identification capabilities. In data pruning experiments, we evaluate the ability to identify and remove low-quality segments by progressively eliminating the lowest-valued segments (0% to 50% removal) and measuring downstream forecasting performance, testing whether our method can effectively filter out detrimental data. Conversely, in data selection experiments, we assess data selection capability by retaining only the highest-valued segments (20% to 50% retention) and evaluating performance, measuring the method's ability to identify the most informative data points. For each scenario, we train linear AR models on the selected segments for multi-step forecasting and measure Root Mean Square Error (RMSE), and coefficient of determination ($R^2$). For experiments with injected noise, we assess each method's ability to identify corrupted segments using AUC-ROC, Precision@20%, and Recall@20%. We set $c = 0$ in Eq. equation 6 since these forecasting tasks focus purely on temporal pattern valuation. For all datasets, we compute valuations per dimension and report the average, restricting inputs to one dimension at a time to avoid excessive runtime for slow methods (e.g., TimeInf) and ensure comparability.

**Datasets.** We evaluate our approach on four widely-used public time series datasets (Wu & Keogh, 2021; Liu et al., 2024).

- **ETT (Electricity Transformer Temperature)**: 2-year hourly data from two counties in China, featuring oil temperature as the target variable and 6 power load features, with the dataset split into 12/4/4 months for train/validation/test.

- **Traffic**: hourly road occupancy rates from 862 sensors across California over 48 months, providing a high-dimensional multivariate time series with complex spatiotemporal patterns.

- **Electricity**: hourly electricity consumption patterns of 321 clients from 2012-2014, representing diverse consumer behavior patterns in energy usage.

- **Exchange**: daily exchange rates for eight countries from 1990-2016, capturing long-term economic trends and currency fluctuations.

For the Electricity, Traffic, and Exchange datasets, we adopt a 70%/10%/20% split for training/validation/testing. We partition each dataset into non-overlapping segments of fixed length, with segment lengths chosen based on dataset characteristics: 96 time steps for ETT and Electricity, 168 for Traffic, and 30 for Exchange. This segmentation allows us to evaluate data quality at a meaningful temporal granularity while maintaining sufficient statistical power for downstream tasks. To test robustness and noise detection capabilities, we systematically corrupt 20% of training segments with four types of realistic time series noise: Gaussian noise (additive white noise scaled by segment standard deviation), spike artifacts (random impulse noise at multiple time points), linear

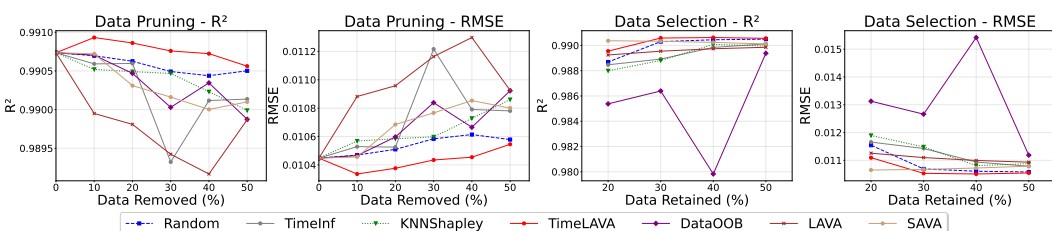

Figure 16: Data Selection and Pruning Results - Exchange

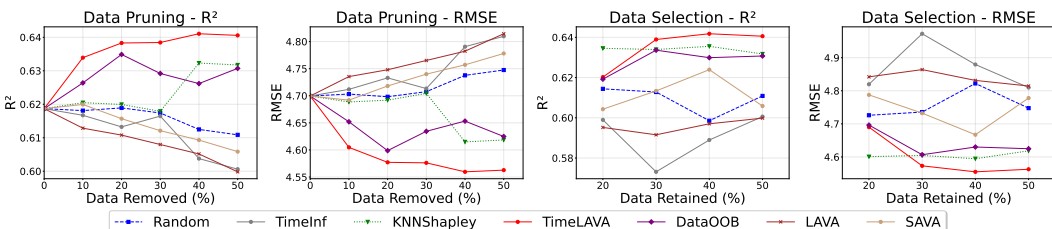

Figure 17: Data Selection and Pruning Results - ETTh1

drift (systematic trend distortion across the segment), and scale corruption (multiplicative scaling by random factors).

**Results.** Figs. 16-19 demonstrate TimeLAVA's comparable performance across all datasets.

*Data selection.* In data selection experiments, TimeLAVA consistently achieves the highest $R^2$ values when retaining high-value segments, with particularly strong performance on the Exchange dataset where it maintains high $R^2$ even when using only 30% of the data. The Traffic dataset results highlight TimeLAVA's robustness to high-dimensional, complex temporal patterns, where while baseline methods show significant performance degradation as data is reduced, TimeLAVA maintains stable forecasting accuracy across different retention rates, suggesting that our wavelet-based feature extraction successfully captures essential temporal structures that generalize well to forecasting tasks.

*Data Pruning.* The data pruning results indicate that TimeLAVA effectively removes redundant and low-quality segments without compromising model performance. Fig. 20 presents ROC curves for noise detection across all datasets, where TimeLAVA achieves consistently high AUC scores across different datasets, significantly outperforming baseline methods. The Influence Function method shows competitive performance on some datasets but lacks consistency, while KNN-Shapley and Data-OOB show moderate detection capabilities.

Across all datasets and experimental conditions, TimeLAVA shows consistent performance advantages, maintaining strong performance across diverse temporal patterns, dataset sizes, and noise conditions, suggesting that our approach captures fundamental aspects of time series data quality that generalize across domains.

D.4 LABEL NOISE

Temporal label noise is a critical challenge in real-world detection systems. Unlike static classification tasks, detection applications often exhibit time-varying labeling quality due to operational factors such as annotator fatigue, sensor drift, or changing environmental conditions. For instance, healthcare monitoring systems show systematic temporal variations: continuous glucose monitors experience calibration drift affecting detection accuracy, while human annotators exhibit time-of-day effects with reduced labeling quality during night shifts (Carpenter, 2003; Hsieh & Kocielnik, 2016; Nagaraj et al., 2025). Traditional approaches assuming static noise distributions fail to capture these temporal dynamics, leading to degraded performance. By explicitly modeling time-varying label quality, our experimental framework enables rigorous evaluation of methods designed to handle temporal label noise versus conventional approaches.

**Experimental Setup.** We consider a controlled semi-synthetic experimental framework that treats original dataset labels as ground truth and systematically injects temporal noise patterns to evaluate

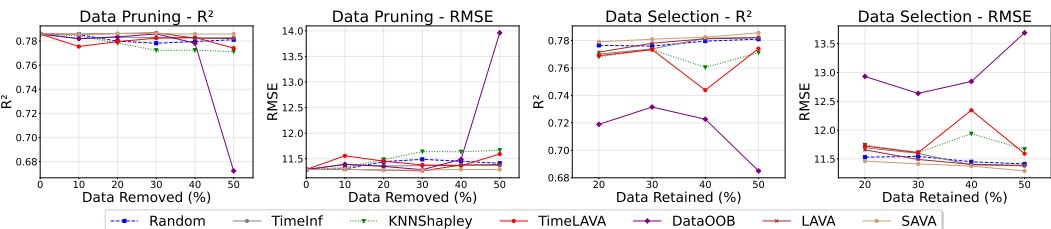

Figure 18: Data Selection and Pruning Results - Traffic

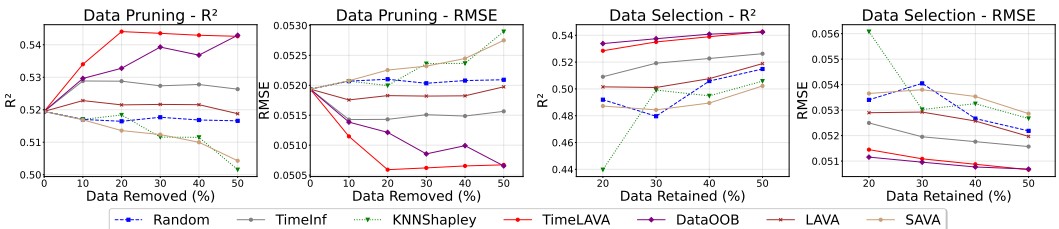

Figure 19: Data Selection and Pruning Results - Electricity

detection robustness. This approach follows established benchmarking practices for label noise research (Jiang et al., 2023; Just et al., 2023; Kwon & Zou, 2023) while extending them to capture temporal dynamics essential for detection applications. Since time series classification typically operates on segments rather than individual points, we inject noise at the segment level. After dividing the time series into non-overlapping segments, we corrupt segment labels according to time-dependent probability functions that reflect realistic error patterns observed in operational systems. For each temporal noise pattern, corrupted segments have their labels flipped to the opposite class, simulating common failure modes such as misclassification errors:

$$\tilde{y}_i = \begin{cases} 1 - y_i & \text{if segment } i \in \mathcal{N} \\ y_i & \text{otherwise} \end{cases}$$

where $\mathcal{N}$ denotes the set of segments selected for corruption according to each temporal pattern, and $\tilde{y}_i$ represents the corrupted label for segment $i$. We evaluate four distinct temporal noise patterns (Random, Periodic, Decay, and Growth) that determine when corruption occurs, as detailed below.

Each dataset is partitioned temporally with 70% allocated for training and 30% for validation, ensuring realistic evaluation conditions where future data remains unseen during model development. Temporal noise is injected exclusively into training labels, while validation labels remain clean to provide unbiased performance assessment. This protocol simulates realistic deployment scenarios where detection systems must learn from noisy historical data while being evaluated on clean test conditions. We vary the average noise rate from 5% to 20% to evaluate across different noise severities

**Baselines.** We compare our proposed methods against the same baseline approaches used in our pruning experiments, ensuring consistent evaluation across different aspects of our work.

**Datasets.** We evaluate our temporal detection methods using two real-world datasets from healthcare applications following (Nagaraj et al., 2025). While these datasets were originally designed for standard classification tasks, we adapt them to study temporal label noise by treating the original labels as ground truth and systematically injecting temporal noise patterns. Each dataset represents binary classification tasks over complex temporal feature spaces that are well-suited for evaluating robustness to time-varying label corruption. The datasets include:

- **Moving**: human activity recognition task where we detect movement states (e.g., walking vs. sitting) using temporal accelerometer data in adults (Reyes-Ortiz et al., 2013).

- **Senior**: similar human activity recognition task as above but in senior citizens (Logacjov & Ustad, 2023).

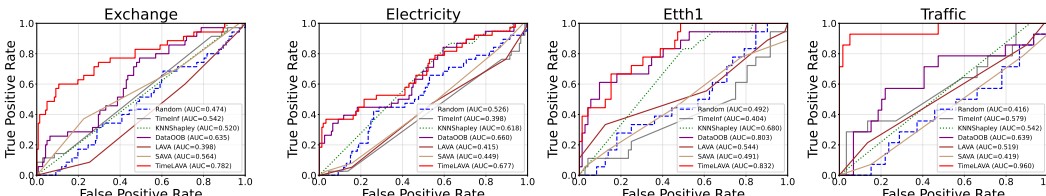

Figure 20: **ROC curves** illustrating the performance of temporal label-noise segment identification on four datasets: **Exchange**, **Electricity**, **ETTh1**, and **Traffic**. The results demonstrate the robustness of TIMELAVA in distinguishing clean from corrupted segments across diverse temporal domains.

- **Blinking**: eye movement (open vs closed) detection task using continuous EEG data (Roesler, 2013).

These classification tasks provide suitable testbeds for temporal label noise evaluation because they involve sequential labeling over time where temporal corruption patterns can realistically occur in practical deployment scenarios.

**Temporal Noise Injection.** Our noise injection mechanism corrupts detection labels according to time-dependent probability functions that reflect realistic error patterns observed in operational detection systems. We evaluate four distinct temporal noise patterns that capture diverse real-world detection error scenarios:

- **Random Noise** provides a baseline with uniform corruption probability: $P(i \in \mathcal{N}) = \eta$ for all segments $i$, representing temporally independent errors.

- **Periodic Noise** models cyclical error patterns: $P(i \in \mathcal{N}) = \frac{\eta}{2} + \frac{\eta}{2}\sin(\omega t_i + \phi)$ where $t_i$ denote the temporal position of segment $i$, $\omega$ controls frequency and $\phi$ is the phase offset, capturing systems affected by daily cycles or periodic maintenance.

- **Decay Noise** represents exponentially decreasing error rates: $P(i \in \mathcal{N}) = \eta \cdot \exp(-\lambda t_i/T)/Z$ where $\lambda$ controls decay rate and $Z$ is a normalization constant ensuring overall noise rate $\eta$. This models systems improving through learning or calibration.

- **Growth Noise** captures deteriorating performance: $P(i \in \mathcal{N}) = \eta/(1 + \exp(-\beta(t_i - T/2)))/Z$ where $\beta$ controls growth rate, $T/2$ is the midpoint, and $Z$ normalizes to achieve target rate $\eta$. This reflects aging or drift effects.

As shown in Fig. 21, these temporal noise patterns create distinct corruption patterns in the actual label sequences, with corrupted labels distributed according to their respective temporal probability functions.

**Evaluation.** We evaluate noise detection capability using F1-score metrics to assess how effectively each method identifies temporally corrupted detection labels.

**Results.** As shown in Figs. 22–24, TIMELAVA achieved the best performance in periodic noise detection, substantially outperforming all baseline methods, while demonstrating competitive performance on other noise patterns. KNN-Shapley performs well as it assumes that segments with similar features should have similar labels, making it effective at detecting mislabeled segments that disagree with their nearest neighbors in the feature space when label noise percentage is large. TIMELAVA's advantage stems from its wavelet-based features that capture multi-scale temporal patterns within each segment, making it more sensitive to systematic labeling errors. Unlike methods that treat time series segments as independent feature vectors, TIMELAVA preserves the sequential nature of temporal data through its sliding window approach and distributional alignment, making it effective for real-world scenarios where labeling errors follow temporal patterns such as periodic system failures, time-dependent annotation quality degradation, or systematic errors that evolve over time.

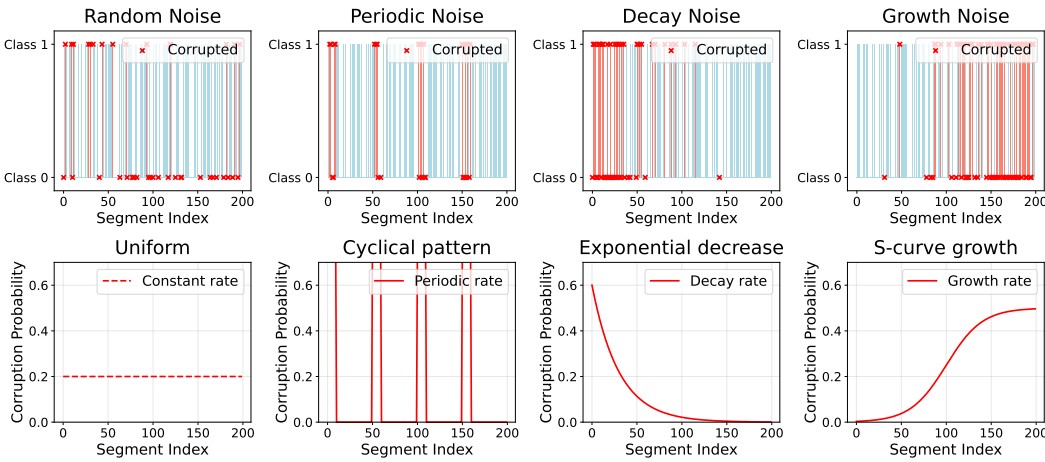

Figure 21: **Temporal label noise patterns in time series segments.** Upper: Binary classification labels (Class 0/1) with corrupted segments marked in red. Lower: Time-varying corruption probability $P(i \in \mathcal{N})$ for each pattern. Random noise maintains constant probability $\eta$, Periodic follows sinusoidal variation simulating cyclical quality changes, Decay shows exponential decrease modeling learning effects, and Growth exhibits sigmoid increase representing degradation over time.

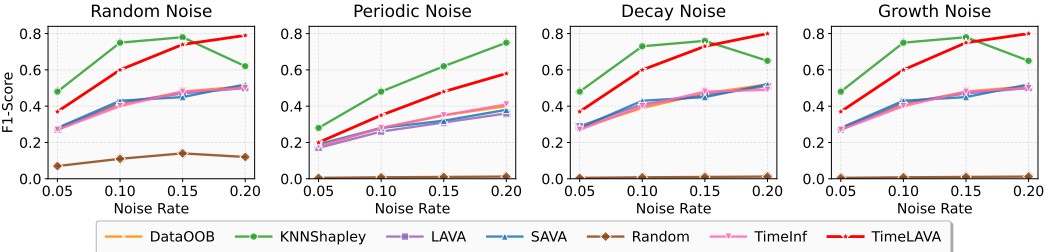

Figure 22: F1-score on **Blinking** across different noise types.

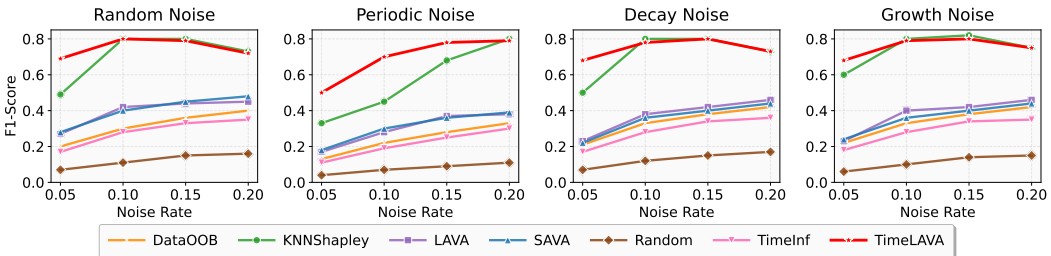

Figure 23: F1-score on **Moving** across different noise types.

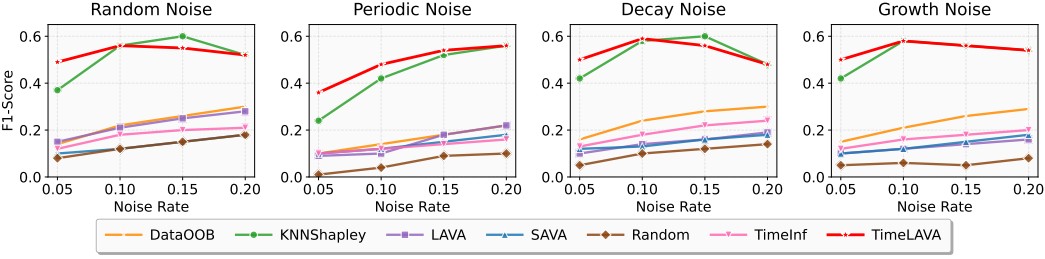

Figure 24: F1-score on **Senior** across different noise types.

**Parameter Sensitivity Analysis.** We conduct sensitivity analyses for the two core hyperparameters in TIMELAVA: the UOT regularization strength $\kappa$ and the label balance parameter $c$. Results are shown in Figure 25.

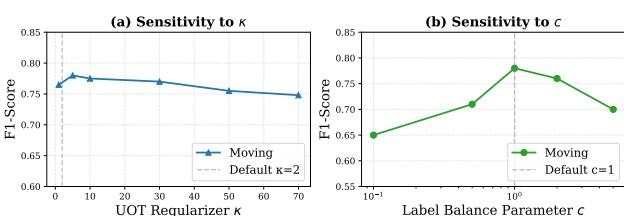

Figure 25: Sensitivity to hyperparameters $\kappa$ and $c$ on Moving dataset.

**UOT Regularizer $\kappa$.** Figure 25 (a) illustrates the effect of the UOT regularization strength $\kappa$ on label noise detection performance using the Moving dataset with 15% periodic noise (fixing $c$=1). TIMELAVA demonstrates remarkable stability for $\kappa$ between 1 and 10, where the F1-score varies by less than 1.3% (0.765–0.775); our default choice $\kappa$=2 (marked by the vertical dashed line) lies well within this flat and reliable region. As $\kappa$ increases beyond 30, the formulation begins to resemble balanced optimal transport, which enforces strict mass-conserving matches between segments. This constraint reduces the method's ability to selectively downweight unmatched or outlier segments, leading to a more pronounced decline in performance (F1-score drops to 0.700 at $\kappa$=70). Overall, the trend demonstrates that TIMELAVA is highly robust to $\kappa$ over a wide operating range, with meaningful degradation only at extreme values.

**Label Balance Parameter $c$.** Figure 25 (b) shows the effect of the label balance parameter $c$ on the same label noise detection task (fixing $\kappa$=2). The performance exhibits a clear inverted-U pattern, peaking at $c$=1 (F1-score of 0.78). When $c$<1, the model underweights label information and behaves closer to a purely unsupervised temporal pattern matcher at $c$=0.1, performance drops to 0.65 as the method largely ignores the informative label mismatch signal. Conversely, when $c$>1, the matcher increasingly overemphasizes label alignment at the expense of temporal feature similarity. At $c$=5, performance degrades to 0.70 as the method becomes less robust to noisy labels by forcing alignment based on potentially corrupted label information. The stability around $c\in[0.5, 2.0]$ (F1-score ranging from 0.71 to 0.78) confirms that TIMELAVA achieves a robust balance between temporal pattern similarity and label consistency, validating our default choice of $c$=1.

