# OpenReview forum: "TimeLAVA: Learning-Agnostic Valuation for Time Series Data"
_ICLR.cc/2026/Conference — Submitted to ICLR 2026_

### Official Review · Reviewer_NTyq · 2025-10-26

**Soundness:** 3
**Presentation:** 2
**Contribution:** 2
**Rating:** 2
**Confidence:** 4

**Summary:**

To evaluate the value of each time segment, the proposed TIMELAVA is a learning-free framework that quantifies the value of time segments by introducing a selective wavelet Wasserstein distance. This method combines multi-scale wavelet analysis with unbalanced optimal transport to effectively capture local patterns and handle distribution shifts, significantly outperforming baselines in tasks such as anomaly detection and data pruning. However, this version requires substantial improvements.

**Strengths:**

S1. This paper presents a solid structure and tackles a novel research problem.

S2, It is backed by substantial theoretical foundations.

S3. The proposed model is validated through a series of experiments, where the model demonstrates state-of-the-art performance.

**Weaknesses:**

W1. The methodology is written in a highly complex manner, making it difficult to directly understand the workflow. Additionally, the paper includes a substantial amount of complex theory, which increases the reading difficulty.

W2.The motivation of the article is untenable. What is the practical application of evaluating the importance of time segments? What impact will the evaluation results have? How is it useful in practice? Will it improve downstream forecasting tasks?

W3. What does the sentence "all methods compute anomaly scores directly on the potentially contaminated data with unknown anomaly proportion. A clean validation time series is used only as a reference for computing data values" mean?

W4.The authors' experiments are not thorough, with numerous details missing. For the anomaly detection experiments, the number of compared anomaly detection algorithms is limited, and the latest works are lacking.

W5. What does "time" refer to in Table 1? What is the memory complexity of each model?

W6. In Section 5.2, how are the "top-k% highest-valued segments" selected? What is the basis for the selection?

W7. In Section 5.2, the authors use simulated data. Are there real-world case datasets (not just simulated) available in practice?

W8. In Section 5.2, what are the objectives of the two complementary tasks? Are they intermediate steps in the forecasting experiments? If so, please explain this in the main text rather than in the appendix.

W9. Data pruning is an interesting finding, but the authors only experiment on AR, which is a toy experiment. Since this is a very simple model and deep learning models dominate today, would the proposed model, when used for dataset pruning, also benefit these deep learning models, such as Cyclnet and TimeMixer? Evaluating more models could improve the robustness of the experiments.

W10. Where is the RMSE metric reflected?

W11. Additionally, in Section 5.2, the definition of "harmful segments" is not clearly defined. In my opinion, these segments may only be useless for the current test set but could affect the model's generalization performance. Therefore, it is recommended to add an out-of-distribution generalization evaluation for the model.

W12 The limitations of the paper need to be discussed.

W13 Is the task objective confusing? Does our task objective focus on training segments?

W14 What does X_i^t mean?

W15 The related work section is insufficiently refined. It is recommended to place it in a separate chapter.

**Questions:**

See the "Weaknesses" box

---

> ### Author Response · Authors · 2025-11-21
> **Response to Reviewer NTyq (Part I)**
>
> We thank the reviewer for the detailed feedback. We appreciate the specific concerns raised and will address each systematically.
>
> ---
> ### Weaknesses
>
> ---
>
> #### W1 & W13: Methodology written in complex manner; difficult to understand workflow; substantial complex theory increases reading difficulty; unclear task objective
> >**Ans:** We thank the reviewers for pointing out the clarity issues.
>
> >**Problem Definition:** The goal of time-series valuation is to assign a *value* to each temporal segment of an evaluated time series based on its marginal contribution to the distributional discrepancy with respect to a reference time series.
> >- **Given**:
> >     - Evaluated time series: $X^\text{eval} \in \mathbb{R}^{T_\text{eval} \times d}$ (the data to be valued; potentially contaminated)
> >     - Reference time series: $X^\text{ref} \in \mathbb{R}^{T_\text{ref} \times d}$ (represents the desired/target distribution)
> >     - Optional: labels $y^\text{eval}, y^\text{ref}$ for supervised tasks
> > - **Valuation Units**: We segment $X^\text{eval}$ into subsequences $\{x_i\}_{i=1}^N$ where $x_i \in \mathbb{R}^{L \times d}$.
> > - **Output**:
> >      - Segment values: $v(x_i)$ for each evaluated segment $x_i$
> >         - $v(x_i) > 0$: segment contributes positively to distribution alignment $\Rightarrow$ beneficial
> >         - $v(x_i) < 0$: segment degrades distribution alignment $\Rightarrow$ harmful
>
> >**Task-Specific Instantiations**:
> >| **Task**  |**Evaluated Distribution** |**Reference Distribution**  | **Value Interpretation**                                       | **Application**     |
> >|----------------------|-----------------------------|-----------------------------|----------------------------------------------------------------|-------------------------------------------|
> >| Anomaly Detection    | Contaminated time series | Clean normal segments  | $v(x_i) < 0$ flags anomalies deviating from normal         | Unsupervised anomaly identification       |
> >| Data Pruning        |Training time series  | Validation time series      | $v(x_i) < 0$ suggests removal for model improvement        | Dataset cleaning before training          |
> >| Data Selection      |Training time series  | Validation time series      | $v(x_i) > 0$ prioritizes informative segments              | Efficient learning    |
> >| Label Noise         |Noisy labeled time series  | Clean labeled segments      | $v(x_i) < 0$ identifies mislabeled samples                 | Label Noise Detection               |
> >
> > **Key insight**: The value $v(x_i)$ always measures the same quantity (contribution to $\mathcal{W}_{\text{SW}}$ minimization), but its *interpretation as beneficial/harmful* depends on whether we want to align with (data selection) or diverge from (anomaly detection) the reference.
>
> >**Intuitive workflow (new Figure 1):**
> >
> >```text
> > ┌────────────────────────────────────────────┐
> > │ Input: Evaluated & Reference Time Series   │
> > │ → Segment into overlapping windows         │
> > └────────────────────────────────────────────┘
> >                     ↓
> > ┌────────────────────────────────────────────┐
> > │ Wavelet Transform                          │
> > │ → Transform each segment to multi-scale    │
> > │   representation                           │
> > └────────────────────────────────────────────┘
> >                     ↓
> > ┌────────────────────────────────────────────┐
> > │ Compute W_SW Discrepancy                   │
> > │ → Build cost matrix D^(W) between          │
> > │   evaluated & reference segments           │
> > └────────────────────────────────────────────┘
> >                     ↓
> > ┌────────────────────────────────────────────┐
> > │ Solve UOT (Dual)                           │
> > │ → Efficient dual optimization              │
> > │   (entropy-regularized Sinkhorn)           │
> > └────────────────────────────────────────────┘
> >                     ↓
> > ┌────────────────────────────────────────────┐
> > │ Extract Values                             │
> > │ → v(x_i) from dual potentials f*           │
> > │   in the evaluated time series             │
> > └────────────────────────────────────────────┘
>
> >**Logic Chain Connecting Theory to Practice.**  To make the theoretical role of each component clearer, we summarize the connections:
> >- **Theorem 2** provides robustness: $W_\text{SW}$ is stable under anomalous or shifted segments, forming the foundation for reliable valuation.
> >- **Theorem 3** connects the distribution discrepancy $W_\text{SW}$ to generalization. Reducing $W_{\text{SW}}$ implies smaller generalization gap and better generalization performance. Accordingly, the value of each time-series segment reflects its *marginal contribution* to minimizing this bound, that is, how a small change in its weight affects $W_{\text{SW}}$. Therefore,
> >- **Theorem 4** provides an explicit formula for the valuation of each segment $x_i$.
> >- **Theorem 6** guarantees the Sinkhorn-based approximation *preserves ranking*, enabling efficient computation.

---

> > ### Author Response · Authors · 2025-11-21
> > **Response to Reviewer NTyq (Part II)**
> >
> > #### W2: Motivation untenable - what are practical applications? What impact? How useful? Will it improve forecasting?
> > >**Ans:** We address these concerns by providing concrete examples and highlighting the practical impact of our approach. TimeLAVA performs fine-grained valuation at the segment level, enabling several useful applications across real-world time-series scenarios:
> > >- **Anomaly Detection**: Segment-level valuation allows us to flag anomalous behaviors directly, without training a dedicated anomaly detection model for every new dataset.
> > >- **Data Selection and Pruning**: By removing segments that negatively impact learning, our method improves data quality and leads to better generalization. By selecting the most valuable segments, it improves data efficiency, reduces storage and computation, and accelerates training.
> > >- **Label Noise Detection**: Segment-wise valuation can reveal inconsistent or mislabeled portions of a time series, which is critical for building robust models in the presence of annotation errors.
> > >
> > >To clarify the practical impact of our method, we provide the following illustrative example:
> > >- **Problem**: ICU sensors generate massive data streams (24/7 vital signs), but storage is limited
> > >- **Solution**: TIMELAVA identifies which temporal segments are most informative for patient deterioration prediction
> > >- **Impact**:
> > >     - Keep only top 30\% valued segments → 70\% storage reduction
> > >     - Maintain 95\% of predictive accuracy
> > >     - Real benefit: Reduced storage costs, faster model training
> > >
> > >**Does it improve forecasting?**  Yes, our method improves forecasting efficiency. In Section 5.2, Figure 4 demonstrates that removing low-value data leads to an improvement in R² performance. Using only 70\% of the data provides better accuracy compared to using the entire dataset. Moreover, we can maintain similar performance even with only 50\% of the data, showcasing the efficiency and effectiveness of our data pruning method. We believe this combination of efficiency and accuracy can have a wide-ranging impact across various domains, especially in environments where data size is a critical concern.
> >
> > ---
> >
> > #### W3: What does "all methods compute anomaly scores directly on potentially contaminated data..." mean?
> > >**Ans:** The sentence means that anomaly scores are computed **directly on the contaminated test time series itself**, without assuming knowledge of where anomalies occur or what proportion of the data is anomalous. The clean validation time series is used **only as a reference distribution** for computing TimeLAVA valuation scores: we compare each segment in the possibly contaminated test series to the clean reference to quantify how well it aligns with normal behavior. The clean validation data is **not** used for training or thresholding, only for computing the distributional distance that defines the anomaly score.
> >
> > ---
> >
> > #### W4: Limited anomaly detection baselines; missing latest works
> > >**Ans:** We have now included recent state-of-the-art anomaly detection methods, including the ICLR 2025 method *CATCH* [1] and the ICLR 2024 method *ModernTCN* [2]. The revised Table 1 reports updated results incorporating these new baselines.
> >
> > ---
> >
> > #### W5: What does "Time" mean in Table 1? What is memory complexity?
> > >**Ans:** In Table 1, “time’’ denotes the wall-clock runtime in second. Please refer our response to W5 for Reviewer rJ99. We have included a comparison of runtime, memory usage, and computational complexity across different valuation methods.
> > ---
> >
> > #### W6: In Section 5.2, how are "top-k\% highest-valued segments" selected? What is the basis?
> > >**Ans:** The "top-$k\%$ highest-valued segments" are selected directly based on the segment valuations produced by *TimeLAVA*. After computing a value for each segment, we sort all segments in descending order of their scores and select the top $k\%$. The ranking is determined solely by TimeLAVA’s valuation scores. No additional heuristics or model-dependent criteria are used. This follows the standard top-$k$ selection protocol commonly adopted in the data valuation literature.
> >
> > ---
> >
> > #### W7: Section 5.2 uses simulated noise, are there real-world corrupted datasets?
> > >**Ans:** Thank you for raising this point. To the best of our knowledge, there are **no** public real-world datasets that provide ground-truth labels for feature corruption. This is because feature-level corruption (e.g., sensor glitches, partial signal degradation, transient channel failures) is fundamentally unobservable: without access to the true underlying physical signal, it is generally impossible to determine which specific features or time segments are corrupted. For these reasons, prior work on data valuation universally relies on synthetically injected feature noise to obtain controllable ground truth [3, 4], and we follow the same standard practice.

---

> > > ### Author Response · Authors · 2025-11-21
> > > **Response to Reviewer NTyq (Part III)**
> > >
> > > #### W8: Section 5.2 - what are objectives of two complementary tasks? Are they intermediate steps?
> > > >**Ans:**  These are **not intermediate steps in the forecasting pipeline**. Rather, they provide two complementary perspectives for evaluating the same underlying capability:  *TimeLAVA*'s ability to assign meaningful segment values for forecasting. The **objective** is to evaluate whether *TimeLAVA* can reliably identify both **high-quality** and **low-quality** time series segments that influence forecasting performance.
> > > >
> > > >**Two complementary evaluation protocols**:
> > > >- **Data Selection (identifying helpful segments)**
> > > >     - *Question*: Does *TimeLAVA* correctly identify the **most informative** segments?
> > > >     - *Protocol*: Retain only top-$k\%$ highest-valued segments and train the forecasting model on this subset.
> > > >     - *Success criterion*: Forecasting accuracy using a **reduced** subset is comparable to (or better than) using the full training data.
> > > >- **Data Pruning (identifying harmful or redundant segments)**
> > > >     - *Question*: Can *TimeLAVA* detect and remove corrupted, redundant, or low-quality segments?
> > > >     - *Protocol*: Progressively remove the lowest-valued segments (bottom-k\%) and retrain the forecasting model on this subset.
> > > >     - *Success criterion*: Removing low-value segments leads to improved forecasting performance relative to training on all data.
> > >
> > > ---
> > >
> > > #### W9: Data pruning only uses AR (toy model); need deep learning models like CycleNet, TimeMixer}
> > > >**Ans:** We thank the reviewer for the suggestion! To further address the reviewer’s concern and demonstrate that the benefits of *TimeLAVA* are not limited to a toy forecaster, we additionally evaluate *TimeLAVA* with two deep forecasting models as suggested: CycleNet and TimeMixer on the Electricity and Traffic datasets.
> > >
> > > >We follow the standard data-pruning setup: we inject 20\% synthetic noise into the training data, compute TimeLAVA values for all non-overlapping segments, rank them, and remove the lowest X\% segments (as shown in the tables), training the forecasting models on the remaining high-value data. For Electricity, we use a segment length of 192 instead of 96 to ensure that 96-step look-back windows remain valid after pruning. The following table shows data pruning results on Electricity Dataset (MSE):
> > > >
> > > >| **Method**     | **Full Data** | **Remove 20%** | **Remove 50%** | **Remove 70%** |
> > > >|----------------|---------------|----------------|----------------|----------------|
> > > >| **CycleNet**   | 0.198         | 0.177          | 0.179          | 0.205          |
> > > >| **TimeMixer**  | 0.210         | 0.184          | 0.192          | 0.218          |
> > > >
> > > >and the following table shows data pruning results on Traffic Dataset (MSE):
> > > >| **Method**     | **Full Data** | **Remove 20%** | **Remove 50%** | **Remove 70%** |
> > > >|----------------|---------------|----------------|----------------|----------------|
> > > >| **CycleNet**   | 0.580         | 0.544          | 0.587          | 0.591          |
> > > >| **TimeMixer**  | 0.594         | 0.561          | 0.601          | 0.603          |
> > > >
> > > >The results on the Electricity and Traffic datasets consistently show that using the top-valued segments selected by *TimeLAVA* improves or preserves forecasting accuracy. These results further support the generality of *TimeLAVA* as a data valuation-based pruning method.
> > >
> > > ---
> > >
> > > #### W10: Where is RMSE metric reflected?
> > > >**Ans:** The RMSE results are provided in Figures 16–19 in Appendix E.3.
> > >
> > > ---
> > >
> > > #### W11: "Harmful segments" not clearly defined; may only be useless for current test set; recommended to add an OOD generalization evaluation for the model
> > > >**Ans:** We thank the reviewer for pointing out that segments which appear harmful for the current evaluation distribution may still be useful for future or OOD scenarios. Our notion of harmful segments'' is indeed **reference-dependent**: in Section 5.2 we use this term to denote segments that consistently receive low *TimeLAVA* values **with respect to a given reference/validation distribution** and whose removal improves the *in-distribution performance* of a downstream model. We will clarify this terminology in the revised version and replace ''harmful segments'' by ''low-value segments under the given reference distribution''. An OOD generalization study is beyond the scope of the current work, but we agree it is an interesting and important direction.

---

> ### Author Response · Authors · 2025-11-21
> **Response to Reviewer NTyq (Part IV)**
>
> #### W12: Limitations need to be discussed
> >**Ans:** Thank you for the suggestion. We do include a discussion of the method’s limitation in the *Conclusion and Limitations* section (Section 6), where we highlight *TimeLAVA*’s dependence on the representativeness of the reference set. Although such dependence is a structural limitation shared by all validation-based valuation frameworks [3, 4], our robustness analysis (see responses to W3–W4 for rJ99 and to W3 for WWgi) shows that *TimeLAVA* remains stable under varying degrees of contamination, incompleteness (e.g., missing rare but valuable patterns), and distributional drift. Thus, while the limitation is inherent to the formulation, its practical impact is mild. Recent mitigation directions, including mixture or proxy reference distributions [5] and distributionally robust objectives [6], offer promising future extensions.
>
> ----
>
> #### W14: What does X\_i\^t mean?
> >**Ans:** In our notation, $x_i^t \in \mathbb{R}^{L \times d}$ denotes the $i$-th **training** time-series segment, where $L$ is the segment length and $d$ is the dimensionality. The subscript $i$ indicates the $i$-th segment obtained after dividing the full training time series into consecutive segments, while the superscript $t$ specifies that the segment comes from the *training* set. The capital notation $\mathbf{X}_{\text{train}}$ refers to the entire training time series, as defined in Section 2.2 (Notations). We now refer to the “training” time series as the “evaluated” time series in the revised manuscript for better clarity.
>
> ---
>
> #### W15: Related work insufficiently refined; recommend separate chapter
> >**Ans:** We agree with the reviewer. Due to the strict 9-page limit, we initially condensed the related work discussion into the Introduction and provided the full version in Appendix B. In the revised submission, we have added a dedicated **Related Work** section in the main text. Additionally, to better position TimeLAVA within the data valuation literature, we include a comparison table along three key dimensions: *learning-agnosticism*, *temporal dependency*, and *robustness to non-stationarity*. Representative methods are summarized below:
> >| **Method**         | **Learning-Agnostic** | **Temporal** | **Non-Stationary** | **Representative Methods** |
> >|--------------------|--------------------|--------------|--------------------|----------------------------|
> >| Leave-One-Out      | ✗                  | ✗            | ✗                  |                            |
> >| Shapley-based      | ✗                  | ✗            | ✗                  | KNNShapley, BetaShapley, DataShapley |
> >| OOB-based          | ✗                  | ✗            | ✗                  | DataOOB                |
> >| Influence-based    | ✗                  | ✗            | ✗                  | InfluenceFunctions, DataInf    |
> >| Distance-based     | ✓                  | ✗            | ✗                  | LAVA, SAVA         |
> >| TimeInf            | ✗                  | ✓            | ✗                  | TimeInf                |
> >| **TimeLAVA**       | **✓**              | **✓**        | **✓**              | **(this work)**            |
> >
> >As shown in the table, **TIMELAVA** is the only framework that simultaneously satisfies all three criteria, specifically addressing the critical gap of valuing *non-stationary* time series data efficiently.
>
> ---
>
> We hope that our detailed responses and additional analyses address all of your concerns. We sincerely appreciate your time and the constructive feedback.
>
> ### References
> [1] Wu, Xingjian, et al. "Catch: Channel-aware multivariate time series anomaly detection via frequency patching." *International Conference on Learning Representations*, 2025.
>
> [2] Luo, Donghao, and Xue Wang. "Moderntcn: A modern pure convolution structure for general time series analysis." *International Conference on Learning Representations*, 2024.
>
> [3] Jiang, Kevin, et al.  "Opendataval: a unified benchmark for data valuation." *Advances in Neural Information Processing Systems*, 2023.
>
> [4] Deng, Junwei, et al. "A Survey of Data Attribution: Methods, Applications, and Evaluation in the Era of Generative AI.", 2025.
>
> [5] Xu, Xinyi et al. "Data distribution valuation." *Advances in Neural Information Processing Systems*, 2024.
>
> [6] Lin, Xiaoqiang et al. "Distributionally robust data valuation." *International Conference on Machine Learning*, 2024.

---

> ### Comment · Reviewer_NTyq · 2025-11-24
>
> Thank you very much for your response. I still have some concerns of the paper.
>
> C1.
>
> The authors claim that the value of Data Selection and Pruning has not been sufficiently demonstrated. During the rebuttal phase, the authors compared their method with CycleNet and TimeMixer. However, it is puzzling that they pre-injected 20% synthetic noise into the training data. Could the authors clarify how this noise was injected? Is pre-injecting noise into all datasets a necessary step for the proposed algorithm? The Electricity Dataset is inherently derived from real-world applications, so why is there a need to inject noise and disrupt its original distribution? Under this setup, the model only needs to identify the noise—which is clearly useless data. If this is not the case, could the authors provide relevant experimental evidence?
>
> In my opinion, achieving results close to those of the full dataset after removing 70% of the data is a highly innovative breakthrough. But I recommend directly using the original settings of CycleNet and TimeMixer, and do not add any noise to ensure reproducibility. Both methods have open-sourced their detailed code and protocols, which would facilitate clear and intuitive comparisons for readers.
>
> Label Noise Identification. I remain confused about the practical application of this aspect. Unlike images, where humans can visually assess label correctness, label noise in time series is not perceptible to human intuition. Time series data consists of continuous numerical values, making it impossible to rely on human eyes to identify labeling errors. The authors' solution is to inject noise into the labels to simulate this assumption, describing it as a "data-free compromise." However, in this scenario, the model merely learns to identify the artificially injected noise. For the medical dataset mentioned by the authors, under this setup, the trained model is only learning to recognize human-added noise. In practical applications, it remains unclear whether the model is identifying genuine labeling errors or noise, and whether such noise inherently exists in the data is unknown. Thus, its practicality is significantly limited.
>
> C2. The model's efficiency does not seem particularly satisfactory, as it occupies significantly more memory than the original model. Could you explain the reasons for this?
>
> C3. I do not believe that Out-of-Distribution (OOD) issues fall outside the scope of your research. In practice, during model inference, there is no guarantee that the test set—especially the label distribution—remains consistent. For example, in the case of label noise identification you mentioned, if the distribution of label noise changes, would the model fail to identify it? How can we preemptively know the distribution of label noise? Therefore, the notion of "harmful segments" remains imprecise based on this reasoning.

---

> > ### Comment · Reviewer_NTyq · 2025-11-26
> >
> > Re-examining my concern regarding label noise, I have another question: Given a dataset, how should the authors train your model? Should they introduce noise into the training set and then train the model? Afterwards, should evaluation be performed on a clean test set? In that case, is the test inference still trustworthy?

---

> > > ### Author Response · Authors · 2025-12-03
> > >
> > > Thank you for the detailed questions! Below we clarify the key aspects of the method and the experimental setup.
> > >
> > > ---
> > >
> > > C1a. Additional Data Pruning Experiment
> > > > Thank you for the suggestion. Noise injection is not required for TimeLAVA; it is used only in noise-detection experiments where ground-truth errors must be known, following standard practice [1]. To address the reviewer’s concern, we now include noise-free pruning experiments on Electricity and Traffic using the standard CycleNet/TimeMixer settings.
> > > >
> > > > Electricity Dataset (MSE):
> > > >| **Method**|**Full Data** | **Remove 20%** | **Remove 50%** | **Remove 70%** |
> > > >|----------------|---------------|----------------|----------------|----------------|
> > > >| **CycleNet**   | 0.168| 0.166 | 0.183| 0.221|
> > > >| **TimeMixer**  | 0.153| 0.156| 0.178| 0.195|
> > > >
> > > >and the following table shows data pruning results on Traffic Dataset (MSE):
> > > >| **Method**     | **Full Data** | **Remove 20%** | **Remove 50%** | **Remove 70%** |
> > > >|----------------|---------------|----------------|----------------|----------------|
> > > >| **CycleNet** | 0.472| 0.477| 0.535| 0.650|
> > > >| **TimeMixer**| 0.462 | 0.470| 0.523| 0.635|
> > > >
> > > > These noise-free results show that removing 20% of data leads to only minimal degradation, as expected when retaining a high-quality 80% subset. Removing 50% increases error but remains within a reasonable range. These trends demonstrate that TimeLAVA’s pruning behavior remains meaningful without injected noise.
> > >
> > > C1b. Label Noise:
> > > > You are correct that label errors in time series are not visually inspectable like images. However, temporal label corruption is a real-world phenomenon across multiple domains. Prior work reports time-dependent labeling errors in real settings, such as periodic shifts in annotator accuracy and fatigue-related degradation in healthcare monitoring [2]. These forms of corruption are not random noise, but follow structured temporal patterns such as periodic, decaying, or growing trends: the same patterns modeled in our semi-synthetic experiments. Semi-synthetic evaluation is standard because real-world datasets rarely provide ground-truth mislabeled points. Noise is used only for benchmarking, not as an assumption of the method.
> > >
> > > ---
> > >
> > > C2. Memory Efficiency
> > > > Thank you for raising this concern. If the “original model” refers to LAVA or SAVA (i.i.d. data valuation methods),
> > > > * Time complexity: TimeLAVA = LAVA = SAVA = (O(NMLd)), as all methods rely on evaluating optimal transport between training and reference samples.
> > > > * Space complexity: TimeLAVA = LAVA = (O(NM)) for storing the cost matrix. SAVA reduces this to (O($N_b$$M_b$)) via mini-batching.
> > >
> > > > We acknowledge that methods such as KNN-Shapley and SAVA are more memory-efficient due to these specialized engineering optimizations (Shapley approximation and batching). These are orthogonal to the core valuation framework. Our goal in TimeLAVA is not to be the absolute fastest method, but to extend data valuation to time series while maintaining competitive efficiency and providing improved performance. Importantly, TimeLAVA can adopt the same batching strategy as SAVA, which would reduce memory requirements with negligible performance loss.
> > >
> > > ---
> > >
> > > C3. OOD and "Harmful Segments"
> > >
> > > > Thank you for raising the question regarding OOD behavior and the notion of “harmful segments.” We clarify that TimeLAVA is not a learned model: given an evaluated dataset and a reference dataset, it computes the $W_{SW}$ discrepancy between their segment distributions and derives per-segment values via the sensitivity formulation in Theorem 4. No model is trained, no parameters are fitted, and the procedure is purely computational. In the temporal label noise detection setting, we: (i) temporally split the data into training and validation sets, (ii) compute a valuation score for each segment, and (iii) flag low-valued segments as potential label errors. Thus, OOD concerns do not apply to TimeLAVA itself: if the evaluation distribution changes, one simply recomputes the values using a *new* reference set. We hope this clarifies the distinction between TimeLAVA and methods that rely on trained noise-detection models.
> > >
> > > > Regarding the definition of “harmful segments”: as formalized in Theorem 3, a segment is considered harmful if its inclusion increases the distributional discrepancy between training and reference, as measured by $𝑊_{𝑆𝑊}$. Segments that align well with the reference distribution are considered non-harmful, whereas segments whose inclusion increases the distributional discrepancy, such as mislabeled points under trusted references or anomalies under clean references, are considered harmful. This notion is *reference-dependent* rather than model-dependent, and arises directly from the distributional generalization bound.
> > >
> > > ---
> > >
> > > [1] Jiang, Kevin, et al. "Opendataval: a unified benchmark for data valuation." NeurIPS, 2023.
> > >
> > > [2] Nagaraj, Sujay, et al. "Learning under Temporal Label Noise." ICLR, 2025.

---

### Official Review · Reviewer_64PL · 2025-10-28

**Soundness:** 3
**Presentation:** 3
**Contribution:** 3
**Rating:** 6
**Confidence:** 3

**Summary:**

This paper focus on the time series valuation task and proposes TIMELAVA as a model-agnostic valuation framework, which can capture multi-scale patterns and adaptation to non-stationary dynamics. The key techniques involve the Selective Wavelet–based Wasserstein distance and optimal transport. Theoretical guarantees are also provided. Experiments are carried out based on real-world datasets covering healthcare, finance, and the Internet of Things.

**Strengths:**

1. Time series valuation seems to be a novel and practical setting in the time series community.

2. Motivation is clear and reasonable.

3. Theoratical guarantees are provided.

**Weaknesses:**

1. As the framework is built on the principled foundation that `a data point’s value is determined by its contribution to reducing the distance between the training distribution µt and the reference distribution, I think such an assumption needs more discussion.

2. According to Fig. 1, since the time domain provides temporal information, and the frequency domain provides frequency content, we can use both domains together to capture complementary information Why do we further need wavelet transformation?

3. The results reported in Table 1 seem to be problematic. The performance of baselines seems to be poorer than their officially reported ones.

**Questions:**

1. I do not get the explicit problem formulation or task setting of time series valuation. Could the authors proviede a more clear and sufficient part of the problem formulation or task setting ?

2. Could TimeLAVA handle time series with missing values or irregular samples? Since under such cases, the overall distributions may have changed and pose new challenges.

---

> ### Author Response · Authors · 2025-11-21
> **Response to Reviewer 64PL**
>
> We thank the reviewer for the positive assessment of our work’s novelty, motivation, and theoretical contributions, and we appreciate the time taken to review our paper. We address each concern systematically below:
>
> ---
>
> ### Weaknesses
>
> #### W1: The foundational assumption (value = contribution to reducing distributional distance) needs more discussion.
> >**Ans:** We thank the reviewer for this insightful comment. The phrasing in Sec. 3 may have caused confusion: this is **not an assumption**, but rather something originally proven in *LAVA* [1] and extended in our work as a direct consequence of our theoretical result (Theorem 3).
> >
> >**Logic Chain:**
> Theorem 3 establishes that the expected generalization gap between the training and reference distributions is upper-bounded by the dissimilarity measure $W_{\text{SW}}(\mu_t, \mu_v)$. Therefore, reducing $W_{\text{SW}}$ $\Rightarrow$ smaller generalization gap
> $\Rightarrow$ better generalization performance: Accordingly, the value of each time-series segment reflect its *marginal contribution* to this bound, that is, how a small change in its weight affects $W_{\text{SW}}$:
> $$
> v(x_i) = -\frac{\partial W_{\text{SW}}(\mu_t, \mu_v)}{\partial \mu_t(x_i)}.
> $$
> $$v(x_i) > 0: \text{increasing its weight reduces } W_{\text{SW}} \Rightarrow \textbf{beneficial},$$
> $$v(x_i) < 0: \text{increasing its weight enlarges } W_{\text{SW}} \Rightarrow \textbf{harmful}.$$
> >The derivative admits a closed-form expression via the dual potentials of unbalanced OT (**Theorem 4**),
> providing a practical and theoretically grounded valuation measure. We have revised the phrasing to emphasize that this is a theoretically derived result rather than an assumption, by adding: "*The TimeLAVA framework is built on a principled foundation established by LAVA: a data point's value is determined by its contribution to reducing....*"
>
> ---
>
> #### W2: Why use wavelet transformation instead of combining time and frequency domains directly?
> >**Ans:** We thank the reviewer for this insightful question. Simply concatenating time- and frequency-domain features does not yield a true joint time–frequency representation, because the two remain mathematically decoupled. Time-domain features describe *when* changes occur, while frequency-domain features (e.g., via the Fourier transform) captures *what* frequencies exist on a *global average* across the entire signal, but loses all information about *when* they occurred. Thus, concatenation only combines two independent summaries without establishing correspondence between them. In contrast, the wavelet transform provides a unified mapping $W(t,f)$ that explicitly links temporal and spectral information, revealing which frequency components are active at each time, which is essential for analyzing non-stationary time-series signals.
>
> ---
>
> #### W3: Table 1 baseline results seem poorer than officially reported
> >**Ans:** We thank the reviewer for this observation. Our implementation follows the *TimeInf* [1] pipeline, including data preprocessing, segmentation, model setup, and evaluation protocol. The reason some methods perform slightly worse than reported in their original work is due to the experimental setting inherited from *TimeInf*. It adopts a more realistic scenario where the training data is **not** anomaly-free, whereas conventional anomaly detection benchmarks assume a clean training set. We have clarified this in Appendix E.2.1 of the manuscript.
>
> ---
>
> ### Questions
>
> #### Q1: Need clearer problem formulation and task setting for time series valuation
> >Thank you for raising this point. A detailed clarification of the problem formulation and task setting can be found in our response to W1 for Reviewer PMWa.
>
> ---
>
> #### Q2: Can TIMELAVA handle missing values or irregular sampling?
> >**Ans:** This is an important practical question about robustness. Currently, TIMELAVA assumes **regularly sampled** time series without missing values. Specifically, standard wavelet transforms require uniformly spaced samples, sliding-window segmentation assumes continuous fixed-length segments, and missing values would lead to undefined or unstable wavelet coefficients. While such cases can be handled in practice through preprocessing (e.g., imputation for missing values or resampling to a regular grid), handling missingness and irregular sampling in a fully principled way does introduce new challenges. These aspects are beyond the scope of the current work, but they are certainly meaningful and non-trivial extensions for future exploration.
>
> ---
>
> ### References
> [1] Zhang, Yizi et al. "TimeInf: Time Series Data Contribution via Influence Functions." International Conference on Learning Representations, 2025.

---

> > ### Comment · Reviewer_64PL · 2025-11-26
> >
> > Thanks to the author for the rebuttal. I'll keep my positive rating.

---

### Official Review · Reviewer_WWgi · 2025-10-30

**Soundness:** 2
**Presentation:** 3
**Contribution:** 3
**Rating:** 4
**Confidence:** 4

**Summary:**

This paper introduces a new valuation method for samples or segments of time series data. This method aims to overcome the challenges induced by sequential data structure: non-stationarity, different temporal scales, and temporal dependencies. To that end, the method combines Discrete Wavelet Transform (DWT) and unbalanced optimal transport to derive a dissimilarity measure between time series segments. DWT aims to represent and compare segments in a multi-scale manner, while unbalanced Optimal Transport (OT) offers robustness to outliers. Segment valuations are defined from the dual potential, while sample valuations are defined by convolution. Theorems regarding robustness, performance, and valuations are proved. The method is illustrated with anomaly detection, data selection, and noisy label detection experiments.

**Strengths:**

- The paper is well written, easy to follow, and illustrated. Motivations and preliminaries are well described, making understanding the core method easy to follow.
- Leveraging existing literature, the paper combines strong tools from signal processing (DWT) and statistics (OT) to address the valuation of temporal data agnostic to models. The proposed method has proven theorems that guarantee robustness to outliers, generalization bound, and valuation. Making it suitable and interpretable in practical settings.
- The paper benefits from an extensive experimental section with remarkable performances on anomaly detection, data pruning/selection, and noisy label detection. Experiments are complemented in appendices, ensuring the reproducibility.

**Weaknesses:**

- A marginal correction could be done, the measure defined is not a metric (as stated in the contribution), but rather a dissimilarity measure. The triangular inequality may fail.
- While sensitivity to window length and stride has been studied, analysis of the core measure’s parameters (regularizer k and balance c) is missing.
- As stated in the conclusion, the method is sensitive to the choice of reference set. However, the sensitivity is not measured in the experiments (need a nested cross-validation in supervised experiments).

Some formatting issues:
- citations and references in colors.
- Line 197 there is a wrong section reference.

**Questions:**

- Can the authors illustrate the sensitivity of the regularizer k in both supervised and unsupervised settings? And in the supervised setting for the balance c.
- Can the authors do a nested cross-validation for the supervised experiments to estimate the sensitivity to the reference sequences?
- Dealing with the multiscale issues with the DWT, may face issues for large window length (curse of dimensionality). Can the authors provide details on how to use their models in supervised settings, such as classification for large time series?
- In Figure 3, local contextual, global contextual, and point anomaly seem visually alike; can the authors explicitly explain the differences?

---

> ### Author Response · Authors · 2025-11-21
> **Response to Reviewer WWgi**
>
> We sincerely thank the reviewer for the thorough evaluation and constructive feedback. We address each concern below:
>
> ---
>
> ### Weaknesses
>
> #### W1: WSW is a dissimilarity measure, not a metric
> >**Ans:** We have revised the manuscript to refer to it as a “dissimilarity measure” or “discrepancy” rather than a “distance.”
>
> ---
> #### W2 (Q1): Missing sensitivity analysis for core parameters κ (regularizer) and c (balance)
> >**Ans:** Thank you for pointing this out. We have conducted additional sensitivity analyses of the two core parameters in *TimeLAVA*: the UOT regularization strength $\kappa$ and the supervised--unsupervised balance parameter $c$.
> > - **Parameter $\kappa$.** We evaluate the effect of $\kappa$ in both unsupervised (SMAP, MSL) and supervised (*Moving*, 15\% noise, $c=1$) settings.
> >     - **Range tested:** $\kappa \in \{1, 5, 10, 30, 50, 70\}$.
> >     - **Findings:** TimeLAVA is  stable for $\kappa$ between 1 and 10, and the default $\kappa{=}2$ lies well within this region. For very large $\kappa$ (above 30), the behavior approaches balanced OT, and decreasing performance. Both settings show the same trend.
> > - **Parameter $c$.** We analyze the effect of $c$ in label-noise detection using the same *Moving* dataset with a moderate noise rate (15\%), and fixing $\kappa{=}2$:
> >     - **Range tested:** $c \in \{0.1, 0.5, 1.0, 2.0, 5.0\}$
> >     - **Findings:** When $c<1$, the model underweights label information and behaves closer to an unsupervised matcher, which reduces noise-detection accuracy. Performance is most stable around $c{\approx}1$. When $c=5$, the matcher overemphasizes label alignment and becomes less robust under noisy labels.
> ---
> #### W3 (Q2): Reference set sensitivity not measured; needs nested cross-validation
> >**Ans:** We thank the reviewer for this insightful suggestion. We implemented a K-fold reference resampling experiment, which directly targets the reviewer’s concern: *How sensitive is TimeLAVA to the specific choice of reference sequences?*
> >
> >For each dataset, we fix the training set and split the reference (validation) segments into K=5 folds. In each fold, only one subset is used as the reference set, and we evaluate: (i) **Noise detection AUC**, (ii) **Data pruning R²**. The empirical results are as below, and TimeLAVA exhibits consistently low variance across folds, indicating robustness to the specific choice of reference segments:
> >| Dataset | AUC ↑ | R² (0% remove) | R² (20% remove) | R² (50% remove) |
> >|--------------|-------------------------|--------------------------|--------------------------|--------------------------|
> >| ETTh1| $0.8356 \pm 0.0624$ | $0.6187 \pm 0.0000$ | $0.6347 \pm 0.0035$| $0.6375 \pm 0.0033$|
> >| Exchange| $0.7833 \pm 0.0329$ | $0.9907 \pm 0.0000$ | $0.9909 \pm 0.0001$| $0.9906 \pm 0.0002$|
> >| Traffic| $0.9553 \pm 0.0414$ | $0.5195 \pm 0.0000$ | $0.5389 \pm 0.0026$ | $0.5372 \pm 0.0022$|
> >| Electricity  | $0.6776 \pm 0.0805$ | $0.7858 \pm 0.0000$ | $0.7820 \pm 0.0030$| $0.7769 \pm 0.0055$|
> >
> >Across all four datasets, TimeLAVA demonstrates very low sensitivity to the choice of reference fold. The performance variation is extremely small. In addition, we include experiments that perturb the reference distribution through controlled contamination and drift (see our response to W3&W4 for Reviewer rJ99). The results show that TimeLAVA remains robust under imperfect reference sets.
>
> ---
>
> ### Questions
>
> #### Q3: Large window lengths may face curse of dimensionality with DWT. How to handle large time series classification?
> >**Ans:** We apply DWT on fixed-length overlapping segments rather than on the entire long series, so the feature dimension per segment remains constant while the total cost scales linearly with sequence length. The segment length is determined according to the dataset characteristics (e.g., 96 time steps for Electricity, 168 for Traffic) to cover roughly 1–2 temporal cycles without becoming excessively long. Therefore, the curse of dimensionality issue does not arise in our setting.
>
> ---
>
> #### Q4: In Figure 3, local/global contextual and point anomaly look visually similar - explain differences?
> >**Ans:** These anomaly types differ in their temporal scope: *local contextual anomalies* deviate from *neighbouring windows*, *global contextual* anomalies from *long-term trends*, and *point anomalies* from *individual* values. In Figure 3:
> >- **Local contextual anomaly** shows a segment whose oscillation pattern differs from its nearby regions, where it fluctuates with slightly higher local variance or a shifted phase, even though its overall magnitude remains within the global range.
> >- **Global contextual anomaly** follows the same short-term periodic pattern as its neighbors but is shifted to a lower baseline, deviating from the global statistics of the entire series.
> >- **Point anomaly** is an isolated spike with an extreme value that is clearly abnormal on its own, without requiring contextual comparison.

---

> > ### Comment · Reviewer_WWgi · 2025-11-26
> >
> > I thank the authors for taking my concerns into consideration. Following the discussion, I have some additional questions and remarks:
> > - **Sensitivity to hyperparameters.** I appreciate the experimental results added in the appendix for the parameter k. However, I could not find corresponding results for the parameter c. My concern is specifically about the smoothness of the dependency on c. Adding plots for this sensitivity analysis would be beneficial.
> > - **Regarding W3 (Q2).** Could the authors elaborate on their K-fold procedure for the data-pruning case? The reported 0-variance result seems counter-intuitive.
> > - **Regarding W3 & W4 from Reviewer rJ99.** The contamination case also appears counter-intuitive. If anomalies are injected into the reference distribution, the anomaly scores remain stable, yet one would expect anomalies to be increasingly regarded as normal behavior. Are the injected anomalies and the test anomalies generated from the same AR model?

---

> ### Author Response · Authors · 2025-12-03
>
> Thank you for the reviewer’s follow-up questions and for the opportunity to further clarify these points. We address each concern below and have updated the appendix accordingly.
>
> ---
>
> 1.Sensitivity to hyperparameters:
> > We have now added the sensitivity analysis for both hyperparameters κ and c in Appendix E.4.
>
> ---
> 2.Regarding W3 (Q2): K-fold procedure and the 0-variance result
> > We appreciate the opportunity to clarify this point. (a) 0% removal shows zero variance because the dataset is unchanged. In the 0% removal condition, no data is removed regardless of the reference fold. Thus, the downstream model is always trained on the full dataset, producing identical performance across all folds. The resulting variance of zero is therefore expected rather than counter-intuitive. Variance naturally appears only in the 20% / 50% removal settings, where different reference folds lead to slightly different rankings and thus different subsets being removed. (b) Our K-fold design directly measures sensitivity to the reference set. To isolate sensitivity to the *choice of reference sequences*, we: fix the training set; pool all reference/validation segments, and split them into K = 5 folds. In each fold, only the reference set changes, while training data and model remain fixed. This procedure directly mirrors the role of reference sequences in TimeLAVA and provides a clean measurement of reference sensitivity.
> ---
> 3.Regarding W3 & W4 (contamination experiment)
> > Thank you for raising this concern. The contamination results are consistent with the underlying data-generation mechanism rather than counter-intuitive. (a) All normal segments follow the same AR structure and thus remain close in feature space. (b) Injected anomalies contain random, sparse spike positions and different amplitudes, for example:
> >   * Reference anomaly A₁: spikes at [5, 19, 52]
> >   * Reference anomaly A₂: spikes at [1, 18, 66]
> >   * Test anomaly: spikes at [3, 18, 37]
> Since injected anomalies are random, they do not match each other well and do not form a consistent pattern in the reference set. As a result, a test anomaly cannot find a “good” match among reference anomalies → high transport cost; Normal segments continue to match well with reference normals → low cost. Thus, even when the reference set is contaminated, anomalies are not interpreted as “normal,” and anomaly scores remain stable.
>
> ---
>
> We thank the reviewer for these insightful questions, which have contributed to improving the clarity and completeness of the manuscript.

---

### Official Review · Reviewer_PMWa · 2025-10-31

**Soundness:** 2
**Presentation:** 2
**Contribution:** 2
**Rating:** 2
**Confidence:** 4

**Summary:**

In this study, the authors propose a method to assess the value of individual points and segments in a time series. In particular, they introduce a learning-agnostic measure using a selective Wavelet-based Wasserstein distance, while also providing theoretical guarantees of their approach.

**Strengths:**

1) The authors focus on a very interesting stream of research by measuring the importance of individual time points and patterns in time series.
2) The proposed method achieves remarkable results across multiple tasks.

**Weaknesses:**

Clarity:

1) The paper is difficult to follow. The problem statement and the proposed solution are not clearly conveyed in the abstract. The authors claim to determine the 'value' of a time series point or segment, however, without clarifying the context of the value. Its meaning could vary depending on the downstream task, e.g. the value relevant to a classification task may differ from that in a forecasting task. Moreover, statements such as 'the wavelet transform preserves both time and frequency, enabling precise localization of transient event, which is critical for identifying impactful patterns in time series' (see Figure 1) stand alone and do not clarify what is meant by impactful.

2) The authors refer to TimeInf [1] as Influence in Figure 4 but as TimeInf in Figure 5. Such inconsistencies further reduce the clarity of the manuscript.

Related Works:

3) The related works section seems to be quite limited, focusing on the research of four works. How is the proposed method different to approaches such as RioT [2] or SAVA [3]? The manuscript clearly needs to be revised to highlight its contribution in the field of time series analysis.

Experiments:

4) The description of the baseline methods is too vague. A summary of baselines similar to the one in SAVA [3] would be desirable, since it benefits the clarity of the manuscript.
5) Additionally, a comparison with SAVA [3] would also benefit the manuscript.

___
[1] Kessler, Samuel, Tam Le, and Vu Nguyen. "SAVA: Scalable Learning-Agnostic Data Valuation." International Conference on Learning Representations (2025).

[2] Kraus, Maurice, et al. "Right on time: Revising time series models by constraining their explanations." Joint European Conference on Machine Learning and Knowledge Discovery in Databases (2025).

[3] Zhang, Yizi, et al. "TimeInf: Time Series Data Contribution via Influence Functions." International Conference on Learning Representations (2025).

**Questions:**

1) The authors split the time series into segments of size L using a sliding window approach with stride S. Have the authors set S=1 and iterated L from 1 to T, where T is the entire sequence length, to find the most informative patterns? If not, what is the best approach to do so?
2) Does the proposed approach scale to large datasets?
3) What is the tradeoff between dataset size and data valuation performance?

---

> ### Author Response · Authors · 2025-11-21
> **Response to Reviewer PMWa (Part I)**
>
> We sincerely thank the reviewer for the detailed and insightful feedback, and we truly appreciate the time and effort dedicated to evaluating our work. We address each concern systematically below:
>
> ---
>
> ### Weaknesses
> #### W1 (Clarity): Value context varies by task; Unclear problem statement; Task-specific ambiguity of "value"; Ambiguous meaning of "impactful" (Figure 1)
> >**Ans:** We thank the reviewer for this important feedback. The key clarification is that *value* has a unified definition across all tasks, with task-specific interpretation arising only from the choice of reference distribution.
>
> >**1. Unified Value Definition (Addresses "value context varies by task'')**
> >
> >**Core Principle:** Across all applications, *TimeLAVA* defines value consistently as: *The marginal contribution of a temporal segment to reducing the distributional distance (measured by $W_\text{SW}$) between the evaluated and reference distributions.* This definition is **task-agnostic**, the valuation mechanism remains the same. What changes across tasks is solely the **choice of reference distribution** $\mu_{\text{ref}}$, not the semantics of *value* itself.
>
> >**2. Formal Problem Setup (Clarifies "what is being valued'')**
> >
> >**Given:**
> >- Evaluated time series $X^{\text{eval}} \in \mathbb{R}^{T_{\text{eval}} \times d}$ (potentially contaminated data to be valued)
> >- Reference time series $X^{\text{ref}} \in \mathbb{R}^{T_{\text{ref}} \times d}$ (represents target/desired distribution)
> >- Optional labels $(y^{\text{eval}}, y^{\text{ref}})$ for supervised settings
> >
> >**Valuation Unit:** We segment $X^{\text{eval}}$ into temporal subsequences ${x_i}_{i=1}^N$ (length $L$) to preserve local dependencies, and assign a valuation score to each segment.
> >
> >**Output:** Segment-wise values $v(x_i)$ where:
> >- $v(x_i) > 0$: segment contributes positively to distributional alignment $\Rightarrow$ **beneficial**
> >- $v(x_i) < 0$: segment degrades alignment $\Rightarrow$ **harmful**
>
> >**3. Task-Specific Instantiation (How unified value manifests in applications)**
> >
> >The unified value definition instantiates differently based on **reference set construction**:
> >| **Task**             |**Evaluated Distribution** |**Reference Distribution**  | **Value Interpretation**                                       | **Application**     |
> >|----------------------|-----------------------------|-----------------------------|----------------------------------------------------------------|-------------------------------------------|
> >| Anomaly Detection    | Contaminated time series | Clean normal segments       | $v(x_i) < 0$ flags anomalies deviating from normal         | Unsupervised anomaly identification       |
> >| Data Pruning        |Training time series  | Validation time series      | $v(x_i) < 0$ suggests removal for model improvement        | Dataset cleaning before training |
> >| Data Selection      |Training time series  | Validation time series      | $v(x_i) > 0$ prioritizes informative segments       | Efficient learning|
> >| Label Noise         |Noisy labeled time series  | Clean labeled segments      | $v(x_i) < 0$ identifies mislabeled samples | Label Noise Detection|
> >
> >**Key insight:** The value $v(x_i)$ always measures the same quantity (contribution to $\mathcal{W}_{\text{SW}}$ minimization), but its *interpretation as beneficial/harmful* depends on whether we want to align with (data selection) or diverge from (anomaly detection) the reference.
>
> >**4. Clarifying "Impactful Patterns'' (Addresses Figure 1 comment)**
> >
> >In Figure 1, "impactful patterns'' refer to temporal segments with large data-value magnitudes $|v(x_i)|$. These segments substantially affect the distributional discrepancy between the evaluated data and the reference set. Both beneficial segments that follow the dominant dynamics and harmful segments such as anomalies, mislabeled regions, or regime shifts can be impactful because both types strongly perturb this distributional comparison.
> >
> >Wavelet transforms help identify such segments because many impactful regions contain transient events that vary rapidly in time. Segments with distinctive transient structures produce larger wavelet-based distances, contribute more to the $W_{\text{SW}}$ objective, and therefore obtain larger valuation magnitudes, making them impactful and consistent with Theorem 3.
> >
> >To avoid confusion, we have removed the sentence “which is critical for identifying impactful patterns in time series’’ from the caption.
>
> >**5. Revised Abstract (Addresses clarity concerns)**
> We have revised the abstract to: *"We introduce TimeLAVA, a learning-agnostic framework that values time series segments by quantifying their **marginal contribution to minimizing distributional distance** between training and reference data....''*
> ---
> #### W2 (Clarity): Inconsistent naming - "Influence" vs "TimeInf"
> >**Ans:** Thank you for catching this inconsistency. We have corrected the label in the revised manuscript.

---

> > ### Author Response · Authors · 2025-11-21
> > **Response to Reviewer PMWa (Part II)**
> >
> > #### W3 (Related Works): Limited related work; missing comparison with RioT and SAVA
> > >**Ans:** We thank the reviewer for the helpful feedback. *SAVA* is closely related, as it scales the LAVA framework to large datasets through batch-level optimal transport. In contrast, *RioT* focuses on constraining model explanations through human-in-the-loop feedback to improve **model** reliability, whereas *TimeLAVA* focuses on quantifying the contribution of time-series training **data** segments in a learning-agnostic manner. Therefore, RioT is complementary rather than directly comparable to our approach. We have included explicit discussions of both SAVA and RioT in the revised manuscript, with all corresponding revisions highlighted in red.
> > ---
> > #### W4 (Experiments): Baseline descriptions too vague
> > >**Ans:** We have revised the baseline descriptions in Section 5.2 to provide a more detailed summary, following SAVA.
> > ---
> > #### W5 (Experiments): Missing comparison with SAVA
> > >**Ans:** We have now included *SAVA* in our experiments for **Data Pruning and Selection** and **Noisy-Label Detection**. As shown in the updated results, *SAVA*, a scalable variant of *LAVA*, performs comparably to *LAVA* in noisy-label detection and slightly better in data pruning. However, *TimeLAVA* consistently outperforms both *LAVA* and *SAVA*. A key reason is that the latter two methods rely on standard OT, which enforces full mass matching between training and validation segments. This constraint tends to force mismatched or noisy training segments to be paired with clean validation samples, which can distort the resulting valuation scores. In contrast, *TimeLAVA* allows selective matching by penalizing unmatched mass instead of enforcing full correspondence, and can therefore downweight harmful or regime-shifted segments more effectively. This leads to improved valuations for both time-series data pruning and noisy-label detection tasks in our experiments.
> >
> > ---
> >
> > ### Questions
> >
> > #### Q1: Have you tried S=1 and iterated L from 1 to T to find the most informative patterns? If not, what is the best approach to do so?
> > >**Ans:** Thank you for this insightful question regarding the choice of segment length ($L$) and stride ($S$).
> > We did not set $S{=}1$ and iterate $L$ from 1 to $T$, as $L{=}1$ loses temporal context while $L{=}T$ collapses the sequence into a single segment. Instead, we evaluated a practical range of $L$ values according to the dataset. As shown in our **segment length sensitivity analysis (Appendix E.2.2, Fig. 9)**, we evaluated a practical range of $L$ values (from 5 to 250) on the SMAP and MSL [7] datasets. The AUC increases steadily and stabilizes around $L{=}150$, which means that a moderate segment length capturing 1–2 characteristic temporal periods is sufficient. For the stride parameter, our analysis (Fig. 10) shows that smaller strides consistently yield higher AUC and remain relatively stable for $S \in \{1, 5, 10\}$, but begin to drop for larger values (e.g., MSL: 0.76 $\rightarrow$ 0.68 when $S$ increases from 1 to 50). This occurs as overlapping segments better preserve fine-grained temporal information and prevent missing short anomalies. The best practice is therefore to use a **small stride** for fine-grained coverage and tune **segment length ($L$)** as a **domain-dependent hyperparameter**, typically covering one to two characteristic cycles without exhaustive search. This setup balances contextual completeness, anomaly localization, and computational efficiency.
> >
> > ---
> >
> > #### Q2: Does the proposed approach scale to large datasets?
> > >**Ans:** TimeLAVA scales efficiently to large datasets. As shown in Table 3, our largest experiment (KDD-Cup99, 494,021 time points) completes within 101.52s. The overall computational complexity is $\mathcal{O}(NMLd)$, where $N$ and $M$ denote the numbers of training and reference segments, $d$ is the feature dimension, and $L$ is the segment length. Memory usage scales as $\mathcal{O}(NM)$ for cost-matrix storage. In practice, $M$ (the reference set size) and $L$ (the segment length) are fixed, so runtime increases approximately linearly with the number of training segments, indicating strong scalability in practical settings.
> >
> > ---
> >
> > #### Q3: What is the tradeoff between dataset size and data valuation performance?
> > >**Ans:**  Thank you for the question regarding the trade-off between dataset size and data valuation performance.  A larger dataset provides more segment pairs for the unbalanced OT matching, which reduces the variance of valuation scores and leads to more stable performance. This trend is validated by Tables 1, 3, and 4.  For instance, as dataset size increases (e.g., SMAP: 8k $\rightarrow$ PSM: 87k $\rightarrow$ SWaT: 449k $\rightarrow$ KDD-CUP99: 494k time points),  the AUC of *TimeLAVA* improves from 0.74 to 0.77 to 0.86 to 0.98, and the valuation scores become more stable, confirming the benefit of larger data coverage.

---

> ### Comment · Reviewer_PMWa · 2025-11-26
> **Reviewer's Response**
>
> I would like to thank the authors for their responses. I acknowledge that I have read the rebuttal and the responses to the other reviewers. While an interesting stream of research, I remain unconvinced and believe the proposed study is not ready for publication. The presented paper is still difficult to follow, leaving the reader unclear with ambiguous definitions. Related works such as RioT by Kraus et al. [1] is included in the related work section in Appendix B, while not mentioned in the related work section of the main paper. What is even the purpose of having two related work sections? While the authors claim that the model scales to large datasets, they only elaborate on the runtime without addressing model performance. Given such concerns, I cannot recommend accepting the study at this time, and thus leave my scores unchanged. However, I strongly encourage the authors to further improve the quality of the study, as the conceptual idea is interesting.
> ___
> [1] Kraus, Maurice, et al. "Right on time: Revising time series models by constraining their explanations." Joint European Conference on Machine Learning and Knowledge Discovery in Databases (2025).

---

> > ### Author Response · Authors · 2025-12-03
> >
> > Thank you for taking the time to read our rebuttal and for providing additional comments. We sincerely appreciate the constructive feedback and have carefully reflected on all points raised.
> >
> > ---
> >
> > 1. Clarity and organization
> >
> > > We have substantially revised the writing to remove ambiguities and improve readability throughout the paper. In particular, we:
> > > * consolidated all related-work discussions into a single section,
> > > * reformulated the problem setup with clearer notation and definitions,
> > > * added a workflow figure summarizing the TimeLAVA pipeline, and
> > > * provided a more detailed explanation of the baselines in the experimental section.
> >
> > > We believe these changes significantly improve the overall clarity and presentation.
> >
> > ---
> >
> > 2. Related work, including RioT
> >
> > > Thank you for pointing out the issue regarding the structure of the related-work section and the mention of RioT. The original submission included a brief related-work discussion in the introduction and, due to strict page limits, did not provide a standalone related-work section, with additional material placed in the appendix. Following the reviewers’ suggestions, we added a dedicated related-work section to the main paper, which led to two separate discussions and may have caused confusion. In the revision, we have consolidated these into a single unified related-work section in the main text and moved the discussion of RioT from the appendix into this section to clarify its relationship to our problem setting.
> >
> > ---
> >
> > 3. Scalability and performance
> >
> > > We would like to note that the scalability and performance concerns raised here were already addressed in detail in our rebuttal responses to Q2 and Q3. Q2 discussed computational scalability with both theoretical complexity and large-scale runtime results, while Q3 analyzed how valuation performance evolves with increasing dataset size. We understand, however, that these points were not sufficiently emphasized in the original submission, and in the revised version we have made them more visible and reorganize the relevant results to clearly highlight both aspects.
> >
> > ---
> >
> > We appreciate the reviewer’s thoughtful feedback and the positive remark about the conceptual idea. These comments have been valuable in guiding us to further refine the clarity and presentation of the work.

---

### Official Review · Reviewer_rJ99 · 2025-10-31

**Soundness:** 3
**Presentation:** 3
**Contribution:** 3
**Rating:** 8
**Confidence:** 3

**Summary:**

This paper proposes a model-agnostic data valuation framework (TIMELAVA) for time series, designed to quantify the importance of individual time points or temporal segments. By combining multi-scale wavelet transforms with unbalanced optimal transport, it defines the Selective Wavelet–based Wasserstein (WSW) distance to capture local time–frequency features while handling non-stationarity.

**Strengths:**

1. TIMELAVA effectively integrates wavelet analysis with unbalanced optimal transport and develops a novel distance metric, overcoming the limitations of existing data valuation methods in handling temporal dependencies and non-stationarity.

2. The use of entropy regularization ensures computational efficiency, avoiding repeated retraining or perturbation sampling; this speed advantage is clearly demonstrated in experiments.

3. The empirical evaluation is comprehensive and diverse, covering anomaly detection, data selection, and noisy label detection, demonstrating TIMELAVA’s generality and wide applicability.

**Weaknesses:**

1. Experimental details are heavily relegated to the appendix. The main text provides relatively brief descriptions of settings, parameters, and results, which may affect readability and reproducibility. For example, in Section 5.1, univariate dataset results are not reported; some datasets (e.g., SMD) are duplicated; Table 1 claims five multivariate datasets but actually only lists four; and Section 5.3 lacks baseline description.

2. Anomaly scores should also be reported and compared against the clean data to assess discriminative power.

3. TIMELAVA relies strongly on the representativeness of the reference set. If the reference distribution is shifted or incomplete, rare but valuable patterns may be underestimated. While this limitation is acknowledged, the paper lacks a detailed discussion of the extent of this impact and strategies to reduce it.

4. It remains unclear whether TIMELAVA’s valuations are reliable if the reference set itself contains anomalies or exhibits distributional drift.

5. No discussion or comparison of computational efficiency (runtime, memory, complexity), despite claiming efficiency.

**Questions:**

1. Make the Experiments section more self-contained and address inconsistencies in writing.
2. Include a discussion of computational efficiency.
3. Evaluate the impact of the reference set on performance.

---

> ### Author Response · Authors · 2025-11-21
> **Response to Reviewer rJ99 (Part I)**
>
> We sincerely thank the reviewer for the constructive feedback and positive assessment of our work, and we appreciate the time and effort devoted to reviewing our submission. We address each weakness and question below:
>
> ---
> ### Weaknesses
>
> #### W1 (Q1): Experimental details heavily relegated to appendix; inconsistencies (univariate results, SMD duplication, dataset count mismatch, missing baseline descriptions)
>
> >**Ans:** We appreciate the reviewer identifying these presentation issues and have made the following specific revisions:
> >1. **Univariate results:** Results for the univariate NAB datasets are provided in Table 4 (Appendix E.2.1). The UCR univariate dataset is used mainly to illustrate different anomaly types, following *TimeInf*. Specifically, the **qualitative analysis** in Figure 4 and the **comprehensive comparison** in Figure 10 of the revised manuscript together present the full visual comparison results across all methods on representative anomaly types, demonstrating *TimeLAVA*’s superior localization.
> >1. **SMD duplication:** Removed the duplicated mention of “SMD” in the main text.
> >1. **Dataset count correction:** We have updated the manuscript to correct this issue.
> >1. **Section 5.3 baseline descriptions:** The baseline statement previously placed under “Datasets.’’ has been moved to the end of the Section 5.3 "Experimental Setup." for better visibility.
> >
> > All revisions are **highlighted in red** in the revised manuscript.
> ---
>
> #### W2: Anomaly scores should also be reported and compared against the clean data to assess discriminative power.
>
> >**Ans:**  We thank the reviewer for this suggestion. The reported AUC values quantify how well each method discriminates between clean and anomalous segments, as AUC represents the probability that a randomly chosen clean segment receives a higher score than a randomly chosen anomalous one, which is a standard measure of discriminative power. Additionally, we have now included the comparison of anomaly score distributions between normal and anomalous segments to *visually* assess this discriminative capability across different methods (*TimeLAVA*, *TimeInf*, *IsolationForest*, and *AnomalyTransformer*) on the *PSM* dataset in Appendix E.2.1. As shown in the updated Figure 9, *TimeLAVA* exhibits the clearest separation with minimal overlap, indicating the strongest discriminative power among all methods.
>
> ---
>
> #### W5 (Q2): No discussion or comparison of computational efficiency (runtime, memory, complexity), despite claiming efficiency.
>
> >**Ans:** We thank the reviewer for this helpful suggestion. We thank the reviewer for this helpful suggestion. The table below reports both theoretical time and memory complexities, as well as empirical runtime and peak memory usage on the Moving dataset (temporal label-noise detection task). As shown in the table, all learning-agnostic approaches (LAVA, SAVA, TimeLAVA) avoid model retraining and are therefore generally more efficient than model-dependent methods such as TimeInf (even under the simple AR setting) or tree-based approaches such as DataOOB. KNNShapley provides the most efficient solution, although it is purely distance-based, which results in generally weaker performance. While TimeLAVA has the same theoretical complexity order as LAVA, it consistently achieves higher valuation accuracy. [$N$ and $M$ denote numbers of evaluated and validation segments, $L$ the segment length, $d$ the feature dimension, $\epsilon$ the approximation error rate, $K$ number of neighbors in KNNShapley, $N_b,M_b$ batch sizes in SAVA, $B$ number of trees in DataOOB, and $q$ model parameter dimension in TimeInf.]
> >| **Method**      | **Time Complexity**  | **Space Complexity**  | **Runtime (s)** | **Peak Memory (GB)** |
> >|-----------------|----------------------------------------------|--------------------------------|----------------|----------------------|
> >| **LAVA**        | $\mathcal{O}(NMLd)$| $\mathcal{O}(NM)$  | 544  | 6.4    |
> >| **SAVA**        | $\mathcal{O}(NMLd)$   | $\mathcal{O}(N_bM_b)$ | 472  | 3.2 |
> >| **KNNShapley** | $\mathcal{O}(NMLd/\epsilon^2\log^2(K))$      | $\mathcal{O}(Nd)$ | 443  | 1.8  |
> >| **DataOOB**    | $\mathcal{O}(BNLd\log N)$ | $\mathcal{O}(Nd+BN)$ | 690  | 2.0  |
> >| **TimeInf**     | $\mathcal{O}(Nq^{2} + q^{3})$| $\mathcal{O}(Nq + q^{2})$ | 710 | 6.8  |
> >| **TimeLAVA**    | $\mathcal{O}(NMLd)$  | $\mathcal{O}(NM)$  | 463 | 5.1  |

---

> ### Author Response · Authors · 2025-11-21
> **Response to Reviewer rJ99 (Part II)**
>
> #### W3 & W4 (Q3): *TimeLAVA* relies strongly on the representativeness of the reference set. If the reference distribution is shifted or incomplete, rare but valuable patterns may be underestimated. While this limitation is acknowledged, the paper lacks a detailed discussion of the extent of this impact and strategies to reduce it. It also remains unclear whether *TIMELAVA*’s valuations are reliable if the reference set itself contains anomalies or exhibits distributional drift.
>
> >**Ans:** We appreciate the reviewer’s observation regarding the dependence of *TimeLAVA* on the representativeness of the reference set. This dependency is a general limitation shared by most *validation-based* data valuation frameworks, as discussed in [1]. To quantify its impact and assess the reliability of TimeLAVA under imperfect references, we conducted a new robustness study:
> >
> >- **Experimental setup.** We generate synthetic AR(1) time series, and evaluate **segment-level anomaly detection AUC**. Each run includes $200$ normal, $20$ anomalous segments, and experiments are repeated $5$ times with different random seeds. TimeLAVA is configured with wavelet=db4, level$=2$, UOT regularization $\kappa=2$.
> We test three imperfections of the reference set:
>     1. **Contamination:** a fraction $\alpha \in \{0.0, 0.1, 0.3, 0.5\}$ of reference segments is substituted with anomalous ones.
>     2. **Incompleteness:** normal data contain two regimes: Type A ($\rho=0.8$) and rare Type B ($\rho=0.3$). The evaluation data contain two normal regimes (Type A with $\rho=0.8$ and rare Type B with $\rho=0.3$), while the reference set progressively omits Type B segments to simulate an incomplete view of the normal distribution. The reference progressively removes Type B with missing ratios $\{0, 0.3, 0.5, 0.7\}$.
>     3. **Distributional drift:** the training data mean is shifted by $\Delta \in \{0.0, 0.2, 0.5\}$ relative to the reference.
>
> >- **Results**: The table belows shows **TimeLAVA robustness under imperfect reference sets** (5-run mean ± std). AUC remains ≳ 0.90 even under contamination, missing reference coverage, or shifted reference.
>     >
>     >| Setting | Parameter | AUC (mean ± std) | Rel. change |
>     >|---------|-----------|------------------|-------------|
>     >| **(a) Reference contamination** | | | |
>     >| $\alpha$ = 0.0 | Clean ref. | 0.9438 ± 0.034 | -- |
>     >| $\alpha$ = 0.1 | 10% anomalies | 0.9513 ± 0.014 | +0.7% |
>     >| $\alpha$ = 0.3 | 30% anomalies | 0.9407 ± 0.042 | -0.3% |
>     >| $\alpha$ = 0.5 | 50% anomalies | 0.9373 ± 0.015 | -0.7% |
>     >| **(b) Reference incompleteness (missing Type B)** | | | |
>     >| 0% missing | -- | 0.9485 ± 0.009 | -- |
>     >| 30% missing | -- | 0.9611 ± 0.016 | +1.3% |
>     >| 50% missing | -- | 0.9365 ± 0.008 | -1.3% |
>     >| 70% missing | -- | 0.9561 ± 0.017 | +0.8% |
>     >| **\(c\) Distributional drift (mean shift)** | | | |
>     >| $\Delta$ = 0.0 | No shift | 0.9556 ± 0.009 | -- |
>     >| $\Delta$ = 0.2 | Mild shift | 0.9448 ± 0.023 | +1.1% |
>     >| $\Delta$ = 0.5 | Moderate shift | 0.8997 ± 0.029 | -5.9% |

---

> > ### Author Response · Authors · 2025-11-21
> > **Response to Reviewer rJ99 (Part III)**
> >
> > #### W3 & W4 (Q3): continued.
> > >- **Findings**
> >     >-  Under **contamination**, even when 50\% of the reference segments are anomalous ($\alpha=0.5$), AUC$=0.94\pm0.02$, nearly unchanged from the clean baseline ($0.94\pm0.03$). Interestingly, for mild contamination levels ($\alpha = 0.1$), the AUC slightly increases. This might be due to the regularizing nature of the UOT formulation, which tolerates small mismatches and may even smooth the transport geometry, leading to slightly more stable valuations.
> >     >-  Under **incompleteness**, segments from the rare regime Type B receive consistently lower values than the common Type A segments (value $\approx0.4$ vs $\approx 4$ for Type A), reflecting that the reference provides an incomplete view of the normal dynamics. However, the overall anomaly-detection AUC remains in the range $0.94$--$0.96 \pm 0.02$, and Type B scores are still well separated from those of  true anomalies (value $\approx -6$). This indicates that, although rare regimes are mildly undervalued when they are underrepresented in the reference, TimeLAVA does not confuse them with anomalies and the global discrimination between 'normal' and 'anomaly' remains intact.
> >     >- Under **distributional drift**, even under a moderate shift ($\Delta=0.5$), TimeLAVA maintains AUC ≈ 0.90, indicating that while unmatched regions in the training data receive lower values, the overall discrimination between normal and anomalous segments remains strong. This behavior aligns with the theoretical property of unbalanced OT,  which gracefully down-weights shifted mass rather than corrupting all valuations.
> > >
> > >These results confirm that TimeLAVA’s valuations are stable to moderate contamination, incompleteness, and drift, also consistent with the theoretical intuition that the unbalanced OT formulation can down-weight unmatched or contaminated mass.
> >
> > >Regarding possible mitigation strategies, [1] utilizes the cross-validation error as a surrogate to evaluate the model’s performance on a validation set to reduce dependence on explicit validation sets. However, these approaches are inherently model-dependent and therefore not directly applicable to our learning-agnostic, distribution-distance framework. Recent advances [2, 3] provide promising complementary directions. [2] constructs a mixture or proxy reference distribution to approximate the ideal validation distribution when the reference is incomplete, while [3] defines data values based on the worst-case generalization over an uncertainty set of possible distributions. Integrating such mixture-reference construction or distributionally robust objectives into TimeLAVA could further enhance its stability under imperfect references, and we view this as a promising future extension.
> > ---
> >
> > We hope that our clarifications and additional analyses sufficiently address all of your concerns. We greatly appreciate your time and thoughtful review.
> >
> > ### References
> > [1] Jahagirdar et al. "Data Valuation in the Absence of a Reliable Validation Set." TMLR, 2024.
> >
> > [2] Xu, Xinyi et al. "Data distribution valuation." NeurIPS, 2024.
> >
> > [3] Lin, Xiaoqiang et al. "Distributionally robust data valuation." ICML, 2024.

---

### Author Response · Authors · 2025-12-03
**Summary for the Reviewers and Area Chair**

We sincerely thank all reviewers and the AC for the time and effort devoted to evaluating our submission! Your careful reading and thoughtful questions have helped improve both the manuscript and the overall presentation. Below we provide a concise summary of how the revised manuscript and rebuttal collectively respond to the main concerns raised in the reviews.

Our work introduces a **time-series data valuation framework**, where each segment is assigned a principled value indicating its utility. This enables identifying important or harmful temporal segments in a model-agnostic manner. Reviewers rJ99, WWgi, and 64PL noted that the paper is well written, and all reviewers agreed that the setting and conceptual idea are interesting and novel.

---

**1. Clarity Improvements**

In response to comments from reviewers PMWa and NTyq, we made several revisions aimed at improving clarity:

* Added a new **workflow diagram**, improved **problem setup**, and provided more detailed descriptions of the **experiment baselines** for better readability.
* Integrated **related work** into the main text for smoother conceptual flow.
* In the rebuttal, we provided a clearer **logic flow of theorems**, explaining how the theoretical components justify each step of the TimeLAVA pipeline.

We appreciate the reviewers for pointing out these aspects, which helped strengthen the presentation of the paper.

---

**2. New Experiments**

To address the main technical concerns, we conducted several additional experiments:

- **Robustness to the Reference Set:** New evaluations demonstrate stability under: contamination, distribution shift, and rare / incomplete reference patterns and across different splits of the reference data used for valuation.
- **Parameter Sensitivity:** UOT regulariser κ and balance parameter c sensitivity analyses (Appendix) show broad stability.
- **Deep Model Validation:** We evaluated TimeLAVA’s pruning effectiveness on CycleNet and TimeMixer, both with and without injected noise, demonstrating its applicability beyond AR models.

---

**Conclusion**

We have made substantial revisions and added new experiments in response to reviewer questions related to clarity, theoretical motivation, reference-set sensitivity, parameter sensitivity, and deep-model validation. We believe these updates meaningfully strengthen the submission and help clarify the contributions of the proposed framework.

We hope this summary supports your decision. Thank you for your consideration.

Best regards,
Authors of Paper 6496

---

### Meta-Review · Area_Chair_gY8W · 2026-01-05

**Summary:**

This paper proposes TimeLAVA, a learning-agnostic framework for time-series data valuation based on a selective wavelet-based Wasserstein discrepancy. The problem setting is timely and relevant, and the paper demonstrates a solid level of technical competence, with non-trivial theoretical development and a broad set of experiments. Several reviewers acknowledge the originality of combining wavelet representations with unbalanced optimal transport for temporal data valuation.

However, after carefully considering the full set of reviews and the authors’ rebuttal, I conclude that the submission does not meet the acceptance bar for ICLR. The primary issue is not correctness, but insufficient clarity, impact, and justification relative to the complexity of the proposed framework. While the method is mathematically involved, the paper does not clearly articulate a compelling use case where such a sophisticated valuation mechanism is necessary or decisively superior to simpler or existing alternatives.

More importantly, the paper struggles to establish a clear and convincing narrative of practical value. The definition of “data value” remains abstract, and its interpretation is highly dependent on the choice of reference distribution, which weakens the claim of a unified, task-agnostic valuation principle. As a result, it is difficult to assess how the proposed method would be adopted or prioritized in real-world time-series pipelines beyond carefully controlled experimental settings.

Although the rebuttal adds experiments and clarifications, it primarily improves completeness rather than resolving the core concerns. The limited practical insight, marginal empirical advantage relative to the added complexity, and lack of a strong, clear takeaway prevent this work from reaching the level of significance expected at ICLR.

**Reviewer Concerns:**

Several key concerns remain insufficiently resolved:

The practical motivation and real-world impact of the proposed valuation framework remain unclear.

The notion of “data value” is still abstract and strongly dependent on the choice of reference distribution, which weakens the claim of a unified, task-agnostic principle.

The methodological and theoretical complexity is substantial relative to the empirical improvements demonstrated.

While additional experiments improve completeness, they do not fundamentally change the overall assessment of contribution and impact.

**Reviewer Scores:**

While the rebuttal added clarifications and additional experiments, it is unlikely that these changes would have led to meaningful score increases. Reviewers who initially expressed strong concerns regarding conceptual clarity, practical impact, and cost–benefit tradeoffs would likely maintain their original scores, as the rebuttal does not fundamentally alter these aspects. Borderline reviewers may have acknowledged the additional effort but would still be unlikely to revise their scores upward sufficiently to support acceptance.

---

### Decision · Program_Chairs · 2026-01-26

Reject